# Miocene African topography induces decoupling of Somali Jet and South Asian summer monsoon rainfall

Zixuan Han [1,2,3], Niklas Werner [2], Zhenqian Wang [2], Xiangyu Li [4,5], Zhengquan Yao [6,7] & Qiong Zhang [2] ✉

The Miocene epoch, marked by significant tectonic and climatic shifts, presents a unique period to study the evolution of South Asian summer monsoon (SASM) dynamics. Previous studies have shown conflicting evidence: wind proxies from the western Arabian Sea suggest a weaker Somali Jet during the Middle Miocene compared to the Late Miocene, while rain-related records indicate increased SASM rainfall. This apparent decoupling of monsoonal winds and rainfall has challenged our understanding of SASM variability. Here, using the fully coupled EC-Earth3 model, we identify a key driver of this decoupling: changes in African topography rather than other external forcings such as $CO_2$ change. Our simulations reveal that changes in Miocene African topography weakened the cross-equatorial Somali Jet and reduced upwelling in the western Arabian Sea, while simultaneously enhancing monsoonal rainfall by inducing atmospheric circulation anomalies over the Arabian Sea. The weakened Somali Jet fostered a positive Indian Ocean Dipole-like warming pattern, further amplifying the monsoonal rainfall through ocean-atmosphere feedbacks. In contrast, $CO_2$ forcing enhances both Somali Jet and rainfall simultaneously, showing no decoupling effect. These findings reconcile the discrepancies between wind and rainfall proxies and highlight the critical role of African topography in shaping the multi-stage evolution of the SASM system.

The South Asian summer monsoon (SASM), a major component of the global monsoon system[1–3], has a strong impact on the Indo-Pacific moisture transport and governs the stability of the present-day Indian Monsoon. Over geological timescales, the strength of the SASM has fluctuated significantly, as evidenced by multiple proxy records. During the Miocene, significant tectonic activity, atmospheric $CO_2$ fluctuations, and environmental changes[4,5], influenced monsoon dynamics, drawing considerable interest in the evolution and key drivers of SASM variability[6,7].

Marine proxies indicate a major intensification of SASM circulation, particularly the Somali Jet in the western Arabian Sea during the Late Miocene after 8 Ma[8,9]. However, sediment archives from the western Arabian Sea[10–13] and Maldives archipelago[14] suggest that the strengthening of SASM circulation might have commenced as early as the late Middle Miocene (~12.9 Ma). These records link wind-driven coastal upwelling in the western Arabian Sea with low-level tropospheric SASM winds, which drive surface waters offshore, bringing cold, nutrient-rich waters into the euphotic zone and enhancing

[1]Key Laboratory of Marine Hazards Forecasting, Ministry of Natural Resources, Hohai University, Nanjing, China. [2]Department of Physical Geography and Bolin Centre for Climate Research, Stockholm University, Stockholm, Sweden. [3]College of Oceanography, Hohai University, Nanjing, China. [4]Department of Atmospheric Science, School of Environmental Studies, China University of Geosciences, Wuhan, China. [5]Centre for Severe Weather and Climate and Hydrogeological Hazards, Wuhan, China. [6]Key Laboratory of Marine Geology and Metallogeny, First Institute of Oceanography, Ministry of Natural Resources, Qingdao, China. [7]Laboratory for Marine Geology, Qingdao Marine Science and Technology Center, Qingdao, China. ✉e-mail: qiong.zhang@natgeo.su.se

primary production. Nonetheless, the interpretation of wind-based proxies has been challenged by the emergence of rain-related proxies that present a different perspective on SASM rainfall variability[15–21].

Studies of oxygen isotopes ($\delta^{18}O$) from pedogenic carbonates[15,16] and mammalian teeth[22] from the Siwaliks Hills suggest a decline in SASM rainfall since the Middle Miocene. This decline corresponds with a shift in vegetation from forest ecosystems to grassland-dominated landscapes in South Asia[15,22]. This reduction in the SASM rainfall after around 12 Ma is further supported by proxies indicating a weakening of chemical weathering and erosion, including decreased K/Al ratios and clastic fluxes into the ocean[18]. Lower chemical weathering rates and sediment transport imply drier conditions and a weakening of SASM.

Recent studies attribute the SASM circulation and rainfall intensity primarily to the uplift of Himalaya-Tibetan Plateau[3,23,24], the expansion of polar ice sheet development[25,26] or variations in $pCO_2$[27] during the Miocene. However, discrepancies between oceanic wind-based records and terrestrial rain-related records suggest that additional boundary forcings contributed to the multi-stage evolution of the SASM system. Reconstructions of Africa's dynamic topography over the past 30 Ma indicate a gradual uplift of the East African Rift System, including the Ethiopian and the East African plateaus, and the concurrent subsidence of the Congo Basin by up to 500 meters[28]. Notably, these tectonic changes in Africa aligns with the emergence of a "modern-like" SASM during the late Middle Miocene (~13 Ma)[10–14,29]. Recent modeling studies further highlight the sensitivity of the SASM to the uplift of the East African Highlands since the Miocene[6,7], reinforcing the intricate link between African topography, SASM circulation, and regional rainfall. Despite these insights, the mechanisms underlying the apparent decoupling of Somali Jet and rainfall between the Middle to Late Miocene remain poorly understood. Previous modeling efforts have limited by low-horizontal resolution in atmospheric general circulation models or have overlooked the actual African topography of the Miocene, as well as the oceanic feedbacks induced by topographic changes[6,7,30–32]. These limitations complicate efforts to quantify the role of African topography in SASM evolution.

To address this gap, we investigate the climate response to reconstructed African dynamic topography at ~25 Ma (Late Oligocene; referred to here as Early Miocene simulations), ~15 Ma (Middle Miocene), ~5 Ma (Late Miocene), and present-day (pre-industrial) using a high-resolution configuration of the fully coupled EC-Earth3 model. We also perform sensitivity experiments using a linear baroclinic model (LBM) to identify the dominant mechanisms driving the simulated SASM changes (see Methods and Supplementary Figures for a detailed description). Our results demonstrate that the decoupling of Somali Jet and SASM rainfall is primarily driven by changes in African topography during the Miocene, operating through positive ocean-atmosphere feedbacks.

## Results

### Miocene African topography changes cause the decoupling of Somali Jet and SASM rainfall

We use the coupled EC-Earth3 model to simulate the response of the SASM to changes in African topography since the Miocene (see Methods). The model successfully replicates the present-day SASM system, which is characterized by a strong cross-equatorial Somali Jet over the Arabian Sea and increased rainfall over the Indian subcontinent during boreal summer (June to August) (Supplementary Fig. 1a, b). Due to the Coriolis force, the climatological southwesterly summer monsoon winds over the Arabian Sea can drive strong ocean upwelling along the coastal region (Supplementary Fig. 1c, d). The agreement between our pre-industrial simulation and contemporary observations validates the use of EC-Earth3 for exploring the influence of the African topography on the SASM.

Our simulations reveal a notable decoupling of the cross-equatorial Somali Jet and rainfall from Middle Miocene to the present-day in response to changes in African topography. The Miocene epoch was characterized by substantial tectonic activity, especially within the East African Rift System (EARS). Reconstructions of Africa's dynamic topography over the past 30 Ma[28] indicate a gradual uplift of the EARS, including the Ethiopian and the East African plateaus, and subsidence of the Congo Basin by up to 500 meters (Fig. 1a–c and Supplementary Fig. 2). Our simulations show a significant weakening of the meridional cross-equatorial flow of the Somali Jet (Fig. 1d–f and Supplementary Fig. 3), with reductions of $0.36 \, m \, s^{-1}$ and $0.22 \, m \, s^{-1}$ in the Early and Middle Miocene, respectively, compared to the present-day (Fig. 1g). These findings are largely in agreement with previous modeling studies[31–33].

Accompanied by the weakened Somali Jet, upwelling in the western Arabian Sea is significantly reduced in the Early and Middle Miocene (Fig. 1g and Supplementary Fig. 4). This simulated upwelling change aligns with wind-based proxy records (see Methods), which indicate a weakening Somali Jet-induced oceanic upwelling in the western Arabian Sea during the Middle Miocene compared to Late Miocene (Fig. 1i, j). However, recent publication has advanced our knowledge that the weakening of upwelling is more complex than a simple reduction in wind strength and is attributed to changes in both the intensity and structure of the Somali Jet[34]. Specifically, while the jet weakens in its core, low-level tropospheric winds strengthen along the western Arabian Peninsula (Supplementary Fig. 3). This altered wind pattern decreases the wind-stress curl along the western Arabian Sea coast[34–36], where the proxy sites are located, resulting in negative Ekman pumping and weaker upwelling (Supplementary Fig. 4).

In present-day climate, the East African topography acts as a barrier that channels the easterly Somali Jet, yielding intense rainfall over the Indian subcontinent. Given the established relationship between the Somali Jet and SASM rainfall, one would expect the topography-induced weakening of the Somali Jet to result in reduced SASM rainfall due to decreased moisture transport[37]. However, our results indicate a spatially coherent increase in summer rainfall, with the maxima in the northeast of the Indian subcontinent and south of the Western Ghats region during the Early and Middle Miocene (Fig. 1d, e). Specifically, SASM rainfall increases by 15% and 9% during the Early and Middle Miocene, respectively, compared to the present-day (Fig. 1h). This simulated increase in SASM rainfall is consistent with qualitative rainfall-based proxies for the Miocene (see Methods), which suggest more humid conditions over the South Asia during the Middle Miocene compared to the Late Miocene (Fig. 1k–o).

Our experiments successfully reproduce the major observational features of Somali Jet-induced upwelling and monsoon rainfall decoupling from Middle to Late Miocene, as seen in proxy records. This highlights the significant role of altered African topography in modulating the SASM system during the Miocene, offering a previously unrecognized explanation for the observed discrepancies between wind-based and rainfall-based proxy reconstructions.

During the boreal summer, the East African Highlands act as a major topographic barrier, redirecting low-level easterly winds over the Southern Indian Ocean across the equator, forming the south-westerly Somali Jet[38] (Supplementary Fig. 1). Numerous idealized modeling studies have demonstrated that removing or lowering the East African Highlands allows easterly winds south of the equator to penetrate further inland over Africa, creating a broader, less concentrated cross-equatorial flow, rather than channeling it into a strong Somali Jet[31–33].

Our simulations support this mechanism, showing a significant weakening of the core of Somali Jet in response to the lowered East African Highlands during the Miocene (Fig. 1, Fig. 2a, d, g, and Supplementary Fig. 3). This highlights the critical role of reduced topographical blocking in weakening the cross-equatorial Somali Jet,

consistent with previous modeling studies[6,31–33]. Consequently, the weakened Somali Jet during the Early and Middle Miocene results in a marked downwelling motion anomaly along the coast of East Africa (Fig. 2c, f). This would have reduced coastal upwelling and primary productivity[39], affecting wind-based proxies such as TOC and *G. bulloides* in the western Arabian Sea (Fig. 1).

## Dynamic effects induced by atmospheric circulation anomalies control the changes in SASM rainfall

The significant increase in SASM rainfall during the Early and Middle Miocene, despite the weakened Somali Jet, is primarily associated with anomalous cyclonic circulation over Arabian Sea and an anticyclonic pattern over the Bay of Bengal (red bold vectors in Fig. 1d, e). These circulation patterns enhance meridional moisture transport from the tropical Indian Ocean to the SASM region (Supplementary Fig. 5), providing a dynamic explanation for the increased precipitation.

This hypothesis is further confirmed by a linearized moisture budget analysis (see Methods). The moisture budget decomposition reveals that the combined thermodynamic and dynamic terms (Supplementary Fig. 6a, d and g) effectively capture the primary characteristics of SASM rainfall changes during the Early, Middle and Late Miocene (Fig. 1d–f), with pattern correlation coefficients of 0.90, 0.88,

and 0.67, respectively. The increased SASM rainfall is mainly driven by dynamic effects resulting from changes in atmospheric circulation (Supplementary Fig. 6c, f and i), particularly during the Early and Middle Miocene, whereas the thermodynamic effects due to atmospheric moisture changes are negligible (Supplementary Fig. 6b, e, and h). It is important to note that this dynamic response does not contradict the hypothesis of a decoupling between monsoon winds and rainfall, but rather provides a mechanistic explanation for how SASM rainfall increased despite a weaker Somali Jet, emphasizing the importance of atmospheric circulation changes and ocean-atmosphere feedbacks in modulating the SASM system.

The analysis indicates that cyclonic and anticyclonic circulation anomalies over the northern Indian Ocean are the key drivers of increased SASM rainfall during these periods (Fig. 1d–f and Supplementary Fig. 5). This raises a critical question: how do changes in African topography induce these atmospheric circulation anomalies that, in turn, affect the SASM rainfall? The following section explores the role of ocean-atmosphere interaction in this process.

## Increased SASM rainfall due to direct atmospheric processes

The anomalous cyclonic circulation over the Arabian Sea in the low-level troposphere is directly linked to a significant reduction in

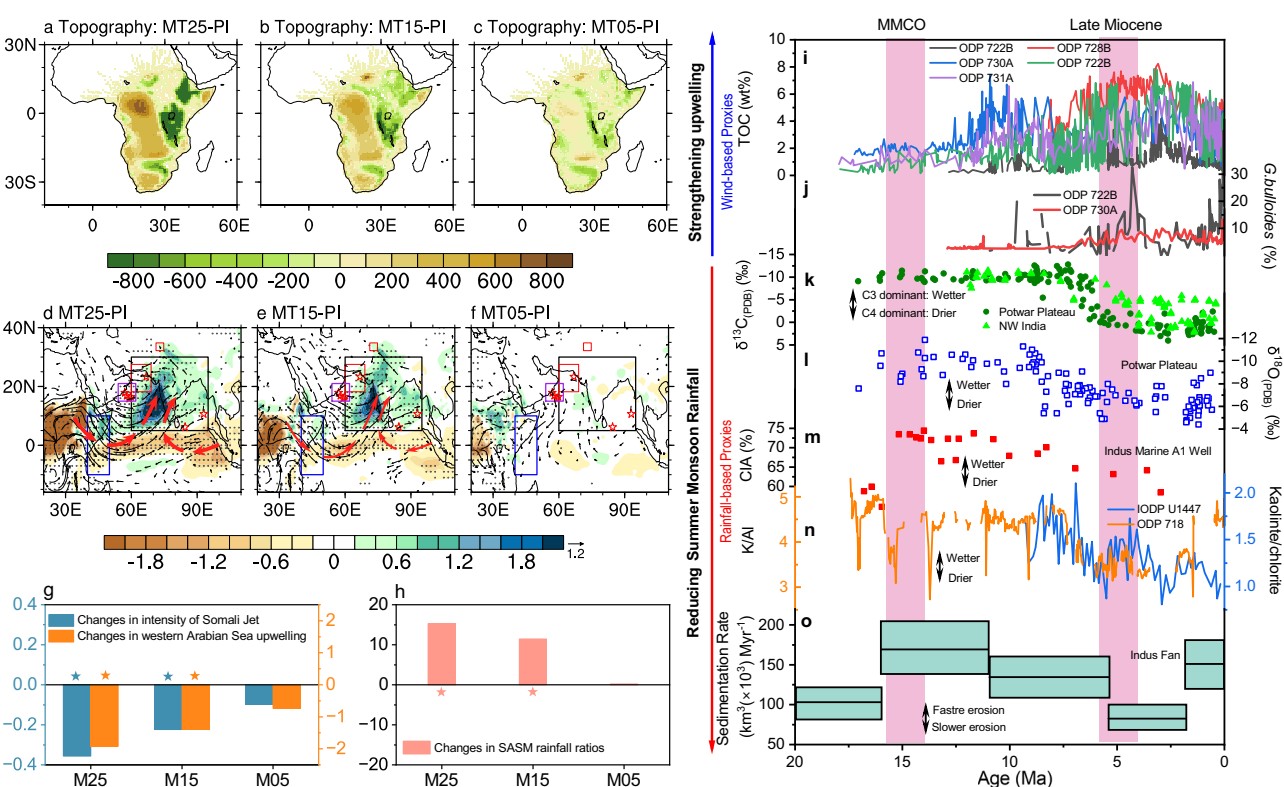

**Fig. 1 | Response of simulated South Asian summer monsoon (SASM) to African topography changes during the Miocene in the coupled EC-Earth3 experiments, and comparisons with SASM proxy records.** Reconstructed African Miocene topography (m) for (**a**) Early Miocene, (**b**) Middle Miocene and (**c**) Late Miocene, based on the paleo-topography map from ref. 28. Changes in rainfall (shading, mm day⁻¹) and low-level atmospheric circulation at 700 hPa (vectors, m s⁻¹) in the (**d**) MT25, (**e**) MT15 and **f** MT05 simulations relative to the pre-industrial simulations. In (**d–f**) gray stippling and vectors denote regions in which the changes are significant at the 95% confidence level according to Student's *t*-test. And the solid black boxes mark the SASM region (5° N-30° N, 60° -95° E), and the red asterisks or boxes denote the locations of the proxy records in (**i–o**). The red bold vectors in (**d**) and (**e**) indicate the cyclonic/anticyclonic circulation anomalies. The changes in (**g**) intensity of Somali Jet (m s⁻¹) and western Arabian Sea upwelling (cm day⁻¹), and (**h**) ratios of SASM rainfall (%), with asterisks above or below the bars denoting changes that are significant at the 95% confidence level according to Student's *t*-test. The solid blue boxes (10° S–10° N, 40° −50° E) and solid purple boxes (15° −21° N, 54° −62° E) in (**d–f**) are used to define the intensity of Somali Jet and western Arabian Sea upwelling, respectively (see Methods). **i** Total organic carbon (TOC, wt%) values from ODP Sites 722B, 728B, 730 A, and 731A[10,11]. **j** Planktic foraminifer *Globigerina bulloides* (*G. bulloides*; %) from ODP 722B[11] and 730A[9]. **k** δ¹³C (PDB) of paleosol carbonate (‰) from the Potwar Plateau[15] and NW India[87]. **l** δ¹⁸O (PDB) of soil carbonate (‰) from the Potwar Plateau[15]. **m** Chemical weathering proxies (CIA, %) from the Indus Marine A-1 well in the northern Arabian Sea[20]. **n** Kaolinite/chlorite ratio from IODP Site U1447 in the western Andaman Sea[21] and K/Al ratio from ODP Site 718 in the Bengal fan[20]. **o** Total sediment flux into the Indus fan (10³ km³ Ma⁻¹)[19]. The vertical shaded areas in (**i–o**) represent the time intervals of Middle Miocene Climate Optimum (MMCO, -15 Ma) and Late Miocene (-5 Ma).

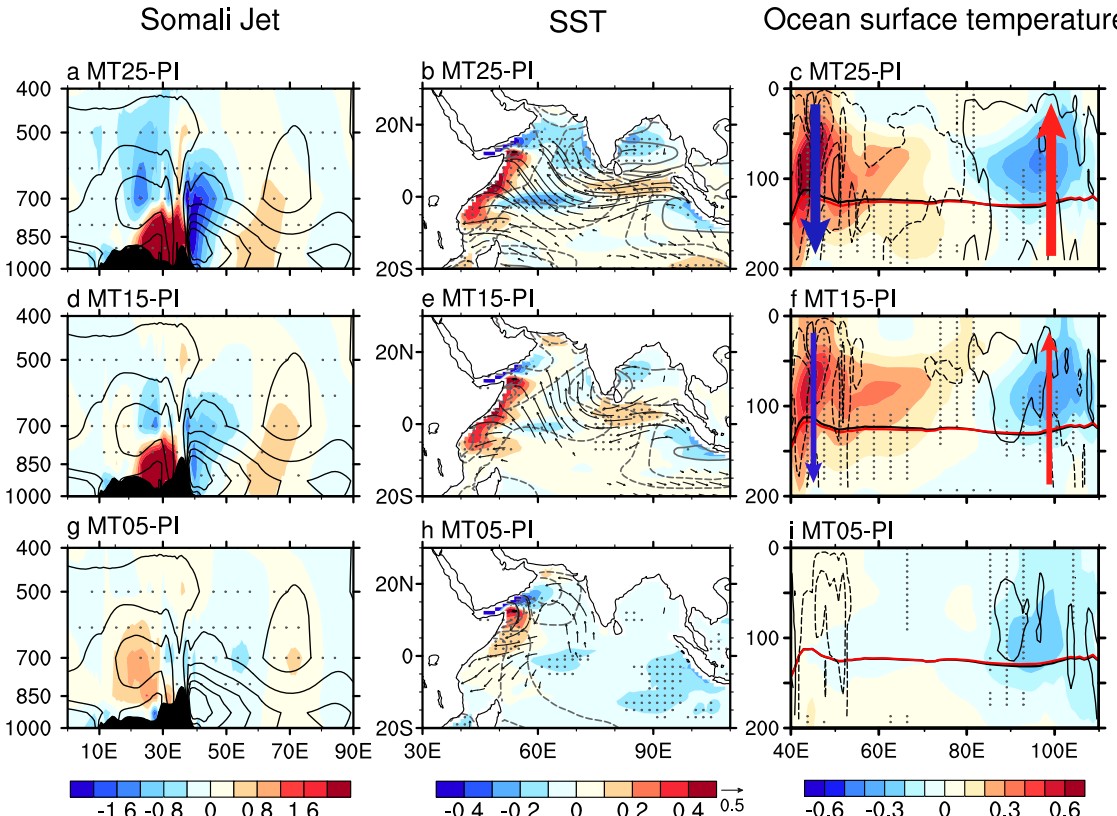

**Fig. 2 | Responses of ocean processes to African topography changes during boreal summer.** Changes in meridional wind at the equator (shading, m s⁻¹) in the (**a**) MT25, (**d**) MT15 and (**g**) MT05 simulations compared to pre-industrial simulations, overlaid with their corresponding climatological means in pre-industrial simulations (contours; solid lines represent positive values). Black shading in (**a**–**g**) indicates African topography in Early Miocene, Middle Miocene and Late Miocene, respectively. **b**–**h** Same as (**a**–**g**) but for changes in sea surface temperature (SST; shading, °C), sea level pressure [contours; hPa; solid (dashed) lines represent the positive (negative) values], and 1000-hPa wind (vectors, m s⁻¹). Gray stippling in (**a**–**i**) and vectors in (**b**–**h**) denote regions in which the changes are significant at the 95% confidence level according to Student's *t*-test. **c**–**i** Same as (**a**–**g**) but for the changes in meridional averaged (10° S-5° N) ocean subsurface temperature (shading, °C), ocean vertical velocity [contours; cm s⁻¹; solid (dashed) lines represent the upward (downward) motion], and thermocline represented by 23 °C isotherm [black (red) line indicates the thermocline in the pre-industrial (Miocene) simulation]. Vertical red and blue vector in (**c**) and (**f**) represent ocean upwelling and downwelling anomalies, respectively.

summer rainfall over tropical central Africa. During boreal summer, prevailing southwesterly monsoonal winds transport moisture from the tropical Atlantic Ocean and the Gulf of Guinea into central Africa, sustaining regional rainfall (Supplementary Fig. 7a). However, during the Early to Middle Miocene, large-scale tectonic uplift over the East African Highlands likely resulted in a higher elevation of the Congo basin (Fig. 1a, b, and Supplementary Fig. 2). This elevated topography acted as a barrier, reducing moisture transport from the Atlantic Ocean into central Africa, and resulting in a notable decrease in atmospheric water vapor (Supplementary Fig. 7b and f).

The resulting weakening of the West African summer monsoon causes significantly reduced rainfall over central Africa (Fig. 1d, e). This drying trend is further supported by enhanced descending air anomalies (Supplementary Fig. 7c and g) and a substantial reduction in latent heat release in the region (Supplementary Fig. 7d and h). Additionally, surface warming anomalies over central Africa (likely due to suppressed convection) and increased surface pressure indicate the presence of subsidence driven atmospheric stability (Supplementary Fig. 7e and i). These conditions likely triggered a Kelvin wave response, generating westerly wind anomalies over the tropical Indian Ocean[40], which gradually weakened toward the northern Indian Ocean, thereby reinforcing cyclonic circulation in the low-level troposphere (Fig. 1d, e).

To confirm that suppressed precipitation over tropical central Africa could enhance SASM rainfall via atmosphere dynamics, we conducted an experiment using an LBM with prescribed cooling over tropical central Africa (see Methods and Supplementary Fig. 8a). According to Gill's theory[41], localized atmospheric heating and cooling anomalies generate distinct wave responses in the tropics, where Rossby waves propagate westward and Kelvin waves propagate eastward. While Kelvin waves are strongest near the equator (-10° S–10° N), their influence can extend up to ~20° N, depending on the structure and strength of the forcing[40]. Our LBM experiment demonstrates that suppressed convection over North-Central Africa induces a cooling anomaly, triggering a Kelvin wave response with westerly wind anomalies over the tropical Indian Ocean. This result is consistent with a previous study[40]. Furthermore, maximum wind speeds are found north of the equator, decreasing with latitude in the northern Indian Ocean, this favors the establishment of a cyclonic circulation over the Arabian Sea, which enhances moisture transport into the SASM region, increasing local rainfall.

However, it is worth noting that the simulated cyclonic circulation in the LBM experiment is positioned slightly northward compared to the EC-Earth simulations. This discrepancy suggests the involvement of additional mechanisms, such as a positive precipitation-atmosphere feedback and the impact of a weakened Somali Jet. To investigate this further, we performed another set of LBM experiments with prescribed heating over the SASM region (see Methods). These experiments show that enhanced SASM rainfall can trigger a Rossby wave response over its northwestern region (Supplementary Fig. 8b), which

further enhances cyclonic circulation over the Arabian Sea, highlighting the role of positive precipitation-atmosphere feedback. Moreover, our results demonstrate that the Somali Jet weakens significantly when the East African Highlands are lowered during the Early and Middle Miocene. This reinforces northerly wind anomalies over the western Arabian Sea, shifting the cyclonic circulation further south compared to its position in the LBM experiment results. This displacement of the cyclonic anomaly is a direct result of the altered Somali Jet structure, further confirming the significant role of African topography in modulating the SASM system during the Miocene.

### Increased SASM rainfall due to indirect ocean-atmosphere feedbacks

The increased SASM rainfall during the Miocene was further amplified by the weakened Somali Jet through positive ocean-atmosphere feedback mechanisms. Figure 2 shows how the weakened Somali Jet induces a positive Indian Ocean Dipole (IOD)-like warming pattern during the Miocene. The pressure-longitude cross section at the equator shows that lower East African Highlands during the Miocene led to northerly wind anomalies along the East African coast, weakening the climatological southwesterlies summer monsoon circulation (Fig. 2a, d). This weakening, in turn, reduced ocean upwelling along the East African coast, leading to anomalously warming SSTs in the western Indian Ocean (Fig. 2b, e and Supplementary Fig. 9). This warming of the western Indian Ocean lowers sea level pressure (SLP), reinforcing easterly wind anomalies along the equator, which enhance ocean cooling in the eastern Indian Ocean (Fig. 2b, e). These processes are further supported by their seasonal variation (Supplementary Fig. 9). Specifically, warm SST anomalies develop in the western equatorial Indian Ocean by June, accompanied by significant easterly wind anomalies extending from the west to central Indian Ocean. In the following months, this warming intensifies, with easterly wind anomalies migrate further eastward, while the southeastern tropical Indian Ocean begins to cool. These processes are particularly pronounced in the Early and Middle Miocene when the East African Highlands were much lower compared to the Late Miocene (Supplementary Fig. 10).

According to ocean-atmosphere coupling theory[42], these anomalous easterly winds enhance downwelling in the western Indian ocean and upwelling along the Java and Sumatra coasts in eastern Indian Ocean[43–45] (Fig. 2c, f). Simultaneously, the thermocline deepens in the equatorial western Indian Ocean while becoming shallower in the eastern Indian Ocean, producing a dipole-like temperature pattern characterized by a "warm West and cold East" structure, indicating a positive IOD-like warming pattern[43–46] (Fig. 2c, f). It is important to note that during the Late Miocene, African topography was similar to present-day, resulting in minimal changes in the Somali Jet and Indian Ocean dynamics (Fig. 2g–i). This supports the conclusion that changes in Somali Jet plays a critical role in controlling Indian Ocean dynamics on geological timescales.

The positive IOD-like warming pattern induced by the weakened Somali Jet during the Early and Middle Miocene, further increases SASM rainfall through atmosphere dynamics. Specifically, the reduced gradient of zonal SLP over the Indian Ocean caused by the IOD-like warming pattern leads to a weakened Indian Ocean Walker circulation[45], featuring anomalously descending motion over the eastern Indian Ocean and ascending motion over the western Indian Ocean (Fig. 3a, c). This anomalous atmospheric circulation results in reduced latent heat release due to suppressed precipitation over the eastern Indian Ocean, which in turn triggers a Rossby wave response with an anticyclonic circulation pattern over the Bay of Bengal[40,41,47,48] (red bold vectors in Fig. 3b, d). Consequently, enhanced southerly winds transport more moisture away from the tropical ocean as a part of the large-scale anticyclonic flow over the Bay of Bengal, further contributing to increased rainfall (Supplementary Fig. 5).

To further investigate the impact of suppressed convection over the eastern Indian Ocean on SASM rainfall, we designed an LBM experiment with prescribed cooling over the eastern Indian Ocean (see Methods). Consistent with Gill's theory[41], cooling in the eastern equatorial Indian Ocean produces a pair of anticyclones in the off-equatorial region, resulting in the equatorial easterly wind anomalies (Supplementary Fig. 8c). This sustains and strengthens positive IOD-like warming anomalies. The LBM experiment supports our EC-Earth simulation results, indicating that the anomalous cyclonic circulation over the Bay of Bengal also contributes to increased SASM rainfall by reinforcing positive ocean-atmosphere feedbacks.

### The effects of the $p\mathrm{CO_2}$ forcing show no decoupling effect

The Miocene period was generally warmer than today[5], particularly during the Middle Miocene Climatic Optimum (~15 Ma), largely due to elevated atmospheric $CO_2$ concentrations[4] (Fig. 4a). However, multi-proxy estimates of atmospheric $CO_2$ during this period remain highly uncertain[49–68] (Fig. 4a), making it difficult to precisely assess the role of $p\mathrm{CO_2}$ in SASM changes. To determine whether the SASM response to African topography changes depends on variations in $CO_2$ concentration, we conducted sensitivity experiments using the EC-Earth3 model (see Methods).

Unlike the decoupling of Somali Jet and SASM rainfall observed in experiments driven by changes in African topography, our simulations with elevated $CO_2$ consistently produce both an enhanced Somali Jet and increased SASM rainfall (Fig. 4), demonstrating no decoupling effect. The increase in SASM rainfall due to increased $CO_2$ concentration (Fig. 4b–d) aligns with previous studies, highlighting the role of increased atmospheric moisture in response to global warming[69–72]. This follows the "wetter-gets-wetter" mechanism[71,73], whereby warmer temperatures intensify moisture availability in already humid regions. As expected, the rise in surface air temperature due to higher $CO_2$ level leads to increased atmospheric moisture content (Supplementary Fig. 11a), consistent with the Clausius-Clapeyron relation[71]. Thus, while changes in African topography affect SASM rainfall through dynamic circulation anomalies, $CO_2$ forcing primarily drives rainfall changes via thermodynamic effects (Supplementary Fig. 11).

Notably, uncertainties in $p\mathrm{CO_2}$ estimates do not significantly alter the SASM circulation response to African topography changes during the Miocene. Simulations that combine $CO_2$ forcing and African topography changes (see Methods) show clear cyclonic and anticyclonic atmospheric circulations over the Arabian Sea and the Bay of Bengal, respectively (Supplementary Fig. 12). This pattern is similar to the atmospheric circulation anomalies due to African topography changes (Fig. 1d, e), indicating that SASM circulation changes during the Miocene are mainly controlled by the African topography rather than $p\mathrm{CO_2}$ levels. Furthermore, the decoupling of Somali Jet and SASM rainfall—characterized by a weakened Somali Jet alongside increased SASM rainfall—is also evident in these combined simulations. This reinforces the conclusion that African topography played a dominant role in shaping the evolution of the SASM during the Miocene, whereas $CO_2$ forcing alone is insufficient to drive the observed decoupling.

## Discussion

We investigated the hydroclimate over the SASM region using the fully coupled EC-Earth3 model, constrained by reconstructed African topography at three key time slices across the Miocene. Our findings reveal that African topography changes played a crucial role in driving the decoupling of a weakened Somali Jet and increased SASM rainfall, while $CO_2$ changes primarily affected the thermodynamic response. The key processes are summarized in Fig. 5.

During the Early and Middle Miocene, significant African topography changes directly weakened and altered the structure of the cross-equatorial Somali Jet, while also suppressing convection over tropical central Africa. These changes triggered an anomalous cyclonic

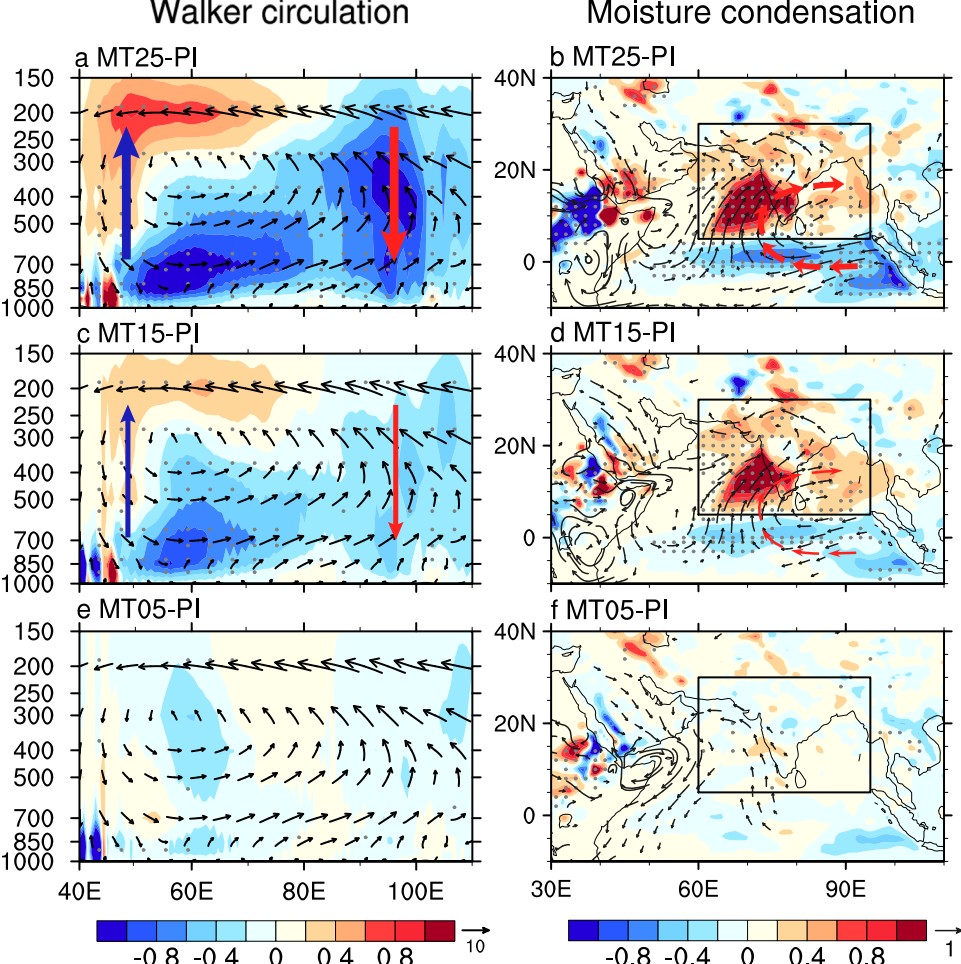

**Fig. 3 | Responses of atmospheric circulation to an Indian Ocean Dipole (IOD)-like warming pattern during boreal summer.** Changes in vertical velocity (shading, $-10^{-2}$ Pa s$^{-1}$; the positive values indicate upward velocity) in the (**a**) MT25, (**c**) MT15 and (**e**) MT05 simulations relative to pre-industrial simulations, overlaid with the climatological mean of meridional (m s$^{-1}$) and vertical wind (vectors) in pre-industrial simulations. **b–f** Same as (**a–e**) but for changes in moisture condensation ($Q_L$; shading; $10^{-2}$ m$^2$ s$^{-3}$; see Methods for details) at 500 hPa and horizontal wind at

low-level troposphere (m s$^{-1}$; averaged from 925 hPa to 700 hPa). Vertical red and blue vector in (**a** and **c**) represent anomalous upward and downward motion, respectively. Red bold vectors in (**b** and **d**) indicate the anomalous anticyclonic circulation. Solid boxes in (**b**, **d** and **f**) mark the South Asian summer monsoon (SASM) region. Only winds >0.2 m/s are shown. Gray stippling in (**a–f**) and vectors in (**b–f**) denote regions in which the changes are significant at the 95% confidence level according to Student's *t*-test.

circulation in the low-level troposphere over the Arabian Sea, enhancing moisture transport from the tropical Indian Ocean to the SASM region, thereby increasing local monsoonal rainfall (Fig. 5a). Additionally, the weakened Somali Jet contributes to the establishment of a positive IOD-like warming pattern, resulting in a weakened Indian Ocean Walker circulation (Fig. 5b). This weakened large-scale atmospheric circulation in the tropics further suppresses convection over the eastern Indian Ocean, triggering anticyclonic circulation in the low-level troposphere over the Bay of Bengal, enhancing the meridional moisture convergence in SASM region and increasing rainfall (Fig. 5b).

The primary contribution of our study is the identification of a previously overlooked tectonic mechanism that drives the decoupling of Somali Jet and SASM rainfall during the Miocene. Unlike previous studies that primarily attribute SASM evolution to the changes in land/ocean configuration[74,75], changes in polar ice sheet development[25,26], changes in ocean gateways[23,76], closure of the Tethys Seaway[13,74,77], and uplift of Himalaya-Tibetan and other regions[6,7,24], our results provide compelling evidence that African topography changes were a dominant driver of SASM evolution.

Our high-resolution EC-Earth simulations reconcile contradictory proxy evidence, resolving the discrepancy between wind-based proxies in the Arabian Sea and rain-related proxies over the Indian

continent and nearby Indian Ocean for SASM changes from the Middle to Late Miocene. This is the first study to provide a comprehensive dynamical explanation of this decoupling phenomenon using a state-of-the-art coupled climate model, highlighting the importance of accurately reconstructing the timing and magnitude of African topographic changes when interpreting past monsoon evolution. Moreover, as climate models project significant changes in precipitation over Africa due to global warming, similar latent heat-induced atmospheric circulation changes could influence future SASM dynamics. This raises the possibility that a decoupling of the Somali Jet and monsoonal rainfall could occur again in a warming world, with potential consequences for regional hydroclimate, monsoon predictability, and water resource management in South Asia.

## Methods
### EC-Earth3 model and simulations
The EC-Earth3 is a fully coupled Earth system model developed by a consortium of European research institutions[78], contributing to the Climate Model Intercomparison Project (CMIP) efforts[79–81]. It has been widely used to explore the climate dynamics of past, present and future[72,79,82]. The EC-Earth model incorporates various state-of-the-art components, including atmosphere, ocean, sea ice, land and

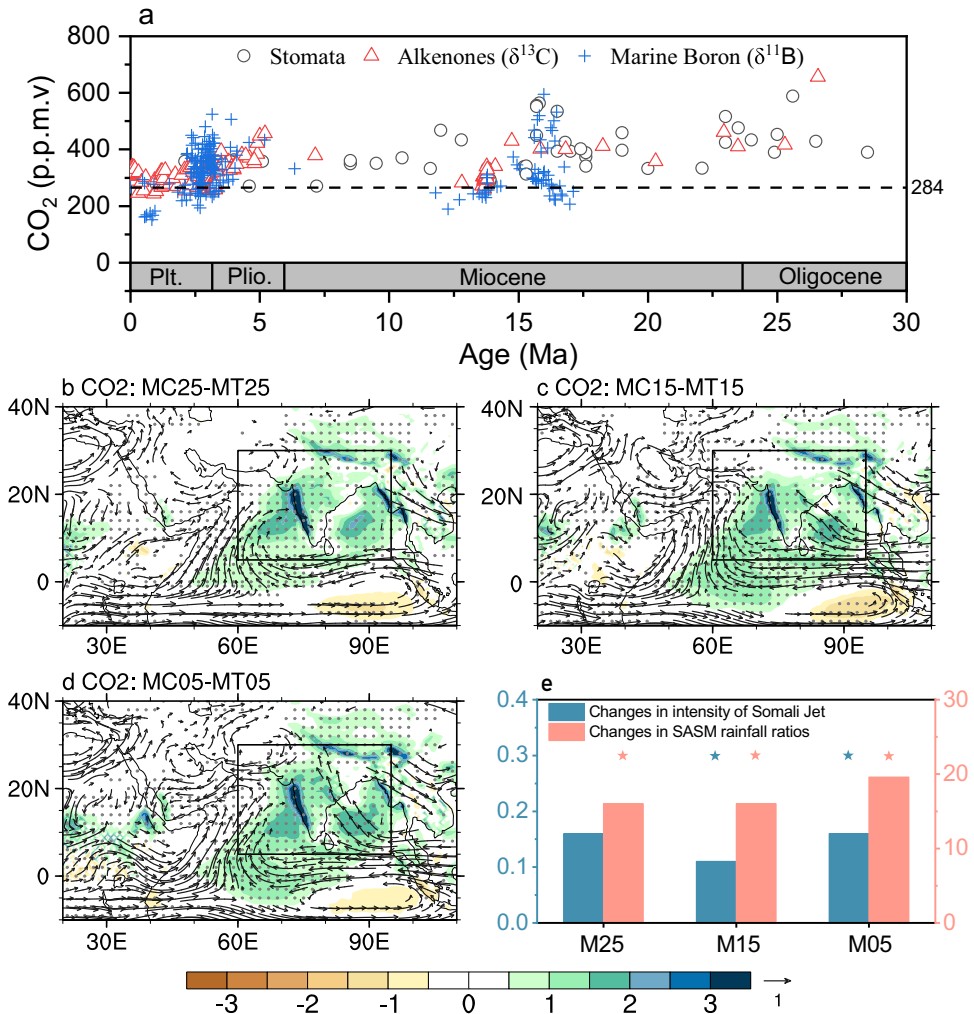

**Fig. 4 | Responses of atmospheric circulation to increased CO₂ concentration during boreal summer. a** Multi-proxy atmospheric $CO_2$ levels (p.p.m.v) compiled from previous literature, including stomata[49–59] (black circles), $\delta^{13}C$ of alkenones[60–63] (red triangles) and marine boron[60,62,64–68] (black crosses). Changes in rainfall (shading, mm day⁻¹) and 700 hPa wind (vectors, m s⁻¹) in the (**b**) MC25, (**c**) MC15 and (**d**) MC05 simulations relative to the MT25, MT15 and MT05 simulations, respectively. In (**b**–**d**) gray stippling and vectors denote regions in which the changes are significant at the 95% confidence level according to Student's *t*-test, and the solid boxes mark the South Asian summer monsoon (SASM) region. **e** Area-averaged changes in Somali Jet intensity (m s⁻¹; see Methods) and ratios of summer rainfall (%) over the SASM region due to increased CO₂ concentration in (**b**–**d**) with asterisks above the bars denoting changes that are significant at the 95% confidence level according to Student's *t*-test.

biosphere. The atmospheric component is the Integrated Forecast System (IFS, version cycle 36r4) model from the European Center for Medium-Range Weather Forecasts (ECMWF), including the land surface hydrology (H-TESSEL) model. The ocean component is the Nucleus for European Modelling of the Ocean (NEMO) version 3.6[83], which is coupled with a sea ice model (LIM3), and the biogeochemical model PISCES. EC-Earth3 also incorporates the dynamic vegetation component from the Lund-Postdam-Jena General Ecosystem Simulator (LPJ-GUESS). Detailed descriptions of the EC-Earth3 have been introduced previously[79].

In this study, we use model configurations EC-Earth3-veg (standard CMIP6 model name), with coupled dynamic vegetation model. The atmosphere component IFS has a resolution of T255 spherical grid in the horizontal (~0.7°, 85 km) and 91 levels in the vertical; the resolution of the ocean model is ORCA1L75, representing ~1° in the horizontal, and 75 layers in the vertical.

We conducted seven sensitivity experiments to investigate the roles of the African topography and $p$CO₂ in shaping the Miocene climate. Due to computational limitations, transient simulations covering the entire Miocene are not feasible, so we focus on three key time slices representing different stage of African topography change. In addition to a pre-industrial experiment (PI) with modern topography and a CO₂ concentration of 284 p.p.m.v., we performed three experiments based on the PI setup, incorporating the African topography changes during the Early Miocene (Oligocene-Miocene transition around 25 Ma ago; MT25), Middle Miocene [Middle Miocene Climate Optimum (MMCO) around 15 Ma ago, MT15], and Late Miocene (Miocene-Pliocene transition around 5 Ma ago, MT05). The African topography details during the Miocene are set according to the geological maps from ref. 28, which reconstructs the evolution of dynamic topography of Africa over the past 30 Ma.

Furthermore, we conducted three additional experiments to clarify the relative impact of high CO₂ on SASM rainfall compared to African topography changes. In these experiments, CO₂ concentrations are set to 500 p.p.m.v. with African topography in Early, Middle and Late Miocene, referred to as MC25, MC15 and MC05, respectively. Orbital forcing was not considered, and all simulations were run with pre-industrial orbital settings (longitude of perihelion = 100.33, obliquity = 23.549, eccentricity = 0.016764). Thus, climate anomalies due to African topography changes are represented

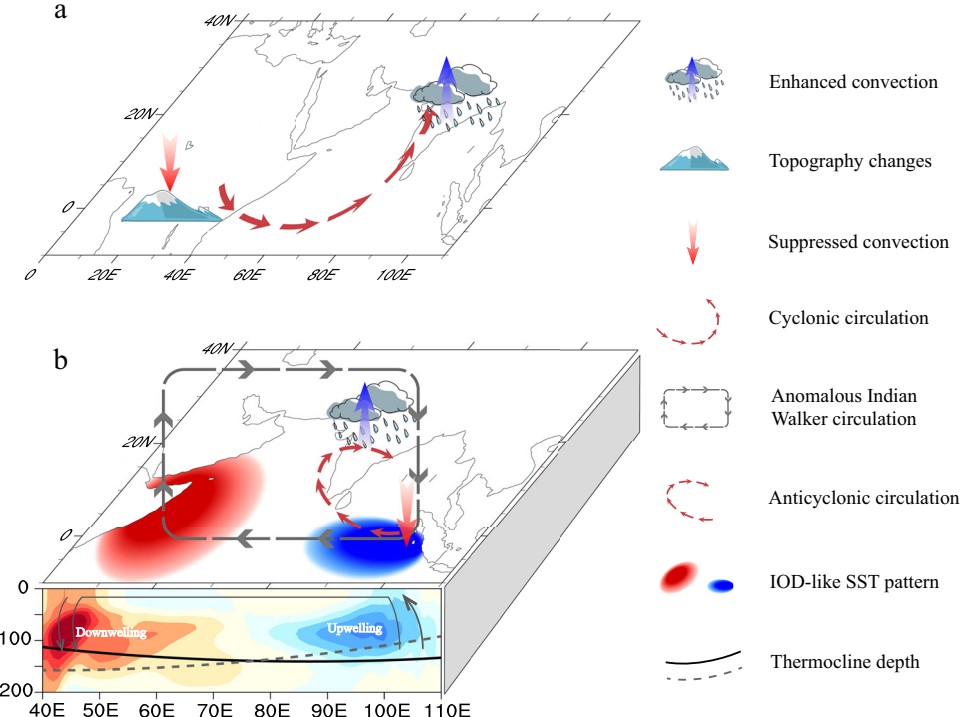

**Fig. 5 | Summary schematic.** Schematic diagram illustrating the mechanism for (**a**) the effect of Miocene African topography changes on increased South Asian summer monsoon (SASM) rainfall through direct atmospheric processes, and (**b**) the effect of weakened Somali Jet on increased SASM rainfall through indirect oceanic processes.

by the difference between the MT25/MT15/MT05 experiments and the PI experiment. The response of SASM changes to higher $CO_2$ concentrations is denoted by the comparisons between the MC25 and MT25, MC15 and MT15, or MC05 and MT05. The combined effects of African topography changes and elevated $CO_2$ levels are indicated by the differences between MC25/MC15/MC05 and PI experiment.

Supplementary Table 1 provides an overview of experiments, detailing the corresponding topography and $p$CO₂ values. The pre-industrial simulation was run for 1000 model years to ensure equilibrium, while the other experiments were run for 200 model years. Since all simulations reach quasi-equilibrium during the last 100 years, with the trend in mean global surface temperature being <0.005 K, the last 100 years of each simulation are used for analysis.

### Definition of South Asian summer monsoon metrics

The South Asian summer monsoon (SASM) region is defined as the area spanning 5° N-30° N, 60° E-95° E[84,85]. This study focuses on the boreal summer months from June to August, unless otherwise specified. Including May and/or September in the summer season does not significantly alter the results. The intensity of SASM rainfall is measured by averaging summer rainfall over SASM domain. The variation in the Somali Jet significantly impacts the SASM rainfall and produces notable oceanic features in the western Arabian Sea. During summer, these winds force upwelling of colder waters along the coasts of Oman and Somalia, increasing nutrient levels and sustaining distinct floral and faunal groups[9]. Due to the relationship between the Somali Jet, SASM rainfall, and ocean circulation, the Somali Jet is often used to measure the strength of SASM circulation, especially in the paleoclimate research[8–13]. In this study, the intensity of Somali Jet is defined as the wind speed at 850 hPa averaged over the region of 10° S–10° N, 40°–50° E during summer[32,86]. In addition, the intensity of western Arabian Sea upwelling is defined as the average vertical velocity of the surface ocean (0–100 m) over the western Arabian Sea (15°–21° N, 54°–62° E)[36].

### Proxy-model comparison

The proxy records reflecting the Somali Jet and humid conditions over South Asia during the Miocene are collected based on previous studies[9–11,15–17,19–21,87]. Total organic carbon (TOC) values and Planktic foraminifer *Globigerina bulloides* (*G. bulloides*)[10,11] are widely used to estimate the SASM wind intensity, as they predominantly reflect wind-driven oceanic upwelling and primary production in the western Arabian Sea. Both the TOC and *G. bulloides* values from the western Arabian Sea are lower during the Middle Miocene compared to the Late Miocene[9–11] (Fig. 1i, j), suggesting weakened SASM winds-induced oceanic upwelling in the western Arabian Sea during the Middle Miocene. Our African topography experiments can capture this feature in the wind-based proxies, showing significant weakened upwelling in the western Arabian Sea, particular in the MT25 and MT15 simulations compared to the pre-industrial simulation (Fig. 1g and Supplementary Fig. 4). We found that this weakened upwelling induced by African topography changes is due to the changes in both the strength and structure of Somali Jet (Supplementary Fig. 3 and 4), indicating the complex changes in Arabian Sea upwelling[34–36].

In contrast, these wind-based proxies are challenged by rainfall-related proxies. For example, $\delta^{13}C$ values in paleosols provide a direct measure of the relative contribution of grasses (C4 plants) and woodland or/and forest (C3 plants) to former vegetation, with C4 grasses favoring drier environment and C3 plants favoring moister environment. $\delta^{13}C$ in paleosols from Siwalik Group sediments[15,87] is relatively depleted before 7 Ma and more enriched after 7 Ma (Fig. 1k), suggesting that C3 trees dominated in the Middle Miocene while C4 grasses became more prevalent in the Late Miocene[16,17]. The $\delta^{18}O$ of pedogenic carbonate from the Indian Siwalik sequences also indicates decreased rainfall in the Late Miocene[15–17], coinciding with the rise of C4 plants (Fig. 1l). This indicates a hydroclimatic change that might have played a crucial role in the transition from C3 to C4 vegetation. Our simulations also capture this forest-to-grassland transition, showing less tree coverage and more grass coverage across the Indian continent and Himalayan foreland during the Late Miocene than those

in the Early and Middle Miocene (Supplementary Fig. 13). Thus, our results suggest that the dramatic aridification of the Indian continent from the Middle and Late Miocene, driven by changes in African topography, may have been a key mechanism behind the expansion of C4 plants. This hypothesis is supported by other rainfall-related proxies. Reconstructions of chemical weathering and erosion budgets indicate that faster erosion, associated with more humid and warm climates in the Early to Middle Miocene, transitioned to less erosive and drier climates in the Late Miocene[19–21] (Fig. 1m–o).

All these qualitative climate proxies for the Miocene, derived from various sources and locations, generally indicate weakened SASM-induced Arabian Sea upwelling and more humid conditions over South Asia during the Middle Miocene compared to the Late Miocene. This significant feature is well captured by our African topography experiments (Fig. 1d–h).

However, there are some uncertainties in proxy records that should be noted. For instance, while $\delta^{18}O$ of soil carbonates is often interpreted as a precipitation proxy, it is also influenced by other factors, such as moisture source variability, wind system shifts, and temperature variations[88]. As for the wind proxies in western Arabian Sea, such as TOC and *G. bulloides* records, considered as indicators of monsoon winds, can also be influenced by local oceanographic processes and thermocline variability, adding complexity to their interpretation[36]. Additionally, most proxies come from specific regions (e.g., Arabian Sea, Siwaliks Hills), which may not fully capture spatial monsoon variability. Despite these uncertainties, the consistent multi-proxy changes support the conclusion that these proxies reliably reflect SASM evolution during the Miocene.

### Wind-driven upwelling

The monsoon wind can induce ocean upwelling through "Ekman pumping" caused by wind-stress curl[36,89]. In this study, we calculate the Ekman pumping velocity using monthly wind stress data as follow:

$$W_{Ekman} = \frac{1}{\rho_w} \nabla \times \left(\frac{\tau}{f}\right) \approx \frac{1}{\rho_w f} \nabla \times \tau \tag{1}$$

where $\rho_w$ is seawater density (1024 kg m$^{-3}$), $f$ is the Coriolis parameter, and $\nabla \times \tau$ is the wind-stress curl.

### Latent heat of condensation

In this study, the latent heat of condensation ($Q_L$)[90,91] is calculated to reveal the anomalous atmospheric cooling due to significant rainfall decrease. The formula of $Q_L$ is as follows:

$$Q_L = \begin{cases} -L\omega\frac{\partial q_s}{\partial p}, & \omega < 0 \\ 0, & \omega \geq 0 \end{cases} \tag{2}$$

where $L$, $\omega$, $q_s$, and $p$ indicates the latent heat of condensation, vertical velocity, saturation specific humidity, and pressure, respectively.

### Decomposed atmospheric moisture budget

The atmospheric moisture budget is a widely used analysis to reveal the dominant process controlling the rainfall changes[69,73,92], which can be written as:

$$P' = \langle \bar{\omega} \cdot \partial_p q' \rangle + \langle \omega' \cdot \partial_p \bar{q} \rangle - \langle V \cdot \nabla q \rangle' + E' + Residual \tag{3}$$

where $P$, $\omega$, $q$, $p$, $V$, and $E$ denotes the precipitation, vertical velocity, specific humidity, pressure, and evaporation, respectively. The over-bars represent the climatological mean in pre-industrial conditions, and the primes indicate deviations from this mean in sensitive experiments. The symbol $\langle \cdot \rangle$ indicates the vertical integration, and the *Residual* term includes the transient eddy and surface boundary effects. For tropical rainfall anomalies, previous studies indicate that

the contributions of advection ($\langle V \cdot \nabla q \rangle'$), evaporation ($E'$) and *Residual* terms are relatively negligible[70,93]. Hence, Eq. (3) can be simplified as follows[70,72,85]:

$$P' \sim \underbrace{\bar{\omega}_{500} \cdot \text{Pr } w'}_{\delta Th} + \underbrace{\omega'_{500} \cdot \overline{Prw}}_{\delta Dy} \tag{4}$$

where the $\omega_{500}$ represent the vertical velocity at 500 hPa, and *Prw* is the vertical integrated atmospheric moisture through the column. Based on the simplified moisture budget equation, changes in SASM rainfall ($P'$) can be decomposed into the effects of atmospheric moisture changes (thermodynamic term; δTh) and the atmospheric circulation change (dynamic term; δDy). Therefore, in this study, the sum of the δTh and δDy can be roughly considered as the diagnosed changes in summer rainfall over the SASM region.

### LBM experiments

Compared to the complex Earth System models, the linear baroclinic model (LBM)[94] includes only linear processes, providing a more accurate investigation of the atmospheric response to diabatic heating or cooling[95,96]. To verify the impact of significantly suppressed convection over tropical central Africa and the equatorial eastern Indian Ocean on SASM rainfall, we performed four LBM experiments based on climatological summer mean of the National Centers for Environmental Prediction (NCEP) reanalysis from 1980-2010, using a time integration method for 40 days. Perturbations from the basic state are regarded as the linear response to the forcing. We used an LBM version with 20 vertical sigma levels and a horizontal resolution of T21, taking the last 10 days as the steady-state solution.

In experiment 1 (Exp_1), we prescribe the idealized cooling in tropical central Africa (5° S-15° N; 20°-40° E) with a vertical profile following a gamma function, peaking at a cooling rate of 6 to 7 K day$^{-1}$ at about 450 hPa, based on climatology from June to August (ref. 95). Experiments 2 (Exp_2) and 3 (Exp_3) follow the same methodology as in Exp_1, but applied heating over SASM region (5°-30° N; 60°-95° E) and cooling over tropical eastern Indian Ocean (5° S-5° N; 90°-110° E), respectively. Exp_4 combined diabatic cooling over tropical central Africa and eastern Indian ocean, and heating over SASM region. Indeed, Exp_4 (Supplementary Fig. 7d) reproduced the cyclonic and anticyclonic circulation anomalies over the northern Indian Ocean observed in fully coupled EC-Earth3 simulations (Fig. 1d, e), suggesting that suppressed rainfall in the tropics plays a crucial role in driving large-scale atmospheric circulation anomalies.

## Data availability

The GPCPv2.3 data from the Global Precipitation Climatology Project is available at https://psl.noaa.gov/data/gridded/data.gpcp.html. NCEP2 data is downloaded from https://psl.noaa.gov/data/gridded/data.ncep.reanalysis2.html. Source data underlying the main figures are provided with this paper on Zenodo at https://doi.org/10.5281/zenodo.15663883.

## Code availability

All Figures in this article are produced by the NCAR Command Language (Version 6.4.0)[97], and the source codes for the main results are available on Zenodo at https://doi.org/10.5281/zenodo.15663846.

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

## Acknowledgements
This work was supported by the National Key Research and Development Program of China (No. 2024YFF0807903), Swedish Research Council (Vetenskapsrådet, Grant 2022-03129), and National Natural Science Foundation of China (No. 42305053, 42275047). The EC-Earth3 simulations and data analysis were performed using ECMWF's computing and archive facilities and the National Academic Infrastructure for Supercomputing in Sweden (NAISS), partially funded by the Swedish Research Council through grant agreement no. 2022-06725.

## Author contributions
Q.Z. and Z.H. conceived and designed the study. Z.H. performed the data analyses and wrote the draft of the paper. W.N. and Z.W. carried out the EC-Earth experiments. Z.H., W.N., Z.W., X.L., Z.Y. and Q.Z. discussed, reviewed, and edited the manuscript.

## Funding

## Competing interests
The authors declare no competing interests.
