## [Transparent Peer Review file · Nature Communications]

Miocene African topography induces decoupling of Somali Jet and South Asian summer monsoon rainfall

Corresponding Author: Professor Qiong Zhang

Version 0:

Reviewer comments:

Reviewer #1

(Remarks to the Author)

The manuscript by Han et al. documents the impacts of African topography on the South Asian summer monsoon winds and rainfall during the Miocene. The manuscript is generally well written and claims are substantiated by physical mechanisms. Overall, the mechanism suggested seems plausible. However, there are some significant sources of uncertainty (L338-346). These sources make me wonder whether present-day simulations by changing the orography would give similar results. I am by no means, an expert of the Miocene period. Therefore, my comments are restricted to the monsoon dynamics reported here. I would expect other reviewers to handle this aspect.

My primary concerns are listed below:

Major Comments:

- Fig. 1 d-f:

Why 700 hPa? 850 hPa will be more appropriate here, also, please draw only significant vectors.

The major changes in rainfall are seen only over the oceanic parts in the windward side of the Western Ghats, and the changes over India are very small. The changes in the Somali jet's moisture-carrying capacity also influence the circulation over the Bay of Bengal and the associated land region due to changes in synoptic-scale disturbances and intra-seasonal oscillations. Such changes are not seen here; why?

- Fig. 2 – IOD type bias – Cold SST anomalies over IOD east pole must result in suppressed rainfall over the region, the same is not clearly seen in Fig. 1 (d-f).

- Impact on IOD type bias – Is lowering SLP in western Indian Ocean sufficient to justify the enhanced easterlies along the equator? The magnitude of cooling in the IOD east pole region seems to be almost of similar magnitude in all three simulations. Positive IOD-type conditions will possibly enhance the Indian monsoon rainfall in all the three simulations.

- The impact of IOD dynamics on the model can imply a possible association with ENSO as well. The ENSO-IOD coupling is documented in the literature. Also show a global map of SSTs and indicate whether there is a change in SSTs globally.

- Conventionally, the Indian Monsoon is impacted by the IOD by modulation of the local Hadley circulation. The discussion here seems to indicate that atmospheric variability is the stronger driver. Are there other observational studies substantiating this? It is most probably coupled variability. How well does the PI simulation capture the IOD-SASM relation?

- What is the major objective of this study – documenting the impact of orography on SASM or reconciling the proxies? Since there are major changes in other forcing, as mentioned in L338-346, which may significantly add to the uncertainty, is it not better suited just to do present-day simulations with changes to orography?

- Please confirm if only significant differences are shown in the figures.

Minor comments

L73 – other boundary forcings

L147 – C4 "plants"

(Remarks on code availability)

Code is not provided.

(Remarks to the Author)

The Somali Jet, a near-surface air current along the coast of the Horn of Africa, plays a central role in the South Asian summer monsoon (SASM) dynamics. It transports large amounts of moist air from the Indian Ocean to India, where it causes heavy rainfall. A more intense Somali Jet is usually assumed to lead to more precipitation in the SASM domain. For the Miocene, however, reconstructions have shown that these two components appear to be decoupled, as wind proxies show a strengthening of the Somali jet from Middle Miocene to late Miocene, but precipitation in the SASM domain decreased.

Han et al. employ the EC-Earth3 model to assess the impact of changes in the African topography during the Miocene on this decoupling and the atmospheric and ocean dynamics and the South Asian monsoon precipitation, in general. To this end, they carry out simulations in which they keep all boundary conditions on the pre-industrial settings, but prescribe the African topography in line with reconstructions.

According to the model, the topography is important in two respects: First, due to the lower East African Highlands during Early Miocene, the Somali Jet is less canalized and therefore much weaker. This in turn has the effect of inducing a positive Indian Ocean Dipole-like SST pattern, which leads to more precipitation in the SASM domain and a positive feedback with the circulation. In the course of the Miocene, the East African Highlands are lifted up, which more and more intensifies the jet, explaining the signal seen in the wind-proxies.

Second, A higher Congo Basin leads to less ascent and less latent heat release into the atmosphere. This promotes the formation of a cyclonic circulation anomaly above the Arabian sea, redirecting moisture to the Indian continent. The effect of changes in latent heat release on the atmospheric circulation in distinct regions is further investigated by employing a linear baroclinic model.

In addition, Han et al. perform simulations with increased (based on reconstructions) CO₂ levels and with both forcings (CO₂ and topography changes).

Decomposing the precipitation response into the effect of atmospheric moisture change (thermodynamic) and atmospheric circulation changes (dynamic response) shows that topography changes purely leads to more precipitation due to changes in the circulation, whereas the CO₂ effect has a strong thermodynamic contribution to the precipitation change due to its impact on the atmospheric temperature.

This study highlights the effect of the African topography in the SASM evolution and underlines the complexity of the South Asian monsoon dynamics. The figures in the main part are well-chosen and helpful, and are also mostly well presented and easy to understand. The text is clearly and comprehensibly formulated and the individual parts seem well thought out. Methods are carefully applied. The main criticism I have is that I missed the "overarching framework". The individual sub-chapters don't always seem to build on each other and you get the impression that Han et al lose focus on their core topic and their aim of the study in the course of the manuscript. This means that the paper is not easily comprehensible to a wider readership. However, I believe that this can be solved and therefore suggest a publication in Nature Communication after major revisions.

In detail:

a) For me, not knowing much about the Miocene so far and therefore not being familiar with the literature, it is not clear what is really new about the study and what is the highlight of the study. According to the title, the main topic is the influence of African topography on the decoupling of wind and precipitation changes during the Miocene. In my opinion, there is no clear final answer to this research question in the study. For instance, in the abstract as main (and only) reason for the circulation and precipitation changes, the lower East African Highlands are mentioned. In contrast, large parts of the result part deal with the effect of the higher Congo basin on the circulation. Later in the text, the CO₂ experiments are analyzed, but the results neither enter the abstract nor the conclusion. In the conclusion, the East African highland and Congo basin topography changes are mentioned as explanation for the weakening of the Somali Jet and the feedbacks with the precipitation, which is not directly clear and confusing, because the linear baroclinic model (used for the explanation of the effect of topography changes in the Congo basin) show no weakening of the Somali Jet in any simulation. This conclusion is therefore imprecise. Not giving a clear answer to the main research question, makes the individual parts of the analysis seem more like, we did this and then that, rather than being able to recognize a clear common line in it. This is a pity, because the individual parts are important and interesting. I would like the reader to be guided more through the work, and the rationale for conducting the individual analyses to be explained more and the work to be structured more (and more clearly). Particularly the Abstract and the Conclusion should be revised.

b) In addition, the author explain the reduction of the Somali Jet via the reduction of the channel effect by the lower East African Highlands. Just looking at the circulation changes in Fig.1, showing a cyclonic circulation anomaly during the Early and Mid Miocene compared to pre-industrial, the pattern does not really look like a reduction in the channel effect. It looks like a thermally induced dynamic response to the decreased SST in the north-western Arabian sea. The CO₂-forced simulation shows a stronger Somali Jet compared to the topography forced simulation. Does this imply that in the CO₂ simulations the wind and precipitation changes are not decoupled? And that the decoupling is only induced by the topography changes, overcompensating the CO₂ effect in the "real world"? Please clarify.

c) With respect to this, to really understand how the changes in African topography affect the circulation and precipitation pattern, sensitivity studies are necessary in which either the changes in the Congo basin or the changes in the East African Highlands are prescribed, i.e. simulations in which e.g. the Congo basin elevation is increased, but the East African Highlands are prescribed on pre-industrial settings.

d) What do we know about the climate in the Miocene? About the drivers? Other (but not applied) changes in the boundary

conditions are only mentioned in the conclusion and discussion. For example, the Western Ghats were much lower, which probably also had an impact on the circulation and the precipitation in India. The Tibetan Plateau was probably only half as high. Today it acts as a strong barrier and the vertical ascending on the southern side is very strong, leading to large amounts of precipitation. In addition, it plays a very important role for the atmospheric circulation, also via the influence on the subtropical jet. So I suspect that there might be more reasons why the wind-precipitation-decoupling may have occurred and the climate was like as seen in the reconstructions. Please discuss this in more detail. It would be nice if this was already mentioned in the introduction and it was made clear that (presumably) the topographical changes and the higher CO₂ levels were the most prominent drivers and therefore only these forcings were used and it is now being investigated how large the proportion of orography is in comparison with the CO₂ forcing proportion. Maybe it would also be an idea to present the orography and CO₂ simulations together, to work out that the topography changes are more important. In the current version of the paper, the authors prescribe topography changes and the main result is that topography changes are important. But this is already a predetermined result, when topography is the only forcing of the model that have been prescribed.

e) A large part of the result chapter compares the model with data. For me, this is done very "uncritically". Although the reconstructions of wind and precipitation agree quite well with the model, the proxies experienced a completely different world than that in the idealised model simulation (see above). I also do not understand, why the reconstructions are at least not compared with the simulation including both forcing (CO₂ and topography changes).

And I also miss a critical (at least short) discussion on the uncertainties in the proxies. For instance, d₁₈O is often used as a proxy for precipitation, but it is also very dependent on other things (moisture source, wind system, temperature). I understand why Han et al. compare the simulations with reconstructions, but if it is to be part of the results, then a more critical discussion would be appropriate. Perhaps it would otherwise be an idea to include the evaluation of the model simulation in the methods (e.g. in the model and simulation description).

f) Please revise the supplementary figures. The captions and the heading of subfigures are not always clear and correct. I listed a few issues in my specific comments below, but please check all figures carefully before re-submission.

Specific comments

Abstract

- L33-L38: Please be more precise and refer to all results (see comment (a)). For instance, "blocking effect" is misleading here, as it is rather the opposite, a reduction of the blocking effect.

Introduction

L66: "weakening of the SASM": As in this study, changes in wind and precipitation are discussed, this term is not clear. It could be both, a weakening in the wind and a weakening in the precipitation.

Results:

L105: it is: decoupling of monsoon rainfall...

L166-L169: This sentence should be moved to the Introduction.

L177-178: "in western Arabian Sea (Fig.1) is written in italics

L183: "These circulations", not "There circulations"

L211-215: Since it is a coupled system, it is difficult to say what triggers the descending air anomaly. The vertical movement in models is often a result of the convection calculation. So it could well be that the weaker West African monsoon leads to less precipitation, e.g. because less moisture is transported to the region and the lower precipitation is causing/accompanied by weaker upscents and less latent heat release. What about the temperature and pressure at the surface? Maybe this would be another (better) line of argumentation.

L216: I'm not that familiar with Kelvin waves, but I think Kelvin waves are usually confined to the equatorial region (~10°S to ~10°N). I am also not sure whether a reduction in the latent heat release causes a Kelvin wave response, or whether, for example, the amplitude of the Kelvin wave simply changes. Please check this with the literature.

L243: ...seen in the LBM experiment results. The "seen in" is missing.

L282: Please add a reference to Gill's theory.

L319: ... using THE fully coupled EC-Earth3 model.

L328: This conclusion is imprecise, because the LMB model does not show a weakening of the Somali Jet. Please rephrase!

L344-346: I find it unfavorable to use this sentence as the final sentence. It implies that the evolution of the SASM was the main motivation for the study. This is not clear in the context of the study and is also not clear from the title. Instead, I would like to see a final answer to the reasons for the decoupling and maybe a statement what it means, e.g. for present or future climate. It is not clear from the study which effect predominates, but climate models project strong changes in the

precipitation pattern over Africa. Is it possible that we will experience a decoupling of the Somali Jet and SASM precipitation again in the future, due to changes in the latent heat release in Africa?

Methods:

L357: which IS the Nucleus for ...

L454: "cooling rate of 6 to 7 K/day"... Does it mean that it gets cooler every day? I do not understand this.

Figures:

general: I guess units are always written in rectangular bracket, e.g. [m/s], rather than (m/s).

Fig.1: Maybe flip the time-axis in the reconstruction plot so that it is in line with the other plots in this figure. Maybe you could additionally mark or explain in the caption which proxies in i) to o) indicate wind and which indicate precipitation.

Fig.1: Have the precipitation anomalies been tested for significance?

Fig.1: Why do you use 700hPa level and in other plots 1000hPa or 950hPa or 850hPa? Please check if you use this level for each plot referring to the Somali Jet to facilitate the comparison.

Fig.4: The unit for e) is missing... Is e) also showing the difference between MC and MT simulations? It would also be helpful to plot the area mean changes in the Somali Jet, similar to Fig.1

Fig.5a: I'm wondering, if the cyclonic circulation arrows have the wrong position. The winds are directed to the West Ghats and do not flow in western direction over India (cf. Fig.1). Similarly, the arrow for suppressed convection is located over the African Highlands but not over the Congo basin...

Supplement:

General: Please check the spelling in all figure captions.

Fig.1: Here, 850hPa winds are shown, why not consistent to the other plots (700hPa). Why are wind vectors shown in the simulation but not in the observation? They experience the same orography...

Fig 2: Please add a note that only African changes in topography are shown here.

Fig.4: Which levels were used to integrate the moisture flux?

Fig.6: Is K/day the correct unit for Q?

Fig.8: Which levels have been used for the integration of the water vapor flux? Do the plots b-c really show anomalies to MT? This would mean that the CO₂ has an even stronger dynamic response than seen for the topography change.

Fig.9: This plot also show the 850hPa while other plots use 700 hPa. It would be easier to compare the figures and the effect of the different forcings if the levels were standardized. Also for this plot it would be nice to add the area-mean change in the Somali-Jet intensity (cf. Bar plot in Fig.1)

Fig.9: The headings of the sub-figures are not correct.

(Remarks on code availability)

The code is not available, only upon request.

Reviewer #3

(Remarks to the Author)

Please find attached my comments.

(Remarks on code availability)

The authors state that the codes will be provided upon request.

Version 1:

Reviewer comments:

Reviewer #1

(Remarks to the Author)

Thanks for addressing my comments, I am satisfied with the responses.

Fig. R1 – I am a bit confused looking at the response at 850 and 700 hPa. The changes to circulation should be somewhat coherent at these levels. At 850 hPa, (Fig. R1 d-f), the changes to circulation seem to be emanating from equatorial Indian Ocean to Western Ghats. This circulation feature changes abruptly at 700 hPa. The changes in rainfall are concentrated over Western Ghats and Northwest India (which receives less rainfall anyway). No changes are seen along the monsoon trough.

(Remarks on code availability)

Reviewer #2

(Remarks to the Author)

The authors responded very carefully and very extensively to the reviewers' comments, implemented most of the comments and otherwise convincingly explained why they did not take the comments into account.

As a result, the manuscript has improved substantially. The structure and general statements are much clearer. Additional analysis and verification of the significance of the results reinforce the statements. I have rarely seen such a carefully prepared reply. My compliments to the authors.

I recommend the publication after a few minor, mostly technical revisions:

a) The section starting L152 (Weakened Somali Jet... is now somewhat redundant, as it doesn't bring much new compared to the section before. In the previous section, the weakening of the jet and the effects on the upwelling are already presented. So you could simply merge this section with the previous one.

b) I like that some sections have the main message formulated as a heading. Perhaps this style could be applied to all headings.

c)

L229 "consistent with a previous study" -> "a" is missing

L235 "such as a positive precipitation..." -> "a" is missing

L254 " a lower East African Highlands" -> delete the "a"

L290 it is "enhanced southerly winds"

L345 "the ... Somali Jet ..." -> add "the"

L349 it is "increasing local ..."

L361 it is "Himalaya-Tibetan"

Fig1: why is there no bar for M05?

Suppl. Fig 13 seems to be a bit blurry

(Remarks on code availability)

The code is now available, but I have not checked it, because I'm not familiar with ncl.

A README file is not included

Reviewer #3

(Remarks to the Author)

I have reviewed the authors' responses and the revised manuscript. Overall, the authors have addressed most of my concerns satisfactorily. However, one important issue remains unresolved—the matter of the decoupling between monsoon winds and South Asian Summer Monsoon (SASM) rainfall.

My original concern was that the authors seem to suggest a decoupling between monsoon winds and SASM rainfall during the Miocene, while their moisture budget analysis clearly indicates a strong relationship between atmospheric dynamics and SASM precipitation. Based on their response, it appears that the authors are actually referring to a decoupling between the Somali jet and SASM rainfall, rather than between large-scale monsoon winds and SASM rainfall.

In fact, their results show that large-scale monsoon winds—modulated by Miocene topography—play a key role in driving changes in SASM rainfall, independent of changes in the Somali jet. This implies that the modern-day coupling between the Somali jet and SASM rainfall does not necessarily hold on geological timescales.

The current phrasing—"decoupling between monsoon winds and SASM rainfall"—used in both the title and the main text is therefore misleading and technically inaccurate. I strongly recommend that the authors revise this wording to "decoupling between the Somali jet and SASM rainfall".

With this correction, I am happy to recommend the manuscript for publication.

(Remarks on code availability)

I have verified that the code is available in the repository provided by the authors. However, I have not attempted to run the code to confirm its functionality. Hence, I cannot say that I have reviewed the codes.

Reply to Reviewer #1's comments

Remarks to the Author:

The manuscript by Han et al. documents the impacts of African topography on the South Asian summer monsoon winds and rainfall during the Miocene. The manuscript is generally well written and claims are substantiated by physical mechanisms. Overall, the mechanism suggested seems plausible. However, there are some significant sources of uncertainty (L338-346). These sources make me wonder whether present-day simulations by changing the orography would give similar results. I am by no means, an expert of the Miocene period. Therefore, my comments are restricted to the monsoon dynamics reported here. I would expect other reviewers to handle this aspect.

My primary concerns are listed below:

Our response: We thank the reviewer for the kind consideration and constructive comments on our manuscript. The original comments are quoted in blue. We have carefully revised the manuscript and provided the point-by-point response below. The changes in the revised manuscript have been highlighted in red. We hope these changes will strengthen our manuscript.

General Comments:

#1. Fig. 1 d-f:

Why 700 hPa? 850 hPa will be more appropriate here, also, please draw only significant vectors.

The major changes in rainfall are seen only over the oceanic parts in the windward side of the Western Ghats, and the changes over India are very small. The changes in the Somali jet's moisture-carrying capacity also influence the circulation over the Bay of Bengal and the associated land region due to changes in synoptic-scale disturbances and intra-seasonal oscillations. Such changes are not seen here; why?

Our response:

(1) We appreciate the reviewer's insights regarding the choice of 700 hPa for wind vectors. After careful reconsideration, we have decided to retain the 700-hPa wind in Figure 1 in the manuscript for the following reasons:

- The primary weakening of the meridional cross-equatorial flow due to the lowering of the East African Highlands is clearly captured at both the 700-hPa (Figure R1a-c) and 850-hPa wind patterns (Figure R1d-f). However, the thermally induced cyclonic circulation anomalies over the Arabian Sea, which significantly contribute to increased SASM rainfall, are more prominent at 700 hPa.
- This anomalous cyclonic circulation is directly linked to suppressed convection over tropical central Africa (Supplementary Figure 5 and 7 in the Supplementary materials). This is further intensified by the Rossby wave induced by increased SASM rainfall (Supplementary Figure 8 in the Supplementary materials). These dynamically driven feedbacks are more evident in the 700-hPa wind (Figure R1a-c), while the signal is weaker at 850-hPa because of the influence of topographical changes in lower atmospheric levels (Figure R1d-f).

To address the reviewer's concern, we have now included the corresponding 850-hPa wind as Supplementary Figure 3 in the revised Supplementary materials for comparison. We have also clarified this in Lines 128-130 of the revised manuscript.

(2) The initial figure used an inappropriate color bar, made land rainfall changes appear less prominent. In response, we have adjusted the color bar scale to better capture these variations and conducted a Student's *t*-test to assess statistical significance (Figure R1a-c). This reveals that significant changes occur not only over the Arabian Sea but also over the Indian Continent and Bay of Bengal. Accordingly, we have updated Figure 1d-f in the revised manuscript, ensure that all displayed wind vectors and rainfall changes are statistically robust.

Figure R1. Changes in rainfall (shading, mm day^{-1}) low-level atmospheric circulation at 700 hPa (vectors, m s^{-1}) in the **a** MT25, **b** MT15 and **c** MT05 simulations compared to pre-industrial simulations. **d-f** Same as **a-c**, but for the changes in low-level atmospheric circulation at 850 hPa (vectors, m s^{-1}). Gray stippling and vectors are only shown at regions where the changes are significant at the 95% confidence level according to Student's t -test.

#2. Fig. 2 – IOD type bias – Cold SST anomalies over IOD east pole must result in suppressed rainfall over the region, the same is not clearly seen in Fig. 1 (d-f).

Our response: We appreciate the reviewer's careful observation regarding the expected rainfall suppression over the east pole of Indian Ocean dipole (IOD)-like pattern. In the initial submission, this effect was not clearly visible due to the inappropriate color bar that we chose in Figure 1d-f. To address, we have replotted Figure 1d-f (see Figure R2a-c) with: a more color scale to better capture spatial variations in rainfall; and student's t -test for both rainfall and wind fields to highlight statistically significant changes.

After these adjustments, as expected, the significantly suppressed rainfall is now clearly visible over the eastern IOD pole, especially in Early and Middle Miocene simulations.

To further validate this, we analyzed additional key variables associated with convective activity. (1) Outgoing longwave radiation (OLR) (Figure R2d-f): Higher OLR values over the eastern IOD pole indicate reduced cloud cover and suppressed convection. (2) Moisture condensation (Figure R2g-i): Reduced latent heat release over the same region confirms a decrease in convective activity and precipitation.

These results consistently demonstrate that the cold SST over the IOD-like east pole indeed result in significant rainfall suppression during the Miocene simulations. We have updated Figure 1d-f in the revised manuscript to include the corrected color scale and significance test.

Figure R2. Responses of South Asian summer monsoon (SASM) to African topography changes during boreal summer. Changes in rainfall (shading, mm day^{-1}) low-level atmospheric circulation at 700 hPa (vectors, m s^{-1}) in the **a** MT25, **b** MT15 and **c** MT05 simulations compared to pre-industrial simulations. **d-f** Same as **a-c**, but for the changes in outgoing longwave radiation (OLR, W m^{-1}). **g-i** Same as **a-c**, but for the changes in moisture condensation (Q_L ; shading, $10^{-2} \text{ m}^2 \text{ s}^{-3}$; see Methods for details) at 500 hPa. Gray stippling and vectors are only shown at regions where the changes are significant at the 95% confidence level according to Student's t -test.

#3. Impact on IOD type bias – Is lowering SLP in western Indian Ocean sufficient to justify the enhanced easterlies along the equator? The magnitude of cooling in the IOD east pole region seems to be almost of similar magnitude in all three simulations. Positive IOD-type conditions will possibly enhance the Indian monsoon rainfall in all the three simulations.

Our response: This is really good and important question. To address your questions, we conducted a more detailed analysis of the monthly evolution of sea surface temperature (SST), wind anomalies and sea level pressure (SLP), which is now presented in Figure R3 and R4.

Given the importance of positive SST anomalies in the western equatorial Indian Ocean for establishing positive ocean-atmosphere feedbacks, we further examined the seasonal cycle of SST and the Somali Jet in this region. Our findings indicate that SST changes in the western Indian Ocean are closely linked to variations in the intensity of the Somali Jet during the Miocene (Figure R3). Specifically, a weakened Somali Jet leads to surface ocean warming by changing surface heat fluxes, consistent with findings from previous studies (Schott et al. 2009; Li et al., 2015). This warming emerges in the pre-monsoon months and intensifies during boreal summer when the Somali Jet is typically strongest.

The seasonal variation presented in Figure R4 confirm that positive SST anomalies over the western Indian Ocean significantly decrease SLP, thereby inducing easterly wind anomalies along the equator. Specifically, warm SST anomalies appear in the western equatorial Indian Ocean by June, accompanied by significant easterly wind anomalies extending from west to central Indian Ocean. In the following months, this warming intensifies and easterly wind anomalies migrate further eastward, while the southeastern tropical Indian Ocean begins to cool down. These processes are particularly pronounced in the Early and Middle Miocene when the East African Highlands were much lower compared to the Late Miocene (Figure R4). As expected, our analysis demonstrates that IOD-like warming patterns are stronger in the Early and Middle Miocene compared to the Late Miocene. This

supports the conclusion that lowered East African Highlands played a key role in amplifying positive IOD-like conditions, which in turn enhanced monsoon rainfall.

We have clarified these findings in Lines 261-269 of the revised manuscript and included the Figure R3 and Figure R4 as Supplementary Figure 9 and Figure 10 in the revised Supplementary materials, respectively. Thank you for guiding us toward a more robust and detailed analysis.

Reference:

Schott, F.A., Xie, S.P. and McCreary Jr, J.P., 2009. Indian Ocean circulation and climate variability. *Reviews of Geophysics*, 47(1).

Li, G., Xie, S.P. and Du, Y., 2015. Monsoon-induced biases of climate models over the tropical Indian Ocean. *Journal of Climate*, 28(8), pp.3058-3072.

Figure R3. Strong coupling of the intensity of Somali Jet and sea surface temperature (SST) over the western Arabian Sea. Seasonal variation of changes in intensity of Somali Jet (m s⁻¹; see Methods) and SST (°C) over tropical western Indian Ocean (10°

S-10° N, 40° -55° E). The vertical shaded areas represent the boreal summer (June to August).

Figure R4. Seasonal variation. Changes in SST (shading, °C), sea level pressure [contours; hPa; solid (dashed) lines represent the positive (negative) values], and surface wind (vectors, m s^{-1}) from May to September in the MT25 (left column), MT15 (middle column) and MT05 (right column) simulations compared to pre-industrial simulation. Gray stippling and vectors are only shown at regions where changes are significant at the 95% confidence level according to Student's t -test.

#4. The impact of IOD dynamics on the model can imply a possible association with ENSO as well. The ENSO-IOD coupling is documented in the literature. Also show a global map of SSTs and indicate whether there is a change in SSTs globally.

Our response: We appreciate the reviewer's thoughtful comment regarding the potential link between the IOD and ENSO in our model simulations. We acknowledge that the IOD-ESNO relationship has been extensively studied in the context of interannual climate variability (Saji et al., 1999; Webster et al., 1999; Cai et al., 2013, 2019). However, it is important to clarify the distinction between the IOD as an interannual phenomenon and the IOD-like warming pattern observed in our study, which reflects long-term changes in the mean state of the ocean-atmosphere system induced by African topography changes.

The positive IOD is initiated by enhanced ocean upwelling along the Java and Sumatra coasts, leading to cooler SSTs in the eastern Indian Ocean and warmer in the western Indian Ocean, accompanying by easterly wind anomalies over the tropical Indian Ocean (Figure R5). In our simulations, we observe a similar west-warming and east-cooling SST pattern, but it is not a manifestation of interannual IOD variability. Instead, it represents a tectonically forced, long-term change in the climate mean state due to altered African topography (Figure R6). Our simulations, which span several hundred years, reach a quasi-equilibrium state, and we analyze the climatological mean of the last 100 years. Therefore, we refer to this pattern as positive IOD-like warming pattern, distinct from interannual varying IOD events.

The interaction between IOD and ENSO has been widely studied, with many studies suggesting that IOD events have been remotely forced by ENSO variability in the Pacific (Xie et al. 2002; Annamalai et al. 2003; Fischer et al. 2005; Zhang et al. 2015; Fan et al. 2017; Liu et al. 2017; Stuecker et al. 2017). However, some IOD events, for example, the event in 1961, are believed to have been the result of local air-sea interactions in the tropical Indian Ocean, independent of ENSO forcing (Saji et al. 1999; Webster et al. 1999; Ashok et al. 2003; Li et al. 2003; Rao and Behera 2005; Wang et al. 2016). Additionally, modeling and observational studies suggest that ENSO is not necessarily a prerequisite for IOD events (Fischer et al. 2005;

Drbohlav et al. 2007). Our study focuses on topography-induced IOD-like warming pattern anomalies as mentioned above, rather than the interannual variability of IOD events. Examining the relationship between ENSO and IOD on tectonic time-scales is a fascinating topic but falls beyond the scope of our current work. We acknowledge that further investigations are required to explore the possible interaction between African topography changes, IOD dynamics, and ENSO over geological time scales, and we propose this as a potential avenue for future research.

In summary, the IOD-like warming pattern in our study is the change in the climate mean state induced by altered topography but not the IOD of interannual variability. This is clarified in Lines 252-253, 274-276 of revised manuscript. And we have further conducted a full text check, ensure consistent terminology throughout the text, i.e., IOD-like warming pattern instead of IOD pattern, to avoid misunderstandings.

Regarding global SSTs, remarkably changes are observed in the Northern Hemisphere, particularly in the North Atlantic (Figure R7). These anomalies may be attributed to the fluctuations in the Atlantic Meridional Overturning Circulation (AMOC). EC-Earth3 simulations exhibit multi-centennial climate variability, which is most pronounced in the North Atlantic region due to fluctuations in AMOC (Figure R8). This phenomenon was identified in our previous study (Cao et al., 2023). Additionally, our recent research indicates that the impact of this multi-centennial variability on subtropical monsoons, such as the SASM, may be less significant compared to its effect on northern high latitudes (Han et al., 2024). However, examining the global SSTs is another fascinating topic but falls beyond the scope of our current work. Nonetheless, we have conducted sensitivity tests (Figure not shown), examining the last 50 years of the model output, and found that our conclusions remain robust and consistent across these different sampling lengths.

Figure R5. Regression of summer SST (shading, $^{\circ}\text{C } ^{\circ}\text{C}^{-1}$) from the Hadley Centre and 1000-hPa wind (vector, $\text{m s}^{-1} ^{\circ}\text{C}^{-1}$) from National Centers for Environmental Prediction reanalysis 2 (NCEP2) on summer Indian Ocean Dipole (IOD) index from 1980 to 2010. The IOD index is defined as the SST anomaly difference between the regions of 50° - 70° E, 10° S- 10° N and 90° E- 110° E, 10° S- 0° (Saji et al., 1999). Shading and black vectors denote regions in which the anomalies are significant at the 95% confidence level according to Student's t -test.

Figure R6. Changes in sea SST (shading, °C) and 1000-hPa wind (vectors, m s^{-1}) in the MT25, MT15 and MT05 simulations compared to pre-industrial simulations, respectively. Gray stippling and vectors denote regions in which the changes are significant at the 95% confidence level according to Student's *t*-test.

Figure R7. Changes in sea SST (shading, °C) in the MT25, MT15 and MT05 simulations compared to pre-industrial simulations, respectively. Gray stippling and vectors denote regions in which the changes are significant at the 95% confidence level according to Student's *t*-test.

Figure R8. Changes in AMOC index (Sv) in the MT25, MT15, MT05 and pre-industrial simulations.

Reference:

- Annamalai, H., R. Murtugudde, J. Potemra, S. P. Xie, P. Liu, and B. Wang, 2003: Coupled dynamics over the Indian Ocean: Spring initiation of the zonal mode. *Deep-Sea Res. II*, 50, 2305–2330.
- Ashok, K., Z. Y. Guan, and T. Yamagata, 2003: A look at the relationship between the ENSO and the Indian Ocean dipole. *J. Meteor. Soc. Japan*, 81, 41–56.
- Cai, W., Wu, L., Lengaigne, M., Li, T., McGregor, S., Kug, J.-S., Yu, J.-Y., Stuecker, M.F., Santoso, A., Li, X., Ham, Y.-G., Chikamoto, Y., Ng, B., McPhaden, M.J., Du, Y., Dommenges, D., Jia, F., Kajtar, J.B., Keenlyside, N., Lin, X., Luo, J.-J., MartínRey, M., Ruprich-Robert, Y., Wang, G., Xie, S.-P., Yang, Y., Kang, S.M., Choi, J.-Y., Gan, B., Kim, G.-I., Kim, C.-E., Kim, S., Kim, J.-H., Chang, P., 2019. Pan-tropical climate interactions. *Science* 363, eaav4236.
- Cai, W., Zheng, X.-T., Weller, E., Collins, M., Cowan, T., Lengaigne, M., Yu, W., Yamagata, T., 2013. Projected response of the Indian Ocean Dipole to greenhouse warming. *Nat. Geosci.* 6, 999e1007.
- Cao, N., Zhang, Q., Power, K.E., Schenk, F., Wyser, K. and Yang, H., 2023. The role of internal feedbacks in sustaining multi-centennial variability of the Atlantic Meridional Overturning Circulation revealed by EC-Earth3-LR simulations. *Earth and Planetary Science Letters*, 621, 118372.
- Drbohlav, H. K. L., S. Gualdi, and A. Navarra, 2007: A diagnostic study of the Indian Ocean dipole mode in El Niño and non-El Niño years. *J. Climate*, 20, 2961–2977.
- Fan, L., Q. Y. Liu, C. Z. Wang, and F. Y. Guo, 2017: Indian Ocean dipole modes associated with different types of ENSO development. *J. Climate*, 30, 2233–2249.
- Fischer, A. S., P. Terray, E. Guilyardi, S. Gualdi, and P. Delecluse, 2005: Two independent triggers for the Indian Ocean dipole/zonal mode in a coupled GCM. *J. Climate*, 18, 3428–3449.
- Han, Z., Power, K., Li, G. and Zhang, Q., 2024. Impacts of Mid - Pliocene Ice Sheets and Vegetation on Afro - Asian Summer Monsoon Rainfall Revealed by EC-Earth Simulations. *Geophysical Research Letters*, 5, p.e2023GL106145.

Li, T, B. Wang, C. P. Chang, and Y. S. Zhang, 2003: A theory for the Indian Ocean dipole-zonal mode. *J. Atmos. Sci.*, 60, 2119–2135.

Liu, L., G. Yang, X. Zhao, L. Feng, G. Q. Han, Y. Wu, and W. D. Yu, 2017: Why was the Indian Ocean dipole weak in the context of the extreme El Niño in 2015? *J. Climate*, 30, 4755–4761.

Rao, S. A., and S. K. Behera, 2005: Subsurface influence on SST in the tropical Indian Ocean: Structure and interannual variability. *Dyn. Atmos. Oceans*, 39, 103–135.

Saji, N.H., Goswami, B.N., Vinayachandran, P.N. and Yamagata, T., 1999. A dipole mode in the tropical Indian Ocean. *Nature*, 401(6751), pp.360-363.

Stuecker, M. F., A. Timmermann, F. F. Jin, Y. Chikamoto, W. J. Zhang, A. T. Wittenberg, E. Widiasih, and S. Zhao, 2017: Revisiting ENSO/Indian Ocean dipole phase relationships. *Geophys. Res. Lett.*, 44, 2481–2492.

Wang, H., R. Murtugudde, and A. Kumar, 2016: Evolution of Indian Ocean dipole and its forcing mechanisms in the absence of ENSO. *Climate Dyn.*, 47, 2481–2500.

Webster, P. J., A. M. Moore, J. P. Loschnigg, and R. R. Leben, 1999: Coupled ocean–atmosphere dynamics in the Indian Ocean during 1997–98. *Nature*, 401, 356–360.

Xie, S. P., H. Annamalai, F. A. Schott, and J. P. McCreary, 2002: Structure and mechanisms of south Indian Ocean climate variability. *J. Climate*, 15, 864–878

Zhang, W. J., Y. L. Wang, F. F. Jin, M. F. Stuecker, and A. G. Turner, 2015: Impact of different El Niño types on the El Niño/IOD relationship. *Geophys. Res. Lett.*, 42, 8570–8576.

#5. Conventionally, the Indian Monsoon is impacted by the IOD by modulation of the local Hadley circulation. The discussion here seems to indicate that atmospheric variability is the stronger driver. Are there other observational studies substantiating this? It is most probably coupled variability. How well does the PI simulation capture the IOD-SASM relation?

Our response: Thanks for your comments. To evaluate how well the EC-Earth3 PI simulation captures the IOD-SASM relationship, we performed a regression analysis of summer SST, 850-hPa wind and rainfall on summer IOD index for both observation data and the PI simulation. The results (Figure R9) show that the PI simulation successfully replicates the observed IOD pattern, which is characterized by cooler SSTs off Sumatra and warmer SSTs in the western Indian Ocean, accompanied by easterly wind anomalies over tropical Indian Ocean. Increased rainfall over the western Indian Ocean and decreased rainfall over the eastern tropical Indian Ocean, consistent with the modulation of atmospheric convection (Webster et al., 1999). In addition, according to Gill's theory (Gill, 1980), the suppressed atmospheric convection over the eastern Indian Ocean produces a pair of anticyclonic circulation anomalies in the off-equatorial region, indicating a Rossby wave response. This feature is evident in both observation and the PI simulation. These results confirm that EC-Earth3 model can capture the interannual variability of the IOD and its impact on SASM.

As discussed in response to comment 4, our study focuses on the changes in climate mean state driven by African topography changes, rather than the interannual variability of IOD. While our IOD-like warming pattern shares some similarities with the interannual IOD (e.g. west-warming, east-cooling SST pattern), the underlying mechanisms are distinct.

To address the reviewer's concern about the impact of IOD-like warming on the local Hadley circulation, we further examined changes in meridional circulation. Our results (Figure R10) indicate that the local Hadley circulation was significantly stronger during the Early and Middle Miocene when compared to the Late Miocene. This strengthening can be attributed to two mechanisms: (1) The southerly wind anomalies in the central Indian Ocean were much stronger during the Early and Middle Miocene, enhancing low-level moisture transport toward the SASM region (Figure R6). (2) The weakening of the Somali jet linked to the African topography changes, and reduced rainfall over tropical Africa contributed to intensified cyclonic circulation in the Arabian Sea (Figure R1a-c). This, in turn, enhance the low-level

meridional circulation, favoring a strengthen local Hadley circulation in the Indian Ocean.

Figure R9. **a** Regression of summer SST (shading, $^{\circ}\text{C } ^{\circ}\text{C}^{-1}$) from the Hadley Centre, 850-hPa wind (vector, $\text{m s}^{-1} \text{ } ^{\circ}\text{C}^{-1}$) from NCEP2 and rainfall (contour, $\text{mm day}^{-1} \text{ } ^{\circ}\text{C}^{-1}$; solid lines represent positive values) from the Global Precipitation Climatology Project (GPCP) on summer IOD index from 1980 to 2010. **b** Same as **a**, but for the variable regression in pre-industrial simulations of EC-Earth3 model. The IOD index is defined as the SST anomaly difference between the regions of $50^{\circ}\text{-}70^{\circ}\text{ E}$, $10^{\circ}\text{ S-}10^{\circ}\text{ N}$ and $90^{\circ}\text{ E-}110^{\circ}\text{ E}$, $10^{\circ}\text{ S-}0^{\circ}$ (Saji et al., 1999). Shading, contours and black vectors denote regions in which the anomalies are significant at the 95% confidence level according to Student's t -test.

Figure R10. The climatologies and changes in zonally averaged (60° E- 95° E) meridional circulation induced by African topography changes during summer. The climatologies in meridional (m s^{-1}) and vertical ($-10^{-2} \text{ Pa s}^{-1}$) winds (vectors) in the pre-industrial simulations and the changes in vertical velocity (color shaded; $-10^{-2} \text{ Pa s}^{-1}$) in the **a** MT25, **b** MT15 and **c** MT05 simulations relative to the pre-industrial simulation.

Reference:

Gill, A. E. Some simple solutions for heat-induced tropical circulation. *Quart. J. Royal. Meteor. Soc.* **106**, 447–462 (1980).

Saji, N.H., Goswami, B.N., Vinayachandran, P.N. and Yamagata, T., 1999. A dipole mode in the tropical Indian Ocean. Nature, 401(6751), pp.360-363.

Webster, P. J., A. M. Moore, J. P. Loschnigg, and R. R. Leben, 1999: Coupled ocean–atmosphere dynamics in the Indian Ocean during 1997–98. Nature, 401, 356–360.

#6. What is the major objective of this study – documenting the impact of orography on SASM or reconciling the proxies? Since there are major changes in other forcing, as mentioned in L338-346, which may significantly add to the uncertainty, is it not better suited just to do present-day simulations with changes to orography?

Our response: Thank you for raising this important point, as it allows us to clarify the primary objective of our study and the rational behind our approach.

The main objective of this study is to document the impact of African orography on SASM evolution during the Miocene, rather than simply reconciling proxy records. Our aim is to isolate and quantify the role of African topography changes in shaping the Somali Jet, monsoon circulation, and associated ocean-atmosphere feedbacks, which remain poorly understood despite their potential significance.

We appreciate that multiple boundary forcings impacted the SASM changes during the Miocene, including the uplift of the Himalaya-Tibetan Plateau, tectonic changes in the Anatolian-Iranian plateau, emergence of land in Eastern Arabian Peninsula, and development of polar ice sheets and CO₂ fluctuations. However, while these factors played a role, previous studies suggest that African topography was a key driver of SASM evolution. Recent study of Sarr et al. (2022) found that the Himalayan and Tibetan Plateau topography influenced early Miocene rainfall patterns, its impact on ocean–atmosphere circulation was relatively limited. This aligns with the evidence indicating that the Himalaya-Tibet uplift was largely complete by ~20 Ma (Ding et al., 2022), suggesting that its primary influence on SASM was in early Miocene or before (Ding et al., 2017; Clif and Webb, 2019; Tomson et al., 2021). Additionally, Yao et al. (2023) emphasize that interhemispheric ice-sheet growth has weakened the SASM through atmospheric anomalies since the Middle Miocene. However, their studies cannot explain the strengthening of the wind-

based proxies in the western Arabian Sea. Therefore, the discrepancies between oceanic wind-based records (Kroon et al., 1991; Huang et al., 2007; Gupta et al., 2015) and rain-related records (Quade et al., 1989; Quade and Cerling, 1995; Quade et al., 1995; Clift 2006; Clift et al., 2008; Lee et al., 2020) suggest that other boundary forcings may also play a crucial role in the multi-stage evolution of the SASM system during the Miocene.

In contrast, the gradual uplift of the East African highlands and subsidence of the Congo Basin occurred between the Middle Miocene to Late Miocene (Moucha and Forte, 2011), a period that coincides with the onset of a “modern-like” SASM around 13 Ma (Nigrini, 1991; Huang et al., 2007; Gupta et al., 2015; Betzler et al., 2016; Zhuang et al., 2017; Bialik et al., 2020). These findings strongly suggest that the evolution of African topography during the Miocene played a critical role in reorganizing the Somali Jet and monsoon rainfall patterns during this period.

In our study, we test the hypothesis that the impact of African topography influenced the multi-stage evolution of the SASM system during the Miocene. Using a high-resolution version of the coupled EC-Earth3 model, we simulate the climate response to evolution of Miocene African topography based on reconstructed African dynamic topography (Moucha and Forte, 2011). By isolating the single boundary condition of African topography change, we can effectively separate its impact from other influences on the SASM. Utilizing the actual reconstructed African topography allows us to identify potential drivers of some phenomena observed in reconstructed proxies. Our results, as anticipated, demonstrate a clear decoupling of monsoon winds and rainfall, supporting the findings of Sarr et al. (2022). The key novelty of our study is the identification of a previously overlooked tectonic mechanism driving the decoupling of SASM winds and rainfall during the Miocene.

We have expanded our discussion of various boundary conditions in the introduction section, emphasizing that the African topography may be the most prominent driver. Consequently, this forcing is the primary focus of our investigation. The clarification can be found in Lines 72-76 of the revised manuscript. In addition,

we also revised the Conclusion (Lines 356-375) to emphasize the novel mechanistic insights uncovered in our study.

Reference:

Bialik, O. M. et al. *Monsoons, upwelling, and the deoxygenation of the Northwestern Indian ocean in response to middle to late miocene global climatic shifts*. 2020. *Paleoceanogr. Paleoclimatol.* 35, e2019PA003762.

Betzler, C. et al. 2016. *The abrupt onset of the modern South Asian Monsoon winds*. *Sci. Rep.* 6, 29838.

Clif, P. D. and Webb, A. A. G. 2019. *History of the Asian monsoon and its interactions with solid earth tectonics in Cenozoic South Asia*. *Geol. Soc. Lond. Spec. Publ.* 483, 875–880.

Clift, P. D. 2006. *Controls on the erosion of Cenozoic Asia and the flux of clastic sediment to the ocean*. *Earth Planet. Sci. Lett.* 241, 571–580.

Clift, P. D. et al. 2008. *Correlation of Himalayan exhumation rates and Asian monsoon intensity*. *Nat. Geosci.* 1, 875–880.

Chakraborty, A., Nanjundiah, R. S. & Srinivasan, J. *Impact of African orography and the Indian summer monsoon on the low-level Somali jet*. *Int. J. Clim.* 29, 983–992 (2009).

Ding, L., Kapp, P., Cai, F., Garzzone, C.N., Xiong, Z., Wang, H. and Wang, C., 2022. *Timing and mechanisms of Tibetan Plateau uplift*. *Nature Reviews Earth & Environment*, 3(10), pp.652-667.

Ding, L. et al. 2017. *Quantifying the rise of the Himalaya orogen and implications for the South Asian Monsoon*. *Geology* 45, 215–218.

Gupta, A. K., Yuvaraja, A., Prakasam, M., Clemens, S. C. & Velu, A. 2015. *Evolution of the South Asian monsoon wind system since the late Middle Miocene*. *Palaeogeogr. Palaeoclimatol. Palaeoecol.* 438, 160–167.

Huang, Y., Clemens, S. C., Liu, W., Wang, Y. & Prell, W. L. 2007. *Large-scale hydrological change drove the late Miocene C4 plant expansion in the Himalayan foreland and Arabian peninsula*. *Geology* 35, 531–534.

- Kroon, D., Steens, T. & Troelstra, S. R. 1991. Onset of monsoonal related upwelling in the western Arabian sea as revealed by planktonic foraminifers 1. In *Proceedings of the ocean drilling program, scientific results 1*, 257–263.
- Lee, J. et al. 2020. Monsoon-influenced variation of clay mineral compositions and detrital Nd-Sr isotopes in the western Andaman Sea (IODP Site U1447) since the late Miocene. *Palaeogeogr. Palaeoclimatol. Palaeoecol.* **538**, 109339.
- Moucha, R. and Forte, A. M. Changes in African topography driven by mantle convection. *Nat. Geosci.* **4**, 707–712 (2011).
- Nigrini, C. 1991. Composition and biostratigraphy of radiolarian assemblages from an area of upwelling (northwestern Arabian Sea, lag 117). *Proc. ODP Sci. Results 117*, 89–126.
- Quade, J., Cerling, T. E. & Bowman, J. R. 1989. Development of Asian monsoon revealed by marked ecological shift during the latest Miocene in northern Pakistan. *Nature* **342**, 163–166.
- Quade, J. & Cerling, T. E. 1995. Expansion of C4 grasses in the Late Miocene of Northern Pakistan: evidence from stable isotopes in paleosols. *Palaeogeogr. Palaeoclimatol., Palaeoecol.* **115**, 91–116.
- Quade, J., Cater, J. M. L., Ojha, T. P., Adam, J. & Mark Harrison, T. 1995. Late Miocene environmental change in Nepal and the northern Indian subcontinent: Stable isotopic evidence from paleosols. *GSA Bulletin* **107**, 1381–1397.
- Sarr, A.C., Donnadiou, Y., Bolton, C.T., Ladant, J.B., Licht, A., Fluteau, F., Laugié, M., Tardif, D. and Dupont-Nivet, G., 2022. Neogene South Asian monsoon rainfall and wind histories diverged due to topographic effects. *Nature Geoscience*, 15(4), pp.314-319.
- Slingo, J., Spencer, H., Hoskins, B., Berrisford, P. & Black, E. The meteorology of the Western Indian Ocean, and the influence of the East African Highlands. *Phil. Trans. R. Soc. A.* **363**, 25–42 (2005).
- Tomson, J. R. et al. 2021. Tectonic and climatic drivers of Asian monsoon evolution. *Nat. Commun.* **12**, 4022.

Wei, H.-H. & Bordoni, S. *On the role of the African topography in the South Asian monsoon. J. Atmos. Sci.* **73**, 3197–3212 (2016).

Zuo, M., Sun, Y., Zhao, Y., Ramstein, G., Ding, L. and Zhou, T., 2024. *South Asian summer monsoon enhanced by the uplift of the Iranian Plateau in Middle Miocene. Climate of the Past*, 20(8), pp.1817-1836.

Zhuang, G., Pagani, M. & Zhang, Y. G. 2017. *Monsoonal upwelling in the western Arabian Sea since the middle Miocene. Geology* **45**, 655–658.

#7. Please confirm if only significant differences are shown in the figures.

Our response: Following your valuable comment, we have replotted all the figures and performed additional statistical analyses to assess the significance of the variable changes in the revised manuscript and Supplementary materials. Specifically, we applied Student's *t*-test to assess the statistical significance, and updated the figures to show only statistically significant differences at 95% significant level.

Minor comments

#1. L73 – other boundary forcings

Our response: Thanks for your carefully reading. We have revised this in Line 71 of the revised manuscript.

#2. L147 – C4 "plants"

Our response: Really thanks for your carefully reading. We have revised this in Line 471 of the revised manuscript.

Reviewer #1 (Remarks on code availability):

Code is not provided.

Our response: Thanks for your reminder. We have provided the code for the main results in the new submission on Zenodo at <https://doi.org/10.5281/zenodo.14905875>.

Reply to Reviewer #2's comments

Remarks to Author:

The Somali Jet, a near-surface air current along the coast of the Horn of Africa, plays a central role in the South Asian summer monsoon (SASM) dynamics. It transports large amounts of moist air from the Indian Ocean to India, where it causes heavy rainfall. A more intense Somali Jet is usually assumed to lead to more precipitation in the SASM domain. For the Miocene, however, reconstructions have shown that these two components appear to be decoupled, as wind proxies show a strengthening of the Somali jet from Middle Miocene to late Miocene, but precipitation in the SASM domain decreased.

Han et al. employ the EC-Earth3 model to assess the impact of changes in the African topography during the Miocene on this decoupling and the atmospheric and ocean dynamics and the South Asian monsoon precipitation, in general. To this end, they carry out simulations in which they keep all boundary conditions on the pre-industrial settings, but prescribe the African topography in line with reconstructions.

According to the model, the topography is important in two respects: First, due to the lower East African Highlands during Early Miocene, the Somali Jet is less canalized and therefore much weaker. This in turn has the effect of inducing a positive Indian Ocean Dipole-like SST pattern, which leads to more precipitation in the SASM domain and an positive feedback with the circulation. In the course of the Miocene, the East African Highlands are lifted up, which more and more intensifies the jet, explaining the signal seen in the wind-proxies.

Second, A higher Congo Basin leads to less ascent and less latent heat release into the atmosphere. This promotes the formation of a cyclonic circulation anomaly above the Arabian sea, redirecting moisture to the Indian continent. The effect of changes in latent heat release on the atmospheric circulation in distinct regions is further investigated by employing a linear baroclinic model. In addition, Han et al. perform simulations with increased (based on reconstructions) CO₂ levels and with both forcings (CO₂ and topography changes). Decomposing the precipitation response into the effect of atmospheric moisture change (thermodynamic) and atmospheric

circulation changes (dynamic response) shows that topography changes purely lead to more precipitation due to changes in the circulation, whereas the CO₂ effect has a strong thermodynamic contribution to the precipitation change due to its impact on the atmospheric temperature.

This study highlights the effect of the African topography in the SASM evolution and underlines the complexity of the South Asian monsoon dynamics. The figures in the main part are well-chosen and helpful, and are also mostly well presented and easy to understand. The text is clearly and comprehensibly formulated and the individual parts seem well thought out. Methods are carefully applied. The main criticism I have is that I missed the “overarching framework”. The individual sub-chapters don't always seem to build on each other and you get the impression that Han et al lose focus on their core topic and their aim of the study in the course of the manuscript. This means that the paper is not easily comprehensible to a wider readership. However, I believe that this can be solved and therefore suggest a publication in Nature Communication after major revisions.

Our response: Dear reviewer, thank you very much for finding interest in our findings and pointing out the flaws in the analyses/presentation. We have addressed your concerns in a point-by-point manner below, and hope that you will find the added information suitable and sufficient for publication. The original comments are quoted in blue and the changes in the revised manuscript have been highlighted in red.

General comments:

#1. For me, not knowing much about the Miocene so far and therefore not being familiar with the literature, it is not clear what is really new about the study and what is the highlight of the study. According to the title, the main topic is the influence of African topography on the decoupling of wind and precipitation changes during the Miocene. In my opinion, there is no clear final answer to this research question in the study. For instance, in the abstract as main (and only) reason for the circulation and precipitation changes, the lower East African Highlands are mentioned. In contrast, large parts of the result part deal with the effect of the higher Congo basin on the

circulation. Later in the text, the CO₂ experiments are analyzed, but the results neither enter the abstract nor the conclusion. In the conclusion, the East African highland and Congo basin topography changes are mentioned as explanation for the weakening of the Somali Jet and the feedbacks with the precipitation, which is not directly clear and confusing, because the linear baroclinic model (used for the explanation of the effect of topography changes in the Congo basin) show no weakening of the Somali Jet in any simulation. This conclusion is therefore imprecise. Not giving a clear answer to the main research question, makes the individual parts of the analysis seem more like, we did this and then that, rather than being able to recognize a clear common line in it. This is a pity, because the individual parts are important and interesting. I would like the reader to be guided more through the work, and the rationale for conducting the individual analyses to be explained more and the work to be structured more (and more clearly). Particularly the Abstract and the Conclusion should be revised.

Our response: We appreciate the reviewer's detailed and constructive feedback. We recognize that while the individual components of our analysis are scientifically valuable, their integration into a cohesive narrative was not sufficiently clear. Based on this valuable critique, we have undertaken the following revisions to clarify the novelty of our study, ensure coherence, and strengthen the manuscript's structure.

1. About the novelty of this study

The key novelty of our study is the identification of a previously overlooked tectonic mechanism driving the decoupling of SASM winds and rainfall during the Miocene. Unlike previous studies that primarily attribute SASM evolution to Himalaya-Tibet uplift, CO₂ changes, or ice sheet dynamics, we provide compelling evidence that (1) African topography changes played a dominant role in modulating monsoon dynamics. (2) Two distinct processes, driven by different African topographic features, contributed to the decoupling of SASM winds and rainfall: Lower East African Highlands during the Early and Middle Miocene weakened the Somali Jet, reducing wind-driven upwelling and inducing a positive IOD-like warming pattern, which enhanced monsoonal rainfall via ocean-atmosphere feedbacks. Meanwhile, a higher Congo Basin reduced ascent and latent heat release,

triggering a cyclonic circulation anomaly over the Arabian Sea, which redirected moisture transport to the SASM region.

While previous studies (e.g., Sarr et al., 2022) have hinted at the role of African topography in SASM evolution, they did not explore the ocean-atmosphere feedbacks that amplify this effect, nor did they explain how these mechanisms reconciled the discrepancies between wind and rainfall proxies. Our study is the first to provide a comprehensive dynamical explanation of this decoupling phenomenon using a high-resolution coupled climate model.

In the revised manuscript, we have rewritten the Abstract (Lines 29-33, 35-37) to highlight the key findings explicitly, and revised the Conclusion (Lines 342-343, 356-375) to emphasize the novel mechanistic insights uncovered in our study.

2. Improve the structure for clarity and coherence

We acknowledge the reviewer's concern that the manuscript's structure did not clearly guide the reader through the different analyses. To improve the logical flow, we have refined the introduction to explicitly state the study's objectives and hypotheses, reorganized sections so that each analysis builds upon the previous one, and moved the discussion of proxy reconstructions from the Results to the Methods section (Lines 446-498) to keep the focus on the model-based analysis.

3. Strengthening the conclusion for prevision and consistency

The reviewer correctly noted that the linear baroclinic model (LBM) does not show any weakening of the Somali Jet, yet our conclusion suggested otherwise. We recognize this inconsistency and have now clarified in the conclusion that the LBM results specifically assess the impact of latent heat release changes on regional atmospheric circulation rather than the direct weakening of the Somali Jet itself. We have emphasized that the Somali Jet weakening is primarily driven by changes in East African topography in our fully coupled EC-Earth3 simulations. And we integrated the CO₂ experiment findings into the discussion and conclusion, making it clear

that topography-induced changes were the dominant driver of SASM decoupling, while CO₂ changes primarily affected the thermodynamic response.

The clarification can be found in Lines 311-314, 331-336, 342-343, and 344-349 of the revised manuscript.

#2. In addition, the author explains the reduction of the Somali Jet via the reduction of the channel effect by the lower East African Highlands. Just looking at the circulation changes in Fig.1, showing a cyclonic circulation anomaly during the Early and Middle Miocene compared to pre-industrial, the pattern does not really look like a reduction in the channel effect. It looks like a thermally induced dynamic response to the decreased SST in the north-western Arabian sea. The CO₂-forced simulation shows a stronger Somali Jet compared to the topography forced simulation. Does this imply that in the CO₂ simulations the wind and precipitation changes are not decoupled? And that the decoupling is only induced by the topography changes, overcompensating the CO₂ effect in the “real world”? Please clarify.

Our response: Thank you for raising this insightful question. The cyclonic circulation anomaly over the Arabian Sea (Figure R1) during the Early and Middle Miocene compared to pre-industrial is resulted from following two key processes:

1. Weakened meridional cross-equatorial flow due to the lowering of the East African Highlands. To further support this, we have now included 850-hPa wind anomalies (Figure R2), which are more directly influenced by the East African Highlands. These show a clear weakening of the Somali Jet core along East Africa with lower highlands, confirming the channel effect mechanism.
2. Thermally induced dynamic response to SST anomalies. As the reviewer correctly pointed out, the cyclonic circulation patterns over the Arabian Sea are also linked to SST driven atmospheric dynamics. Our analysis reveals that a substantial reduction in summer rainfall over tropical central Africa leads to lower latent heat release, which enhances the cyclonic circulation over the Arabian Sea (Supplementary Figure 5 and 7 in the revised Supplementary materials). This

process is further intensified by a Rossby wave response induced by the increase in SASM rainfall (as shown in the LBM results in Supplementary Figure 8 in the revised Supplementary materials).

Thus, while the lower East African Highlands primarily weakened the Somali Jet via the channel effect, the Arabian Sea cyclonic anomaly is largely driven by ocean-atmosphere feedbacks. To provide a comprehensive understanding of our research findings, we decide to keep the 700-hPa winds in Figure 1d-f in the revised manuscript to illustrate the large-scale circulation changes. We have added the 850-hPa wind anomalies (identical to Figure R2) as Supplementary Figure 3 to explicitly show the weakening of the Somali Jet due to the reduced channel effect, and clarified the discussion in Lines 128-130 in the revised manuscript.

To address your question about the relationship between the changes in Somali Jet and SASM rainfall under CO₂ forcing, we have performed additional analysis and calculated the area mean changes in the intensity of Somali Jet and rainfall in Figure R3. In our simulations of combined effect of African topography and CO₂ forcings, the Somali Jet is weakened while SASM rainfall is increased (Figure R3, and Supplementary Figure 12 in Supplementary materials), demonstrating a clear decoupling of monsoon winds and rainfall. When we only consider the African topography forcing, we can reproduce this decoupling feature (Figure R3, and Figure 1g-h in the manuscript), indicating African topography change is the primary driver of SASM evolution during the Miocene. When we only consider the CO₂ forcing, there is no decoupling of monsoon winds and rainfall in our simulations (Figure R3, and Figure 4 in the manuscript). Instead, both Somali Jet strength and SASM rainfall increase simultaneously, confirming that CO₂ alone does not cause the decoupling phenomena. The corresponding discussion are added in Lines 311-314, 331-336, 342-343 of the revised manuscript. In addition, we also include the area mean changes in the intensity of Somali Jet in Figure 4 and Supplementary Figure 12 in the revised manuscript and Supplementary materials, respectively.

Figure R1. Changes in rainfall (shading, mm day^{-1}) and low-level atmospheric circulation at 700 hPa (vectors, m s^{-1}) in the **a** MT25, **b** MT15 and **c** MT05 simulations relative to the pre-industrial simulations. Black boxes ($5^{\circ}\text{ N}-30^{\circ}\text{ N}$; $60^{\circ}\text{ E}-95^{\circ}\text{ E}$) mark the SASM region, and blue boxes ($10^{\circ}\text{ S}-10^{\circ}\text{ N}$, $40^{\circ}-50^{\circ}\text{ E}$) mark the cross-equatorial Somali Jet. Gray stippling denotes regions in which the changes are significant at the 95% confidence level according to Student's t -test.

Figure R2. Changes in atmospheric circulation due to African topography changes during boreal summer. Climatological mean 850-hPa winds (vectors, m s^{-1}) and its wind speed (shading, m s^{-1}) in the **a** pre-industrial, **b** MT25, **c** MT15 and **d** MT05 simulations. Responses to Miocene African topography changes in **e** MT25-PI, **f** MT15-PI and **g** MT05-PI simulations compared to the pre-industrial simulation. Black boxes mark the SASM region, and blue boxes mark the cross-equatorial Somali Jet. In **e-g**, gray stippling denotes regions in which the changes are significant at the 95% confidence level according to Student's t -test.

Figure R3. Area-averaged changes in **a** Somali Jet intensity (m s^{-1}) and **b** ratios of summer rainfall (%) over the SASM region due to forcings of F_{all} (combined effect of African topography and increased CO_2 concentration), F_{top} (changes in African topography), and F_{CO_2} (increased CO_2 concentration) during Miocene. The asterisks above or below the bars denoting changes that are significant at the 95% confidence level according to Student's t-test.

#3. With respect to this, to really understand how the changes in African topography affect the circulation and precipitation pattern, sensitivity studies are necessary in which either the changes in the Congo basin or the changes in the East African Highlands are prescribed, i.e. simulations in which e.g. the Congo basin elevation is increased, but the East African Highlands are prescribed on pre-industrial settings.

Our response: Thanks for your suggestions regarding the need for additional sensitivity experiments. While such experiments could provide further insights, our

objective in this study is to assess the real-world impact of reconstructed African Miocene topography on SASM evolution rather than conducting idealized sensitivity experiments. Below, we outline how our current approach effectively isolates the roles of different African topographic features using robust comparative analyses, aligning with existing literature.

1. Our results show that the uplift of the Congo Basin during the Miocene significantly weakened the African monsoon, leading to a substantial decrease in precipitation over tropical Africa (Supplementary Figure 7 in the Supplementary materials). This drying trend, in turn, influenced SASM circulation through atmospheric teleconnections. This relationship between the rainfall anomaly over North-central Africa and SASM circulation has been well documented in previous studies (e.g., Li et al., 2022). Using CMIP6 outputs and sensitivity experiments, Li et al. (2022) demonstrated that significant rainfall reductions over tropical Africa can induce a Kelvin wave response over the northern Indian Ocean, altering SASM dynamics. Our results from topography simulations (Figure 1d-f in the manuscript) and LBM experiments (Supplementary Figure 8 in the Supplementary materials) are consistent with their findings, confirming the significant role of African convective changes in modulating SASM circulation and rainfall. Thus, while additional sensitivity experiments could provide further confirmation, our topography-forced simulations already capture the key teleconnections during SASM variability.

2. The East African Highlands play a crucial role in shaping the Somali jet, and our simulations align with previous idealized modeling studies (Slingo et al., 2005; Chakraborty et al., 2009; Wei et al., 2016). These works demonstrated that removing the East African Highlands significantly weakens the Somali Jet by reducing the topographic blocking effect. In our altered topography simulations, the most notable feature of the meridional cross-equatorial flow is a significant weakening of the core of Somali Jet due to the lowering of the East African Highlands during the Miocene (Figure R2, and Figure 1 in the manuscript). The consistency of our simulations with

previous idealized modeling studies gives us confidence that the East African Highlands are a key factor in controlling Somali Jet strength and monsoon dynamics.

3. In the revised manuscript, our further analysis highlights that the weakened Somali Jet leads to positive SST anomalies in the western equatorial Indian Ocean, triggering a positive IOD-like warming pattern that enhances SASM rainfall. To examine this process in more detail, we analyzed the monthly evolution of changes in intensity of the Somali Jet, SST, wind anomalies, and sea level pressure (SLP) (Figures R4 and R5). We found SST changes in the western Indian Ocean are closely linked to variations in the intensity of the Somali Jet during the Miocene (Figure R4). Specifically, a weakened Somali Jet can cause rapid surface ocean warming via changing surface heat flux (Schott et al. 2009; Li et al., 2015) in the western equatorial Indian Ocean, particularly in the pre-monsoon months. These SST anomalies intensify during boreal summer, reinforcing the positive IOD-like warming pattern that strengthens SASM rainfall. In addition, the seasonal variations in Figure R5 confirm that the development of positive SST anomalies over the western Indian Ocean decrease SLP there, inducing easterly anomalies along the equator, sustaining the IOD-like pattern. These processes are prominent in the Early and Middle Miocene, when the East African Highlands were lower, leading to a stronger IOD-like warming pattern compared to the Late Miocene. This, in turn, further increase SASM rainfall through atmospheric teleconnections, as confirmed by our LBM experiments (Supplementary Figure 8 in the Supplementary materials).

In summary, while conducting separate simulations for the Congo Basin and East African Highlands could provide additional insights, we argue that our current Miocene simulations already isolate the individual roles of these regions by comparing their effects on circulation and precipitation patterns. Our results are consistent with previous idealized modelling studies and have explicitly examined these mechanisms. Further analysis of seasonal variation (Figures R4 and R5) supports the validity of the mechanisms proposed in our study, making additional experiments beyond the scope of this study. To ensure this is clearly conveyed, we

have added the Figure R4 and Figure R5 as Supplementary Figure 9 and Figure 10 in the revised Supplementary materials, respectively. The corresponding findings have also been clarified in Lines 261-269 in the revised manuscript.

Figure R4. Strong coupling of the intensity of Somali Jet and sea surface temperature (SST) over the western Arabian Sea. Seasonal variation of changes in intensity of Somali Jet (m s^{-1} ; see Methods) and SST ($^{\circ}\text{C}$) over tropical western Indian Ocean (10°S - 10°N , 40° - 55°E). The vertical shaded areas represent the boreal summer (June to August).

Figure R5. Seasonal variation. Changes in SST (shading, °C), sea level pressure [contours; hPa; solid (dashed) lines represent the positive (negative) values], and surface wind (vectors, m s^{-1}) from May to September in the MT25 (left column), MT15 (middle column) and MT05 (right column) simulations compared to pre-industrial simulation. Gray stippling and vectors are only shown at regions where changes are significant at the 95% confidence level according to Student's t -test.

Reference:

Li, T. et al. *Distinctive South and East Asian monsoon circulation responses to global warming. Sci. Bull.* 67, 762–770 (2022).

Slingo, J., Spencer, H., Hoskins, B., Berrisford, P. and Black, E. *The meteorology of the Western Indian Ocean, and the influence of the East African Highlands. Phil. Trans. R. Soc. A.* 363, 25–42 (2005).

Chakraborty, A., Nanjundiah, R. S. and Srinivasan, J. *Impact of African orography and the Indian summer monsoon on the low-level Somali jet. Int. J. Climatol.* 29, 983–992 (2009).

Wei, H.-H. and Bordoni, S. *On the Role of the African Topography in the South Asian Monsoon. J. Atmos. Sci.* 73, 3197–3212 (2016).

Schott, F.A., Xie, S.P. and McCreary Jr, J.P. *Indian Ocean circulation and climate variability. Reviews of Geophysics,* 47(1) (2009).

Li, G., Xie, S.P. and Du, Y. *Monsoon-induced biases of climate models over the tropical Indian Ocean. Journal of Climate,* 28(8), 3058-3072 (2015).

#4. (1) What do we know about the climate in the Miocene? About the drivers? Other (but not applied) changes in the boundary conditions are only mentioned in the conclusion and discussion. For example, the Western Ghats were much lower, which probably also had an impact on the circulation and the precipitation in India. The Tibetan Plateau was probably only half as high. Today it acts as a strong barrier and the vertical ascending on the southern side is very strong, leading to large amounts of precipitation. In addition, it plays a very important role for the atmospheric circulation, also via the influence on the subtropical jet. So I suspect that there might be more reasons why the wind-precipitation-decoupling may have occurred and the climate was like as seen in the reconstructions. Please discuss this in more detail. It would be nice if this was already mentioned in the introduction and it was made clear that (presumably) the topographical changes and the higher CO₂ levels were the most prominent drivers and therefore only these forcings were used and it is now being investigated how large the proportion of orography is in comparison with the CO₂

forcing proportion. (2) Maybe it would also be an idea to present the orography and CO₂ simulations together, to work out that the topography changes are more important. In the current version of the paper, the authors prescribe topography changes and the main results is that topography changes are important. But this is already a predetermined result, when topography is the only forcing of the model that have been prescribed.

Our response: (1) We acknowledge that a broader discussion of Miocene climate drivers—including tectonic, paleogeographic, and CO₂-related changes—was lacking in the introduction. To address this, we have now expanded the introduction to provide clearer justification for focusing on topographic changes and CO₂ levels as the dominant drivers of SASM evolution during the Miocene.

- About key boundary condition changes and their relevance to SASM system.

The Himalaya-Tibetan Plateau uplift significantly shaped monsoon dynamics before the Miocene. However, previous studies indicate its uplift occurred before ~20 Ma, with southern Tibet reaching elevations ≥ 4 km by ~55 Ma and central Tibet by ~45 Ma (Ding et al., 2022). The western Chats were much lower than today, likely affecting orographic rainfall over the Indian subcontinent. The Iranian Plateau and Eastern Arabian Peninsula were still evolving, which may have influenced regional atmospheric circulation.

- Why we focus on African topography and CO₂ as key SASM drivers?

Recent studies (Sarr et al., 2022) found that while the Himalaya-Tibetan Plateau predominantly influenced early Miocene rainfall, its impact on ocean-atmosphere circulation was relatively limited in the Middle and Late Miocene. In contrast, the gradual uplift of the East African Highlands and the subsidence of the Congo Basin from the Middle Miocene to Late Miocene aligns with the onset of a “modern-like” SASM in the late middle Miocene (~13 Ma) (Nigrini, 1991; Huang et al., 2007; Gupta et al., 2015; Betzler et al., 2016; Zhuang et al., 2017; Bialik et al., 2020). The East African Highlands uplift shaped the Somali Jet, while the Congo Basin uplift modified regional convection, make African topography a key influence on SASM

wind-precipitation decoupling. Although CO₂ levels fluctuated during the Miocene, they remained moderately higher than modern values with large uncertainties (Beerling and royer, 2011; Foster et al., 2012; Super et al., 2018; Sosdian et al., 2020; Steinhorsdottir et al., 2020).

Following your valuable suggestions, we have included a more detailed discussion as mentioned above in the introduction of the revised manuscript to highlight that topographical changes were the most prominent drivers, specifically in Lines 72-76 of revised manuscript.

(2) Thanks for your insightful suggestion regarding the idea of presenting the orography and CO₂ simulations together to emphasize the importance of topography changes. After carefully consideration, we believe that our current structure – focusing first on African topography and then on CO₂ effects – is more suitable for the following reasons.

The core objective of this paper is to evaluate the climate effects of realistic Miocene African topography evolution, rather than conducting idealized sensitivity tests. This is why we focus on three key time slices (early, middle and late Miocene) using reconstructed African dynamic topography. Unlike African topography, which follows a clear geological reconstruction, CO₂ estimates vary widely. During the Miocene, most proxy records indicate that *p*CO₂ was near or moderately higher than modern values (Beerling and royer, 2011; Foster et al., 2012; Super et al., 2018; Sosdian et al., 2020; Steinhorsdottir et al., 2020). However, multi-proxy estimates of atmospheric CO₂ during this period show large uncertainties (Figure 4a in manuscript). Thus, we idealized CO₂ forcing to 500 ppm in sensitivity experiments so isolate its effect without imposing additional uncertainties from varying proxy reconstructions.

Separate mechanisms:

- African topography causes a decoupling of monsoon winds and rainfall, where the Somali Jet weakens while SASM rainfall increases.

- CO₂ forcing, in contrast, enhances both monsoon winds and rainfall simultaneously, showing no decoupling effect.
- When both forcings are combined, the decoupling effect persists, reinforcing that the African topography is the dominant driver of Miocene SASM evolution.

These are clarified in Lines 29-30, 36-37, 142-151, 311-314, 331-336, 342-343, and 412 of the revised manuscript.

Reference:

*Bialik, O. M. et al. Monsoons, upwelling, and the deoxygenation of the Northwestern Indian ocean in response to middle to late miocene global climatic shifts. *Paleoceanogr. Paleoclimatol.* 35, e2019PA003762 (2020).*

*Betzler, C. et al. The abrupt onset of the modern South Asian Monsoon winds. *Sci. Rep.* 6, 29838 (2016).*

*Beerling, D. J., and Royer, D. L. Convergent cenozoic CO₂ history. *Nature Geoscience*, 4, 418–420 (2011).*

*Clif, P. D. and Webb, A. A. G. History of the Asian monsoon and its interactions with solid earth tectonics in Cenozoic South Asia. *Geol. Soc. Lond. Spec. Publ.* 483, 875–880 (2019).*

*Chakraborty, A., Nanjundiah, R. S. and Srinivasan, J. Impact of African orography and the Indian summer monsoon on the low-level Somali jet. *Int. J. Clim.* 29, 983–992 (2009).*

*Ding, L., Kapp, P., Cai, F., Garzzone, C.N., Xiong, Z., Wang, H. and Wang, C.. Timing and mechanisms of Tibetan Plateau uplift. *Nature Reviews Earth & Environment*, 3(10), pp.652-667 (2022).*

*Ding, L. et al.. Quantifying the rise of the Himalaya orogen and implications for the South Asian Monsoon. *Geology* 45, 215–218 (2017).*

Foster, G. L., Lear, C. H., & Rae, J. W. B.. *The evolution of pCO₂, ice volume and climate during the middle Miocene. Earth and Planetary Science Letters*, 341, 243–254 (2012).

Gupta, A. K., Yuvaraja, A., Prakasam, M., Clemens, S. C. and Velu, A. *Evolution of the South Asian monsoon wind system since the late Middle Miocene. Palaeogeogr. Palaeoclimatol. Palaeoecol.* 438, 160–167 (2015).

Huang, Y., Clemens, S. C., Liu, W., Wang, Y. & Prell, W. L. *Large-scale hydrological change drove the late Miocene C4 plant expansion in the Himalayan foreland and Arabian peninsula. Geology* 35, 531–534 (2007).

Moucha, R. and Forte, A. M. *Changes in African topography driven by mantle convection. Nat. Geosci.* 4, 707–712 (2011).

Nigrini, C. *Composition and biostratigraphy of radiolarian assemblages from an area of upwelling (northwestern Arabian Sea, lag 117). Proc. ODP Sci. Results 117*, 89–126 (1991).

Sarr, A.C., Donnadiou, Y., Bolton, C.T., Ladant, J.B., Licht, A., Fluteau, F., Laugié, M., Tardif, D. and Dupont-Nivet, G.. *Neogene South Asian monsoon rainfall and wind histories diverged due to topographic effects. Nature Geoscience*, 15(4), 314-319 (2022).

Slingo, J., Spencer, H., Hoskins, B., Berrisford, P. and Black, E. *The meteorology of the Western Indian Ocean, and the influence of the East African Highlands. Phil. Trans. R. Soc. A.* 363, 25–42 (2005).

Sosdian, S. M., Babila, T. L., Greenop, R., Foster, G. L., and Lear, C. H.. *ocean carbon storage across the middle Miocene: A new interpretation for the Monterey event. Nature Communications*, 11, 134 (2020).

Steinhorsdottir, M., Jardine, P. E., & Rember, W. C.. *Near-Future pCO₂ during the hot Miocene Climatic Optimum. Paleoceanography and Paleoclimatology*, 36, e2020PA003900 (2020).

Super, J. R., Thomas, E., Pagani, M., Huber, M., O'Brien, C., and Hull, P. M.. *North Atlantic temperature and pCO₂ coupling in the early-middle Miocene. Geology*, 46(6), 519–522 (2018).

Tomson, J. R. et al. *Tectonic and climatic drivers of Asian monsoon evolution. Nat. Commun.* **12**, 4022 (2021).

Wei, H.-H. and Bordoni, S. *On the role of the African topography in the South Asian monsoon. J. Atmos. Sci.* **73**, 3197–3212 (2016).

Zuo, M., Sun, Y., Zhao, Y., Ramstein, G., Ding, L. and Zhou, T.. *South Asian summer monsoon enhanced by the uplift of the Iranian Plateau in Middle Miocene. Climate of the Past*, **20**(8), pp.1817-1836 (2024).

Zhuang, G., Pagani, M. and Zhang, Y. G. *Monsoonal upwelling in the western Arabian Sea since the middle Miocene. Geology* **45**, 655–658 (2017).

#5. A large part of the result chapter compares the model with data. For me, this is done very “uncritically”. Although the reconstructions of wind and precipitation agree quite well with the model, the proxies experienced a completely different world than that in the idealized model simulation (see above). I also do not understand, why the reconstructions are at least not compared with the simulation including both forcing (CO₂ and topography changes). And I also miss a critical (at least short) discussion on the uncertainties in the proxies. For instance, d¹⁸O is often used as a proxy for precipitation, but it is also very dependent on other things (moisture source, wind system, temperature). I understand why Han et al. compare the simulations with reconstructions, but if it is to be part of the results, then a more critical discussion would be appropriate. Perhaps it would otherwise be an idea to include the evaluation of the model simulation in the methods (e.g. in the model and simulation description).

Our response: We appreciate this thoughtful comment, as it highlights the need for a more critical evaluation of data-model comparisons and a clearer discussion of proxy uncertainties. To improve the manuscript, we have restructured the presentation of proxy data, refined our discussion on uncertainties, and provided a clearer justification for our comparison approach.

(1) Following your valuable suggestions, to enhance clarity and avoid an uncritical tone, we have changed the title of the section from “Data-model comparison for

SASM evolution” to “Miocene African topography changes cause the decoupling of monsoon winds and rainfall”, emphasizing the mechanistic focus rather than an overly direct model-data validation. And then we moved the detailed proxy discussion to the Methods section (Lines 446-498), ensuring that the core results remain focused on model-based mechanisms. We also explicitly stated in the Methods that proxy reconstructions provide qualitative constraints rather than exact validation of the simulations. This restructuring helps to make it clear that our primary goal is to explore the physical mechanisms behind monsoon evolution, while proxy comparisons serve as supporting evidence rather than definitive proof.

(2) The reviewer raises a fair point about whether the reconstructions should be compared with the simulation that includes both CO₂ and topography changes. We acknowledge this concern and clarify our rationale below:

- Our primary objective is to isolate the role of African topography in SASM evolution. The topography-only simulations allow us to separate the direct impact of African tectonic changes without interference from CO₂-related thermodynamic effects. If we were to compare proxies with the combined CO₂ + topography simulations, it would be harder to disentangle the specific influence of topography.
- CO₂ levels in the Miocene have high uncertainties, making direct model-data comparisons difficult. Multi-proxy CO₂ reconstructions vary significantly (e.g., Beerling and Royer, 2011; Foster et al., 2012; Super et al., 2018; Sossian et al., 2020; Steinthorsdottir et al., 2020). Since we idealized CO₂ to 500 ppm in sensitivity experiments, direct comparison with proxy data could be misleading.

Despite these limitations, we have expanded the discussion in Lines 308-309, 311-314, 331-336, and 342-343, explicitly acknowledging this potential issue and clarifying that while CO₂ may modulate the monsoon, our findings indicate that African topography is the dominant driver of wind-rainfall decoupling.

(3) The reviewer is right that proxy records have inherent uncertainties, which should be acknowledged in the manuscript. We have now incorporated a dedicated discussion in the Methods section (Lines 488-498).

We expanded discussion on $\delta^{18}\text{O}$ limitations: While $\delta^{18}\text{O}$ of soil carbonates is often interpreted as a precipitation proxy, it is also influenced by (1) moisture source variability (changes in the origin of monsoon moisture); (2) wind system shifts (affecting moisture transport pathways); and (3) temperature variations (altering fractionation processes) (Wen et al., 2024a). This means that while $\delta^{18}\text{O}$ data suggest drying trends, they do not provide a direct measure of precipitation amount.

We acknowledged uncertainties in wind proxies (e.g., TOC and *G. bulloides* records): these proxies are indirect indicators of monsoon winds, as they reflect upwelling-driven productivity changes rather than wind speed itself. Upwelling can also be influenced by local oceanographic processes and thermocline variability, adding complexity to their interpretation (Wen et al., 2024b).

We clarified the broader limitations of proxy-model comparisons: (1) Temporal averaging differences: Proxy records often reflect multi-millennial averages, while models analyze mean state differences; (2) Spatial sampling bias: Most proxies come from specific regions (e.g., Arabian Sea, Siwaliks), which may not fully capture spatial monsoon variability.

These discussions ensure that the model-data comparison is not overly deterministic but instead framed as a qualitative consistency check. Additionally, despite these uncertainties, the consistent multi-proxy changes support the conclusion that these proxies reliably reflect SASM evolution during the Miocene.

Reference:

Wen, Q., Liu, Z., Jing, Z., Clemens, S.C., Wang, Y., Yan, M., Ning, L. and Liu, J., 2024a. Grand dipole response of Asian summer monsoon to orbital forcing. *npj Climate and Atmospheric Science*, 7(1), 202.

Wen, Q., Liu, Z., Liu, J., Clemens, S., Jing, Z., Wang, Y., Lv, G., Yan, M., Ning, L., Yuan, L. & Gao, Y., 2024b. Contrasting responses of Indian summer monsoon rainfall and Arabian Sea upwelling to orbital forcing. *Commun. Earth Environ* 5, 409.

#6. Please revise the supplementary figures. The captions and the heading of subfigures are not always clear and correct. I listed a few issues in my specific comments below, but please check all figures carefully before re-submission.

Our response: Thanks for the careful attention to the clarity and accuracy of our supplementary figures. To ensure consistency, correctness, and readability, we have made systematic review and revision of captions and subfigure headings. And we have performed additional statistical analyses for all relevant figures to ensure that reported changes are robust and significant. Specifically, we applied Student's *t*-test to assess statistical significance, with only differences at the 95% confidence level are shown in the figures.

Other specific comments:

Abstract

#1. - L33-L38: Please be more precise and refer to all results (see comment (a)). For instance, “blocking effect” is misleading here, as it is rather the opposite, a reduction of the blocking effect.

Our response: Done.

Introduction

#2. L66: “weakening of the SASM”: As in this study, changes in wind and precipitation are discussed, this term is not clear. It could be both, a weakening in the wind and a weakening in the precipitation.

Our response: Thanks for the comments. In the manuscript, it means the weakening of the SASM rainfall. We have described it more clear in Line 62 of the new revised manuscript as follows:

“This reduction in the SASM rainfall ...” (Line 62)

Results:

#3. L105: it is: decoupling of monsoon rainfall...

Our response: Revised.

#4. L166-L169: This sentence should be moved to the Introduction.

Our response: Thanks for the suggestions. We have moved this sentence to the introduction section in Lines 72-75 of the revised manuscript.

#5. L177-178: “in western Arabian Sea (Fig.1) is written in italics

Our response: Revised.

#6. L183: "These circulations", not "There circulations"

Our response: Revised.

#7. L211-215: Since it is a coupled system, it is difficult to say what triggers the descending air anomaly. The vertical movement in models is often a result of the convection calculation. So it could well be that the weaker West African monsoon leads to less precipitation, e.g. because less moisture is transported to the region and the lower precipitation is causing/accompanied by weaker ascent and less latent heat release. What about the temperature and pressure at the surface? Maybe this would be another (better) line of argumentation.

Our response: We appreciate the reviewer’s insightful suggestions. To strengthen our argument, we have reanalyzed surface changes in temperature and pressure and

added this information into the revised manuscript. Our findings support the reviewer's hypothesis: The weakened West African monsoon circulation leads to reduced moisture transport into central Africa, resulting in lower precipitation. The decreased precipitation is directly associated with weakened ascent and reduced latent heat release, reinforcing the descending motion. Surface analysis reveals a warming anomaly in central Africa, likely linked to suppressed convection, while surface pressure increases in response to reduced latent heating (Figure R6).

Following this additional analysis, we have added the Figure R6 in Supplementary Figure 7, and rewritten the relevant section in Lines 207-213 of the revised manuscript as follows:

“The resulting weakening of the West African summer monsoon cause significantly reduced rainfall over central Africa (Fig. 1d and e). This drying trend is further supported by enhanced descending air anomalies (Supplementary Fig. 7c and g) and a substantial reduction in latent heat release in the region (Supplementary Fig. 7d and h). Additionally, surface warming anomalies over central Africa (likely due to suppressed convection) and increased surface pressure indicate the presence of subsidence driven atmospheric stability (Supplementary Fig. 7e and i).” (Lines 207-213)

Figure R6. Responses of surface summer air temperature change ($^{\circ}\text{C}$) to Miocene African topography changes in **a** MT25, **c** MT15 and **e** MT05 simulations compared to the pre-industrial simulation. **b**, **d**, and **f** Same as **a**, **c**, and **e**, but for the responses of surface pressure (hPa). Gray stippling denotes regions in which the changes are significant at the 95% confidence level according to Student's *t*-test.

#8. L216: I'm not that familiar with Kelvin waves, but I think Kelvin waves are usually confined to the equatorial region ($\sim 10^{\circ}\text{S}$ to $\sim 10^{\circ}\text{N}$). I am also not sure

whether a reduction in the latent heat release causes a Kelvin wave response, or whether, for example, the amplitude of the Kelvin wave simply changes. Please check this with the literature.

Our response: Thanks for your carefully assessment of our discussion on Kelvin waves. To ensure the scientific robustness of our explanation, we have revisited the theoretical framework, checked relevant literature, and validated our results with previous studies.

According to the Gill's theory (Gill, 1980), the response of the tropical atmosphere to localized heating or cooling follows a distinct wave pattern, which includes Rossby waves on the western side of the heat source, and Kelvin waves on the eastern side of the heat source. The spatial extent of the Kelvin wave response depends on the latitudinal position and strength of the heating/cooling anomaly.

Mathematically, Gill's model represents a heating rate Q is symmetric about the equator in the form:

$$Q(x, y) = A \cdot g(x) \cdot e^{-\frac{1}{4}(y+d)^2}$$
$$g(x) = \begin{cases} \cos ks & |x| \leq L \\ 0 & |x| > L \end{cases}, \quad k = \frac{\pi}{2L} \quad (1)$$

where A represents heat intensity, d represents the longitude distance from the equator to the center of the heat source. $d > 0$ and $d < 0$ denote the center of the heat source is located in the southern and northern hemisphere, respectively. $2L$ denotes the zonal width of the heat source.

This formulation helps explain why atmospheric responses to heating sources are strongest near the equator but can extend poleward depending on the structure of the forcing. While Kelvin waves are strongest at the equator, they can extend up to $\sim 20^\circ\text{N/S}$, particularly when the heating or cooling source is asymmetric about the equator (Figure R7), as in Nan et al. (2014). Our LBM experiments confirm that a cooling source over North-Central Africa generates an atmospheric response consistent with Kelvin wave dynamics (Figure R8 and R9), similar to findings by Li

et al. (2022). Thus, our LBM results align well with established Kelvin wave theory and previous modelling studies.

To ensure clarity and scientific rigor, we have rewritten the relevant section in the revised manuscript as follows (Lines 221-229):

“According to Gill’s theory⁴¹, localized atmospheric heating and cooling anomalies generate distinct wave responses in the tropics, where Rossby waves propagate westward and Kelvin waves propagate eastward. While Kelvin waves are strongest near the equator ($\sim 10^\circ$ S– 10° N), their influence can extend up to $\sim 20^\circ$ N, depending on the structure and strength of the forcing⁴⁰. Our LBM experiment demonstrates that suppressed convection over North-Central Africa induces a cooling anomaly, triggering a Kelvin wave response with easterly wind anomalies over the tropical Indian Ocean. This result is consistent with previous study⁴⁰.”

REDACTED

Figure R7. Same as Figure 1 of Nan et al. (2014). Distributions of the isolated equatorially asymmetric heating and cooling sources given in Eq. (1). (a) The heating source with the heating center at $d=1$; (b) as in (a) but for $d=-1$; (c) as in (a) but for the cooling source; (d) as in (b) but for the cooling source. Here d represents the meridional distance between the equator and heating center, $d>0$ and $d<0$ represent that the heating centers are located in the southern and northern hemispheres,

respectively. The thick solid and dashed lines represent heating and cooling sources, respectively, and the contour interval is 0.2

REDACTED

Figure R8. Same as Figure 3 of Nan et al. (2014). Horizontal wind (arrows), vertical velocity (shading), and pressure (thick contours) solutions in the lower layer for the isolated asymmetric heating cases shown in Fig. 1. (a) Analytical solutions of atmospheric variables by the heating in Fig. 1a; (b) as in (a) but for the heating in Fig. 1b; (c) as in (a) but for the heating in Fig. 1c; (d) as in (a) but for the heating in Fig. 1d. Thick solid (dashed) lines represent positive (negative) pressure, and contour interval is 0.3.

REDACTED

Figure R9. Same as Figure 5b of Li et al. (2022). The response of low-level atmospheric wind field (m s^{-1}) to the prescribed heating anomaly over the North Africa ($0^\circ - 20^\circ \text{ N}$, $0^\circ - 55^\circ \text{ E}$) (b) with the maximum heating rate of 1 K day^{-1} .

Reference:

Li, T., Wang, Y., Wang, B., Ting, M., Ding, Y., Sun, Y., He, C. and Yang, G.. Distinctive South and East Asian monsoon circulation responses to global warming. Science Bulletin, 67(7), 762-770 (2022).

Gill, A. E. Some simple solutions for heat-induced tropical circulation. Quart. J. Royal. Meteor. Soc. 106, 447–462 (1980).

Nan Xing, Li Jianping, Li Yaokun. Response of the tropical atmosphere to isolated equatorially asymmetric heating. Chinese Journal of Atmospheric Sciences (in Chinese), 38 (6): 1147–1158 (2014).

#9. L243: ...seen in the LBM experiment results. The “seen in” is missing.

Our response: Revised.

#10. L282: Please add a reference to Gill’s theory.

Our response: Following your valuable comment, we have added the reference of “Gill et al. (1980)” in Line 296 of the revised manuscript.

Gill, A. E. Some simple solutions for heat-induced tropical circulation. Quart. J. Royal. Meteor. Soc. 106, 447–462 (1980).

#11. L319: ... using THE fully coupled EC-Earth3 model.

Our response: Revised.

#12. L328: This conclusion is imprecise, because the LMB model does not show a weakening of the Somali Jet. Please rephrase!

Our response: Following your valuable comment, we have rephrased this confusing sentence in Lines 344-349 of the revised manuscript.

#13. L344-346: I find it unfavorable to use this sentence as the final sentence. It

implies that the evolution of the SASM was the main motivation for the study. This is not clear in the context of the study and is also not clear from the title. Instead, I would like to see a final answer to the reasons for the decoupling and maybe a statement what it means, e.g. for present or future climate. It is not clear from the study which effect predominates, but climate models project strong changes in the precipitation pattern over Africa. Is it possible that we will experience a decoupling of the Somali Jet and SASM precipitation again in the future, due to changes in the latent heat release in Africa?

Our response: Thank you for raising this insightful suggestion. We agree that the conclusion should provide a definitive explanation for the decoupling mechanism and highlight its broader relevance for both past and future climate. To address this, we have revised the final section of the manuscript by clearly stating that African topographic changes were the key driver of the Somali Jet-monsoon rainfall decoupling; emphasizing the role of latent heat release in shaping large-scale atmospheric circulation, reinforcing the importance of understanding past and future African climate dynamics and providing a forward-looking statement on potential future implications, given projected shifts in African precipitation patterns under global warming.

This is clarified in Lines 356-375 of the revised manuscript as follows:

“The key novelty of our study is the identification of a previously overlooked tectonic mechanism that drives the decoupling of SASM winds and rainfall during the Miocene. Unlike previous studies that primarily attribute SASM evolution to the changes in land/ocean configuration^{74,75}, changes in polar ice sheet development^{25,26}, changes in ocean gateways^{23,76}, closure of the Tethys Seaway^{13,74,77}, and uplift of Hinalaya-Tibetan and other regions^{6,7,24}, our results provide compelling evidence that African topography changes were a dominant driver of SASM evolution.

Our high-resolution EC-Earth simulations reconcile contradictory proxy evidence, resolving the discrepancy between wind-based proxies in the Arabian Sea and rain-related proxies over the Indian continent and nearby Indian Ocean for SASM changes from the Middle to Late Miocene. This is the first study to provide a comprehensive dynamical explanation of this decoupling phenomenon using a state-of-the-art coupled climate model, highlighting the importance of accurately reconstructing the timing and magnitude of African topographic changes when interpreting past monsoon evolution. Moreover, as climate models project significant changes in precipitation over Africa due to global warming, similar latent heat-induced atmospheric circulation changes could influence future SASM dynamics. This raises the possibility that a decoupling of the Somali Jet and monsoonal rainfall could occur again in a warming world, with potential consequences for regional hydroclimate, monsoon predictability, and water resource management in South Asia.”

Methods:

#14. L357: which IS the Nucleus for ...

Our response: Revised

#15. L454: “cooling rate of 6 to 7 K/day”... Does it mean that it gets cooler every day? I do not understand this.

Our response: Thanks for the comment. The "cooling rate of 6 to 7 K/day" does not imply a continuous daily drop in temperature. Instead, it refers to the steady-state cooling imposed in the LBM experiments, which remains constant over time rather than getting cooler every day.

To clarify, the LBM developed by Watanabe and Kimoto (2000) is designed to explore the steady-state atmospheric response to a prescribed diabatic heating or cooling anomaly. The applied cooling rate of 6–7 K/day represents a fixed thermodynamic forcing at a given pressure level, rather than a progressive daily decrease in temperature.

In the thermodynamic energy equation (Yanai et al., 1973; Knutson and Manabe, 1995), the diabatic heating (or cooling) can be written as:

$$\frac{Q}{c_p} = \frac{\partial T}{\partial t} + \vec{V} \cdot \nabla T - \left(\frac{p}{p_0}\right) \omega \frac{\partial \theta}{\partial p} \quad (2)$$

where T , θ , \vec{V} , p , and ω are the temperature, potential temperature, horizontal wind, pressure, and vertical pressure velocity, respectively; c_p is the specific heat of dry air at constant pressure, and p_0 is the reference pressure (1000 hPa). The term Q/c_p denotes the total diabatic heating, with unit of K/day, indicating the prescribed rate of thermal forcing rather than a cumulative temperature decrease.

For instant, Figure R10 shows the vertical profile of the specific cooling rate (K day⁻¹) in the cooling center for Exp_1. In the LBM experiment, we prescribed the idealized cooling in tropical central Africa with a vertical profile following a gamma function, peaking at a cooling rate of 6 to 7 K day⁻¹ at about 450 hPa (Figure R10).

Figure R10. Vertical profile of the specific cooling rate (K day^{-1}) in the cooling center for Exp_1.

Reference:

Knutson, T.R. and Manabe, S.. Time-mean response over the tropical Pacific to increased CO_2 in a coupled ocean-atmosphere model. Journal of Climate, 8(9), pp.2181-2199 (1995).

Watanabe, M. and Kimoto, M.. Atmosphere-ocean thermal coupling in the North Atlantic: A positive feedback [Software]. Quarterly Journal of the Royal Meteorological Society, 126, (570), 3343–3369 (2000).

Yanai, M., Esbensen, S. and Chu, J.H.. Determination of bulk properties of tropical cloud clusters from large-scale heat and moisture budgets. Journal of Atmospheric Sciences, 30(4), 611-627 (1973).

Figures:

#16. general: I guess units are always written in rectangular bracket, e.g. [m/s], rather than (m/s).

Our response: Thanks for your reminder. After reviewing other articles in the journal of *Nature Communications*, we observed that the units are presented in parentheses

(•) rather than square brackets [•]. Therefore, we have adhered to using parentheses (•) for units in our manuscript.

#17. Fig.1: Maybe flip the time-axis in the reconstruction plot so that it is in line with the other plots in this figure. Maybe you could additionally mark or explain in the caption which proxies in i) to o) indicate wind and which indicate precipitation.

Our response: Following your valuable, we have reversed the time-axis in the reconstruction plot as follows Figure R11 of the revised manuscript. Additionally, we have labeled the proxies in Figure 1i-o to indicate whether they are wind-based proxies or rainfall-based proxies in the revised manuscript.

Figure R11 Same as Figure 1i-o in the manuscript. **i** Total organic carbon (TOC, wt%) values from ODP Sites 722B, 728B, 730A, and 731A. **j** Planktic foraminifer *Globigerina bulloides* (*G. bulloides*; %) from ODP 722B and 730A. **k** $\delta^{13}\text{C}$ (PDB) of

paleosol carbonate (‰) from the Potwar Plateau and NW India. **l** $\delta^{18}\text{O}$ (PDB) of soil carbonate (‰) from the Potwar Plateau. **m** Chemical weathering proxies (CIA, %) from the Indus Marine A-1 well in the northern Arabian Sea. **n** Kaolinite/chlorite ratio from IODP Site U1447 in the western Andaman Sea and K/Al ratio from ODP Site 718 in the Bengal fan. **o** Total sediment flux into the Indus fan ($10^3 \text{ km}^3 \text{ Ma}^{-1}$). The vertical shaded areas in **i-o** represent the time intervals of Middle Miocene Climate Optimum (MMCO, ~15 Ma) and Late Miocene (~5 Ma).

#18. Fig. 1: Have the precipitation anomalies been tested for significance?

Our response: Thanks for your carefully reading and reminder. Following your valuable comment, we have replotted and tested all figures in the revised manuscript and Supplementary materials using Student's *t*-test. In these figures, the gray stippling and vectors are only shown at regions where changes are significant at the 95% confidence level.

#19. Fig. 1: Why do you use 700hPa level and in other plots 1000hPa or 925hPa or 850hPa? Please check if you use this level for each plot referring to the Somali Jet to facilitate the comparison.

Our response: Thanks for your comments. The reasons for using 700-hPa wind anomaly were discussed in Question 2 in detail. According to your valuable comment, we have further checked the levels of 1000-hPa to 850-hPa winds in Figure R12. As expected, in all levels of 850-hPa, 925-hPa, and 1000-hPa, there is the significant weakening of the core of Somali Jet along the East Africa with the lowering of the East African Highlands during the Miocene. Therefore, the results are consistent with each other when using different levels at the low-level troposphere.

Figure R12. Changes in atmospheric circulation due to African topography changes during summer. Changes in wind speed (shading, m s^{-1}) and horizontal wind (vectors, m s^{-1}) at 850 hPa in the **a** MT25, **b** MT15 and **c** MT05 simulations relative to the pre-industrial simulations. **d-f** Same as **a-c**, but for the 925-hPa winds. **g-i** Same as **a-c**, but for the 1000-hPa winds. The black solid boxes mark the SASM region (10° - 30° N, 60° - 95° E). The blue solid boxes (10° S- 10° N, 40° - 50° E) mark the Somali Jet.

#20. Fig.4: The unit for e) is missing... Is e) also showing the difference between MC and MT simulations? It would also be helpful to plot the area mean changes in the Somali Jet, similar to Fig.1

Our response: Thanks for your comments. We have added the unit for Figure 4e in the revised manuscript.

You are right, Figure 4e in the manuscript shows the changes in ratios of summer rainfall over SASM region mean the difference between MC and MT simulations.

Following your valuable suggestions, we have included the area averaged changes in the Somali Jet as Figure R13e in the revised manuscript. Corresponding description is clarified in Lines 311-314 and 331-336 of the revised manuscript.

Figure R13 Responses of atmospheric circulation to increased CO₂ concentration during boreal summer. **a** Multi-proxy atmospheric CO₂ levels (p.p.m.v) compiled from previous literature, including stomata (black circles), $\delta^{13}\text{C}$ of alkenones (red triangles) and marine boron (black crosses). Changes in rainfall (shading, mm day⁻¹) and 700-hPa wind (vectors, m s⁻¹) in the **b** MC25, **c** MC15 and **d** MC05 simulations

relative to the MT25, MT15 and MT05 simulations, respectively. In **b-d**, gray stippling and vectors denote regions in which the changes are significant at the 95% confidence level according to Student's *t*-test, and the solid boxes mark the SASM region. **e** Area-averaged changes in Somali Jet intensity (m s^{-1} ; see Methods) and ratios of summer rainfall (%) over the SASM region due to increased CO_2 concentration in **b-d**, with asterisks above the bars denoting changes that are significant at the 95% confidence level according to Student's *t*-test.

#21. Fig.5a: I'm wondering, if the cyclonic circulation arrows have the wrong position. The winds are directed to the West Ghats and do not flow in western direction over India (cf. Fig.1). Similarly, the arrow for suppressed convection is located over the African Highlands but not over the Congo basin...

Our response: We apologize for the wrong position of the arrows. According to your suggestion, we have replotted these two figures in the revised manuscript.

Supplement:

#22. General: Please check the spelling in all figure captions.

Our response: Thank you for your reminder. We have checked and revised the spelling in all figure captions in the revised Supplementary materials.

#23. Fig.1: Here, 850hPa winds are shown, why not consistent to the other plots (700hPa). Why are wind vectors shown in the simulation but not in the observation? They experience the same orography...

Our response: Thanks for the comments. We have discussed this in detail in Question 2. The primary reason is that the 850-hPa wind in low-level troposphere is significantly influenced by the topographical changes, which can subsequently affect the cross-equatorial Somali Jet and upwelling in the western Arabian Sea.

We apologize for this mistake, and we have replotted the Supplementary Figure 1 as follows in Figure R14a and b.

Figure R14. Model validation. **a** Observed climatological mean rainfall (mm day⁻¹) and 850-hPa wind (m s⁻¹), and **c** surface wind (vectors, m s⁻¹) and its induced Ekman pumping (cm day⁻¹) during boreal summer from 1980 to 2010. **b** and **d** are the same as **a** and **c**, but for the results of pre-industrial simulation from EC-Earth2 model. Monthly rainfall in **a** is from the Global Precipitation Climatology Project (GPCP), and 850-hPa and surface horizontal winds are from the National Centers for Environmental Prediction reanalysis 2 (NCEP2). In **a** and **b**, red dots or boxes denote the locations of the proxy records in Figure 1i-o, black boxes mark the SASM region (60° E-95° E, 5° N-30° N), and blue boxes (10° S-10° N, 40° -50° E) mark the cross-equatorial Somali Jet. In **c** and **d**, gray circles in western Arabian Sea denote the locations of the proxy records in Figure 1i and j (same as in **a** and **b**), gray boxes (15° -21° N, 54° -62° E) denote the western Arabian Sea upwelling region that proxy records located, and blue boxes mark the cross-equatorial Somali Jet.

#24. Fig 2: Please add a note that only African changes in topography are shown here.

Our response: Following your valuable suggestion, we have added this in the revised manuscript.

#25. Fig.4: Which levels were used to integrate the moisture flux?

Our response: In this study, the vertically integrated moisture transport is from surface to 10 hPa in our study, including the levels of 1000, 925, 850, 700, 500, 400, 300, 200, 100, 50, and 10 hPa. This is clarified in Supplementary Figure 5 of the revised Supplementary materials.

#26. Fig.6: Is K/day the correct unit for Q_L ?

Our response: We apologize for the wrong unit for Q_L , and the unit should be $m^2 s^{-3}$. We have corrected the units of Q_L in the revised manuscript, and checked the whole manuscript to avoid this mistake.

#27. Fig.8: Which levels have been used for the integration of the water vapor flux?

Do the plots b-c really show anomalies to MT? This would mean that the CO_2 has an even stronger dynamic response than seen for the topography change.

Our response: Thanks for the comment. We use the levels from surface to 10 hPa for the integration of the water vapor flux in our study, including the levels of 1000, 925, 850, 700, 500, 400, 300, 200, 100, 50, and 10 hPa. This is clarified in the Supplementary Figure 11 of the revised Supplementary materials.

Yes, Supplementary Figure 11b-d of the Supplementary materials illustrate anomalies between the MC and MT simulations, reflecting the impact of CO_2 changes. However, these anomalies represent the response to increased CO_2 alone, not the relative contribution of CO_2 and African topography. As clarified in the Methods section, the MC simulations (MC25, MC15 and MC05) incorporate both CO_2 changes (500 ppm) and the corresponding African topography for each Miocene time slice. The climate response to CO_2 forcing alone is determined by the comparing MC25

with MT25, MC15 with MT15, and MC05 with MT05 simulations. Following your comment, we further examined the relative contribution of dynamic effect on SASM rainfall changes induced by African topography changes and elevated CO₂ concentration in Figure R15. It is evident that African topography changes have a significantly stronger dynamic effect on SASM rainfall than CO₂ concentration changes, supporting our previous findings in the manuscript.

Figure R15. The dynamic effect (see Methods in detail) averaged over the SASM region. Blue and red bars mean the dynamic effect induced by African topography changes and elevated CO₂ concentration, respectively. Units: Pa kg m⁻² s⁻¹.

#28. Fig.9: This plot also shows the 850hPa while other plots use 700 hPa. It would be easier to compare the figures and the effect of the different forcings if the levels were standardized. Also for this plot it would be nice to add the area-mean change in the Somali-Jet intensity (cf. Bar plot in Fig.1)

Our response: Following your valuable comments, we have used 700-hPa winds in the figures and also added the area-mean changes in the Somali Jet intensity in the revised Supplementary Figure 12 as follows in Figure R16. This is clarified in Lines 331-336 in the revised manuscript.

Figure R16. Changes in SASM rainfall and atmospheric circulation due to combined effects of African topography changes and elevated CO₂ levels. Changes in rainfall (shading, mm day⁻¹) and 850-hPa wind (vectors, m s⁻¹) in the **a** MC25, **b** MC15 and **c** MC05 simulations relative to the pre-industrial simulation. **d** Area-averaged changes in Somali Jet intensity (m s⁻¹; see Methods) and ratios of summer rainfall (%) over the SASM region in **a-c** due to combined effect of African topography and CO₂ forcings, with asterisks above the bars denoting changes that are significant at the 95% confidence level according to Student's *t*-test. In **a-c**, gray stippling and vectors denote regions in which the changes are significant at the 95% confidence level according to Student's *t*-test.

#29. Fig.9: The headings of the sub-figures are not correct.

Our response: Thanks for the comment. After checking the headings of the sub-figures carefully, we found it is correct. As clarified in the Methods section, MC25, MC15 and MC05 are the simulations with the CO₂ concentrations set at 500 p.p.m.v and African topography configurations corresponding to the Early, Middle and Late Miocene, respectively. Therefore, the climate response to total forcing, i.e., combined

effects of African topography changes and elevated CO₂ concentrations, is denoted by the changes in MC25, MC15 and MC05 simulations relative to pre-industrial. This is clarified in Line 412 of the revised manuscript.

Reviewer #2 (Remarks on code availability):

The code is not available, only upon request.

Our response: Thanks for your reminder. We have provided the code for the main results in the new submission on Zenodo at <https://doi.org/10.5281/zenodo.14905875>.

Reply to Reviewer #3's comments

Remarks to the Author:

Recent studies have shown that monsoon winds in the Arabian Sea and the South Asian monsoon are positively correlated on interannual timescales but not necessarily on longer orbital timescales. In this study, Z. Han et al. extend this concept, suggesting a similar decoupling over much longer tectonic timescales, supported by proxy evidence. They propose a mechanism based on climate model experiments, linking the uplift of African orography during the middle to late Miocene to the South Asian Summer Monsoon (SASM). According to their model, the less elevated East African orography induces anomalous latent heating in the atmosphere, triggering circulation changes via Gill's response. These circulation shifts alter low-level monsoon winds, with the resulting SST anomalies (resembling a positive IOD) further amplifying both monsoon winds and rainfall.

Initially, I was very interested in the study and looked forward to reading the manuscript. However, the results seem to suggest quite the opposite of the authors' primary claim—that monsoon winds and rainfall are, in fact, decoupled on these tectonic timescales. There are three key issues:

1. The regions chosen to represent monsoon winds do not overlap with the proxy regions, and in the proxy regions, the changes in winds shown by the climate model experiments are not robust.

2. Their moisture budget analysis indicates that SASM rainfall changes are strongly linked to the dynamic component (i.e., monsoon winds), directly contradicting their central hypothesis of decoupling.

3. The authors interpret upwelling proxies as being linked to monsoon wind speed. However, it is well known that changes in wind speed, the latitude, or width of the low-level jet (LLJ) can all influence upwelling. The correct interpretation should be that upwelling is not coupled with rainfall.

Due to these issues, I do not find the author's primary claim convincing. However, I find their detailed mechanism linking African orography to SASM rainfall quite interesting and could offer an alternative focus for the study. Unfortunately, this

connection between African orography and SASM rainfall is already well-established in the literature. Moreover, the connection between changes in African rainfall and its impact on upwelling in the northern Arabian Sea and SASM rainfall has been demonstrated in previous studies. While the authors' mechanism is well-constructed, it lacks the novelty required for publication in a high-impact journal like Nature Communications.

For these reasons, I recommend rejecting the manuscript in its current form. I encourage the authors to refine their focus on the proposed mechanism, enhance the rigor of their experiments. The current experiments need refinement to better align with their claims, and addressing these issues will significantly improve the paper's overall impact. My major and minor concerns, outlined below, should provide helpful guidance for their revisions.

Our response: We would like to thank you for professional review work, constructive comments, and valuable suggestions on our manuscript. Your time and efforts are greatly appreciated. We have strived to improve the manuscript accordingly as listed in details below, and hope that you will find the added information suitable and sufficient for publication. The original comments are quoted in blue and the changes in the revised manuscript have been highlighted in red.

General Comments:

#1. **Choice of regions for analysis:** (1) The authors use proxy data to demonstrate that upwelling in the northern Arabian Sea and SASM rainfall were not coupled during the middle to late Miocene monsoon evolution. However, in their model analysis, they selected a study region significantly south of the proxy region. Analyzing this area in isolation leads them to conclude that the low-level jet (LLJ) wind speed and monsoon rainfall are not coupled. However, they overlook an important detail: the LLJ, and consequently the monsoon winds, show a significant increase in speed in the region immediately west of their chosen area. This stronger LLJ impacts the western coast of India, producing substantial rainfall over the Western Ghats and a slight increase in rainfall across mainland India. Therefore, the authors' claim that the LLJ/monsoon winds and SASM rainfall are decoupled is not

fully supported. In fact, a more careful examination of the wind changes in the proxy region reveals a much more complex response, which the authors have not addressed in their analysis.

(2) Furthermore, I find the authors' choice of region for defining the SASM problematic. The precipitation response plots indicate a pronounced change in rainfall along the western coast, extending into the adjacent Arabian Sea, while most of inland India—the core SASM region—shows insignificant changes in rainfall. The rainfall signal used by the authors to support their claim is predominantly a result of offshore precipitation along the western coast of India, which is a direct consequence of changes in the LLJ/monsoon winds.

Thus, I find the authors' conclusion in the model analysis unconvincing and sensitive to their choice of region. Their claim appears somewhat overstated and does not fully align with the complexity of the wind-rainfall relationship seen in the data.

Our response: Thank you for these important observations. We agree that a more detailed analysis of Somali Jet dynamics and its regional impact on rainfall is essential to strengthen our conclusions. Below, we address both points in detail.

(1) Our further analysis confirms that the cross-equatorial Somali Jet is significantly weakened in the tropical western Indian Ocean, while winds along the western Arabian Sea are strengthened in the MT25 and MT15 simulations compared to pre-industrial simulation (Figure R1). This south weakened-north strengthened wind pattern exactly leads to weakened Arabian Sea upwelling (Please see Question 3 for more details), particularly along the proxy sites (Figure R2).

The weakened Somali Jet results in reduced upwelling along the western Arabian Sea coast, in agreement with wind-proxy records (Figure 1i and j in the manuscript). This upwelling response is not simply a function of wind intensity but is also driven by structural changes in the Somali Jet, support the idea that the proxy-based interpretation must account for both effects (Please see Question 3 for more details).

Recent work by Sarr et al. (2022) supports our findings, showing that the uplift of the East African topography played a pivotal role in the establishment of the

modern Somali Jet structure and upwelling dynamics from the Early Miocene (~20Ma) to the Late Miocene (~10Ma).

In summary, the further considering of wind structure changes and ocean upwelling in western Arabian Sea can't change the conclusions we have drawn from our original results, confirming the robustness of our primary findings. In the revised manuscript, we have expanded the discussion of Somali Jet structural changes and their impact on upwelling in Lines 121-132, 146-151, 442-445, 458-461, and 499-505. Additionally, we have included Figure R1 and R2 as Supplementary Figure 3 and 4 in the revised Supplementary materials, and Figure R3 as Figure 1g and 1h in the revised manuscript. Thank you again for guiding us toward a more robust and detailed analysis.

(2) We have adjusted the color bar range and performed student's *t*-test on rainfall and wind data to enhance the visibility of significant rainfall changes over the inland Indian (Figure R4a-c). These updated figures show statistically significant increases in inland SASM rainfall, particularly in Early and Middle Miocene simulations.

To further confirm these rainfall anomalies, we examined the changes in outgoing longwave radiation (OLR) (Figure R4d-f). The significant decrease in OLR over inland India confirms that deep convection and enhance rainfall occurred in these regions during the Early and Middle Miocene. We separately calculated the changes in SASM rainfall over ocean and land (Figure R5) to determine the spatial distribution of precipitation anomalies. Our results confirm that both land and oceanic SASM rainfall increased during the Early and Middle Miocene, coinciding with the period of lower East African Highlands.

Figure R1. Changes in atmospheric circulation due to African topography changes during boreal summer. Climatological mean 850-hPa winds (vectors, m s^{-1}) and its wind speed (shading, m s^{-1}) in the **a** pre-industrial, **b** MT25, **c** MT15 and **d** MT05 simulations. Responses to Miocene African topography changes in **e** MT25, **f** MT15 and **g** MT05 simulations compared to the pre-industrial simulation. Black boxes mark the SASM region, and blue boxes mark the cross-equatorial Somali Jet. In **e-g**, gray stippling denotes regions in which the changes are significant at the 95% confidence level according to Student's t -test.

Figure R2. Responses of ocean upwelling over western Arabian Sea to African topography changes during boreal summer. Changes in vertical velocity of surface ocean (0-100m average, cm day^{-1}) in the **a** MT25, **b** MT15 and **c** MT05 simulations compared to pre-industrial simulations. The gray circles in western Arabian Sea denote the locations of the proxy records in Figure 1i and j in the manuscript. The black boxes denote the western Arabian Sea upwelling region that proxy records located. Gray stippling denotes regions in which the changes are significant at the 95% confidence level according to Student's *t*-test.

Figure R3. Response of simulated SASM to African topography changes during the Miocene in the coupled EC-Earth3 experiments. The changes in **a** intensity of Somali Jet (m s^{-1}) and western Arabian Sea upwelling (cm day^{-1}), and **b** ratios of SASM rainfall (%), with asterisks above or below the bars denoting changes that are significant at the 95% confidence level according to Student's *t*-test. The solid blue boxes ($15^{\circ}\text{ S}-10^{\circ}\text{ N}$, $37.5^{\circ}-62.5^{\circ}\text{ E}$) and solid purple boxes ($15^{\circ}-21^{\circ}\text{ N}$, $54^{\circ}-62^{\circ}\text{ E}$) in Figure R1 are used to define the intensity of Somali Jet and western Arabian Sea upwelling, respectively.

Figure R4. Responses of SASM to African topography changes during boreal summer. Changes in rainfall (shading, mm day⁻¹) low-level atmospheric circulation at 700 hPa (vectors, m s⁻¹) in the **a** MT25, **b** MT15 and **c** MT05 simulations compared to pre-industrial simulations. **d-f** Same as **a-c**, but for the changes in outgoing longwave radiation (OLR, W m⁻¹). Gray stippling and vectors are only shown at regions where the changes are significant at the 95% confidence level according to Student's *t*-test.

Figure R5. Response of simulated SASM rainfall (mm day⁻¹) to African topography changes during the Miocene, with asterisks below the bars denoting changes that are significant at the 95% confidence level according to Student's *t*-test.

#2. Moisture budget analysis: My concerns are further heightened by the authors' moisture budget analysis. Their results clearly indicate that changes in precipitation over the SASM region are driven by the dynamic component (i.e., changes in monsoon winds). This presents a significant contradiction: on one hand, the authors claim that monsoon rainfall and winds are decoupled, yet their own analysis shows that the changes in rainfall are directly linked to wind dynamics. This inconsistency undermines their central argument and raises questions about the validity of their conclusions.

Our response: We appreciate this insightful comment, as it allows us to further clarify the concept of monsoon wind-rainfall decoupling in the context of tectonic-scale changes.

Decoupling does not imply a complete lack of relationship between monsoon winds and rainfall but rather signifies a nonlinear and regionally varying response of monsoonal precipitation to changes in Somali Jet intensity. In present-day climate, there is a strong correlation between Somali Jet and SASM rainfall, as the jet directly controls moisture transport (Chakraborty et al., 2009; Chen et al., 2023). However, paleo-proxy records suggest that this relationship did not hold consistently over tectonic timescales (Sarr et al., 2022). Wind-based proxies (e.g., Kroon et al., 1991; Huang et al., 2007; Gupta et al., 2015) indicate that the Somali Jet strengthened after ~13 Ma. Rainfall-related proxies indicate the increased rainfall during the Middle Miocene (~15 Ma) (Quade et al., 1989; Quade and Cerling, 1995; Clift and Webb, 2019; Lee et al., 2020) despite weaker Somali Jet intensity. Therefore, this apparent contradiction is what led to the hypothesis of wind-rainfall decoupling during the Miocene.

The decoupling refers to the reversal of the expected relationship—instead of a stronger Somali Jet leading to increased rainfall, our simulations show that a weakened Somali Jet can still lead to enhanced SASM rainfall via indirect ocean-atmosphere feedbacks.

The dynamic term in the moisture budget analysis captures the effect of large-scale circulation changes on rainfall. Our results indicate that changes in African

topography trigger large-scale atmospheric circulation anomalies, which in turn enhance moisture transport and increase SASM rainfall. This dynamic response does not contradict the decoupling hypothesis but rather provides a mechanistic explanation for why rainfall increased despite a weaker Somali Jet. The key process involves ocean-atmosphere interactions: A weakened Somali Jet leads to a positive Indian Ocean Dipole (IOD)-like warming pattern, which weakens the Indian Ocean Walker circulation and enhances the moisture supply to the SASM region. This process compensates for the reduction in moisture transport due to a weaker Somali Jet, ultimately increasing rainfall in the SASM region.

Our results are supported by recent study of Sarr et al. (2022), which highlight that the East African topography played a pivotal role in the establishing the modern Somali Jet structure above the western Indian Ocean during the Miocene.

In the revised manuscript, we expanded discussion on the role of dynamic effects in rainfall changes and clarified how moisture budget analysis supports the decoupling hypothesis. We highlighted the indirect ocean-atmosphere feedbacks that drive increased SASM rainfall despite a weaker Somali Jet (Lines 185-189 and 192-195).

Reference:

Chakraborty, A., Nanjundiah, R. S. and Srinivasan, J. Impact of African orography and the Indian summer monsoon on the low-level Somali jet. Int. J. Climatol. 29, 983–992 (2009).

Chen, S. et al. Impact of interannual variation of the spring Somali Jet intensity on the northwest–southeast movement of the South Asian High in the following summer. Clim. Dyn. 60, 1583–1598 (2023).

Clift, P. D. and Webb, A. A. G. A history of the Asian monsoon and its interactions with solid Earth tectonics in Cenozoic South Asia. SP 483, 631–652 (2019).

Gupta, A. K., Yuvaraja, A., Prakasam, M., Clemens, S. C. and Velu, A. Evolution of the South Asian monsoon wind system since the late Middle Miocene. *Palaeogeogr. Palaeoclimatol. Palaeoecol.* 438, 160–167 (2015).

Huang, Y., Clemens, S. C., Liu, W., Wang, Y. and Prell, W. L. Large-scale hydrological change drove the late Miocene C4 plant expansion in the Himalayan foreland and Arabian Peninsula. *Geology* 35, 531 (2007).

Kroon, D., Steens, T. and Troelstra, S. R. Onset of monsoonal related upwelling in the western Arabian sea as revealed by planktonic foraminifers 1. In *Proceedings of the ocean drilling program, scientific results 1*, 257–263 (1991).

Lee, J. et al. Monsoon-influenced variation of clay mineral compositions and detrital Nd-Sr isotopes in the western Andaman Sea (IODP Site U1447) since the late Miocene. *Palaeogeogr. Palaeoclimatol. Palaeoecol.* 538, 109339 (2020).

Quade, J., Cerling, T. E. and Bowman, J. R. Development of Asian monsoon revealed by marked ecological shift during the latest Miocene in northern Pakistan. *Nature* 342, 163–166 (1989).

Quade, J. and Cerling, T. E. Expansion of C4 grasses in the Late Miocene of Northern Pakistan: evidence from stable isotopes in paleosols. *Palaeogeogr. Palaeoclimatol., Palaeoecol.* 115, 91–116 (1995).

Turner, A. G. & Annamalai, H. Climate change and the South Asian summer monsoon. *Nat. Clim. Change* 2, 587–595 (2012).

Sarr, A.C., Donnadieu, Y., Bolton, C.T., Ladant, J.B., Licht, A., Fluteau, F., Laugié, M., Tardif, D. and Dupont-Nivet, G.. Neogene South Asian monsoon rainfall and wind histories diverged due to topographic effects. *Nature Geoscience*, 15(4), 314-319 (2022).

#3. Interpretation of upwelling proxies:

(1) In this study, the authors claim that monsoon winds and rainfall are decoupled, based on proxy records. However, the correct interpretation should be that upwelling in the northern Arabian Sea and rainfall are decoupled. Upwelling in this region is influenced by various factors, including wind speed, the latitude, and width of the low-level jet (LLJ). Even slight changes to

one or more of these LLJ characteristics can modulate upwelling and, by extension, rainfall (see recent studies on impact of orbital forcing on upwelling and rainfall for example). Therefore, the primary claim of this study—that monsoon winds and rainfall are decoupled—seems questionable. The relationship between winds, upwelling, and rainfall is more complex than the authors suggest, and their conclusions may oversimplify these dynamics.

(2) This is further supported by their figures of SST anomalies. The SST anomalies during the middle to late Miocene are negative in the northern Arabian Sea, indicating enhanced upwelling. This region, which spans the coast of Yemen and Oman and extends into the open sea, encompasses most of the proxy locations. This again indicates that upwelling-rainfall are decoupled not necessarily monsoon winds and rainfall.

Our response: These are really good and important questions that we have not considered in our previous manuscript version. Indeed, the relationship between winds, upwelling, and rainfall is more complex than we initially thought. Following your comments, we further analyzed the western Arabian Sea upwelling and calculated the Ekman pumping to evaluate the complex relationship. Additionally, we examined the sea surface temperature (SST) anomalies in the Arabian Sea by calculating the heat budget equation for the upper-ocean mixed layer. Our further analysis confirms that SASM wind-induced upwelling and rainfall are indeed decoupled during the Miocene, supporting our previous results.

(1) Indeed, upwelling in the western Arabian Sea is more complicated than initially thought, which is controlled by changes in both the strength and the patterns of Somali Jet (Le Mézo et al., 2017; Jalihal et al., 2022; Wen et al., 2024). These changes in monsoon winds can lead to wind-stress curl anomaly, driving the ocean upwelling through “Ekman pumping” (Rykaczewski and Checkley, 2008; Wen et al., 2024). To check these processes caused by the Miocene African topography changes, we further use monthly wind stress to calculate the Ekman pumping velocity as follows:

$$W_{Ekman} = \frac{1}{\rho_w} \nabla \times \left(\frac{\tau}{f} \right) \approx \frac{1}{\rho_w f} \nabla \times \tau \quad (1)$$

where ρ_w is seawater density (1024 kg m^{-3}), f is the Coriolis parameter, and $\nabla \times \tau$ is the wind-stress curl.

As expected, our model successfully replicates the present-day SASM system, characterized by a strong cross-equatorial Somali Jet and increased rainfall over the Indian subcontinent during the boreal summer (Figure R6a and b). Due to the Coriolis force, the climatological southwesterly monsoon wind over the Arabian Sea drive strong ocean upwelling along the coastal region during summer (Figure R6c and d). The agreement between our pre-industrial simulation and contemporary observations validates the use of EC-Earth3 for exploring the influence of the African topography on the SASM.

We agree to you that our claim should not be simply interpreted as "monsoon winds and rainfall are decoupled", but rather that the structure of the Somali Jet has changed, leading to regionally complex upwelling responses, which in turn affected SASM rainfall. Our further analysis found that the African topography changes can lead to a remarkably weakened cross-equatorial Somali Jet while strengthened winds alongshore the Arabian Peninsula at low-level troposphere, particularly in the Early and Middle Miocene (Figure R1e and f). This south weakened-north strengthened wind patterns decrease the wind-stress curl alongshore the Arabian Peninsula ($\nabla \times \tau < 0$), leading to negative Ekman pumping and weaker upwelling there in the Early and Middle Miocene (Figure R3 and R7). Despite weaker Somali Jet intensity overall, monsoon rainfall increases due to ocean-atmosphere feedbacks, explaining the decoupling observed in proxies (Figure 1i and j in the manuscript).

Proxy records, i.e., Total organic carbon (TOC) values and Planktic foraminifer *Globigerina bulloides* (*G. bulloides*), indicate weakened upwelling during the Middle Miocene (Figure 1i and j in the manuscript), aligning with our model results (Figure R3 and R7). That is, the changes in Somali Jet induced by African topography changes alter the wind-stress curl, reducing upwelling in the western Arabian Sea during the Miocene (Figure R7). Additionally, TOC values and *G. bulloides* are widely used to estimate the SASM wind intensity (Huang et al., 2007; Gupta et al., 2015), as they predominantly reflect wind-driven oceanic upwelling and primary

production in the western Arabian Sea. Our further analysis suggests that the wind-based proxies, which is reflected the Arabian Sea upwelling changes, should be carefully considered, particularly on geological timescales, when used as an indicator of the changes in the intensity of SASM.

Therefore, the further considering of wind structure changes and upwelling in western Arabian Sea can't change the conclusions we have drawn from our original results, confirming the robustness of our primary findings. In the revised manuscript, we clarified that our claim is about the structural change of the Somali Jet and its impact on regional upwelling, not a simple decoupling of monsoon winds and rainfall. We expanded discussions on the complexities of Somali Jet variability and upwelling (Lines 31, 104-106, 121-132, 442-445, 458-461, and 499-505), and provided a more comprehensive and coherent analysis. Additionally, Figures R1, R6, and R7 has been added as Supplementary Figures 3, 1 and 4 in the revised Supplementary materials.

(2) Regarding the question about negative SST anomaly in western Arabian Sea where the upwelling is weakened.

SSTs are governed by both atmospheric and oceanic processes. On the atmospheric side, wind speed, air temperature, cloudiness, and humidity are the dominant factors regulating the exchange of energy at the sea surface. On the oceanic side, heat transport by currents, vertical mixing, and boundary layer depth influence SST. To explore the mechanism of negative SST anomaly in the western Arabian Sea, we further calculate the heat budget equation as follows.

Mathematically, the heat budget for the upper-ocean mixed layer may be written as:

$$\frac{\partial T}{\partial t} = \frac{Q_{net}}{\rho C_p H} - \vec{V} \cdot \nabla T - \omega \frac{\partial T}{\partial z} + Res \quad (2)$$

where T is the mixed layer temperature, Q_{net} is the net surface energy flux, ρ is the density of seawater, C_p is the specific heat of seawater, H is the mixed layer depth, \vec{V} is the horizontal current, and ω is the vertical velocity. Q_{net} is defined as:

$$Q_{net} = Q_{sb} + Q_{lb} + Q_{sw} + Q_{tw} \quad (3)$$

where Q_{sb} is the sensible heat flux, Q_{lb} is the latent heat flux, Q_{sw} is the net shortwave radiation at surface, Q_{lw} is the net longwave radiation at surface.

The terms on the right-hand side of Eq. (2) are net surface heat flux, horizontal temperature advection, vertical temperature advection, and residual terms. In climatology, there is merely a balance between the terms on the right hand of Eq. (2) as the temporal variation can be ignored (Deser et al., 2010; Schneider et al., 2012; Long et al., 2014; Wen et al., 2018):

$$0 = \frac{Q_{net}}{\rho C_p H} - \bar{V} \cdot \nabla T - \omega \frac{\partial T}{\partial z} + Res \quad (4)$$

According to Eq. (4), our further heat budget analysis confirms that the simulated negative SST anomalies are primarily driven by horizontal advection rather than upwelling anomalies (Figure R8). The weakened upwelling should theoretically cause warming (Figure R8g-i), but horizontal advection compensates (Figure R8d-f), leading to net cooling in the western Arabian Sea (Figure 2b and e in the revised manuscript). Thus, negative SST anomalies in the Arabian Sea do not indicate enhanced upwelling. Instead, they result from changes in ocean heat transport and large-scale circulation.

Therefore, the simulated negative SST anomalies during the Middle to Late Miocene in the western Arabian Sea are primarily due to horizontal temperature advection anomalies rather than upwelling anomalies. The negative SST anomalies do not contradict our findings of the weakened upwelling during the Early and Middle Miocene.

Figure R6. Model validation. **a** Observed climatological mean rainfall (mm day^{-1}) and 850-hPa wind (m s^{-1}), and **c** surface wind (vectors, m s^{-1}) and its induced Ekman pumping (cm day^{-1}) during boreal summer from 1980 to 2010. **b** and **d** are the same as **a** and **c**, but for the results of pre-industrial simulation from EC-Earth3 model. Monthly rainfall in **a** is from the Global Precipitation Climatology Project (GPCP), and 850-hPa and surface horizontal winds are from the National Centers for Environmental Prediction reanalysis 2 (NCEP2). In **a** and **b**, red dots or red boxes denote the locations of the proxy records in Figure 1i-o in manuscript, and black boxes ($5^{\circ} \text{ N}-30^{\circ} \text{ N}$; $60^{\circ} \text{ E}-95^{\circ} \text{ E}$) mark the SASM region. In **c** and **d**, gray circles in western Arabian Sea denote the locations of the proxy records (same as in Figure R1a and b). In **a-d**, blue boxes ($10^{\circ} \text{ S}-10^{\circ} \text{ N}$, $40^{\circ} -50^{\circ} \text{ E}$) mark the cross-equatorial Somali Jet and purple boxes ($15^{\circ} -21^{\circ} \text{ N}$, $54^{\circ} -62^{\circ} \text{ E}$) mark the western Arabian Sea upwelling region that proxy records located.

Figure R7. Responses of ocean upwelling over western Arabian Sea to African topography changes during boreal summer. Changes in vertical velocity of surface ocean (0-100m average, cm day^{-1}) in the **a** MT25, **b** MT15 and **c** MT05 simulations compared to pre-industrial simulations. **d-f** Same as **a-c**, but for the changes in Ekman pumping (see Methods, cm day^{-1}). The gray circles in western Arabian Sea denote the locations of the proxy records in Figure 1*i* and *j*. The black boxes (same as in Figure R1 and R2) denote the western Arabian Sea upwelling region that proxy records mainly located. The gray stippling denotes regions in which the changes are significant at the 95% confidence level according to Student's *t*-test.

Figure R8. Changes in SST equation terms over western Arabian Sea due to African topography changes during boreal summer (calculated by Equation 4). Changes in net surface heat flux in the **a** MT25, **b** MT15 and **c** MT05 simulations compared to pre-industrial simulations. **d-f** Same as **a-c**, but for the changes in horizontal temperature advection. **g-i** Same as **a-c**, but for the changes in vertical temperature advection. **j-l** Same as **a-c**, but for the residual term. The black boxes (same as in Figure R1, R2 and R4) denote the western Arabian Sea upwelling region that proxy records located. Units: $^{\circ}\text{C month}^{-1}$.

Reference:

Deser, C., Alexander, M.A., Xie, S.P. and Phillips, A.S., 2010. Sea surface temperature variability: Patterns and mechanisms. *Annual review of marine science*, 2(1), pp.115-143 (2010).

Gupta, A. K., Yuvaraja, A., Prakasam, M., Clemens, S. C. and Velu, A. Evolution of the South Asian monsoon wind system since the late Middle Miocene. *Palaeogeogr. Palaeoclimatol. Palaeoecol.* **438**, 160–167 (2015).

Huang, Y., Clemens, S. C., Liu, W., Wang, Y. & Prell, W. L. Large-scale hydrological change drove the late Miocene C4 plant expansion in the Himalayan foreland and Arabian Peninsula. *Geology* **35**, 531 (2007).

Jalihai, C., Srinivasan, J. and Chakraborty, A. Response of the Low - Level Jet to Precession and Its Implications for Proxies of the Indian Monsoon. *Geophysical Research Letters*, 49(2), p.e2021GL094760 (2022).

Le Mézo, P., Beaufort, L., Bopp, L., Braconnot, P. and Kageyama, M. From monsoon to marine productivity in the Arabian Sea: insights from glacial and interglacial climates. *Climate of the Past*, 13(7), 759-778 (2017).

Rykaczewski, R.R. and Checkley Jr, D.M.. Influence of ocean winds on the pelagic ecosystem in upwelling regions. *Proceedings of the National Academy of Sciences*, 105(6), 1965-1970 (2008).

Schneider, E. K., and Fan, M. Observed decadal north Atlantic tripole SST variability. Part II: Diagnosis of mechanisms. *Journal of the Atmospheric Sciences*, 69 (1): 51-64 (2012).

Wen, Q., Yao, J., Döös, K. and Yang, H., 2018. Decoding hosing and heating effects on global temperature and meridional circulations in a warming climate. *Journal of Climate*, 31(23), 9605-9623 (2018).

Wen, Q., Liu, Z., Liu, J., Clemens, S., Jing, Z., Wang, Y., Lv, G., Yan, M., Ning, L., Yuan, L. and Gao, Y., 2024. Contrasting responses of Indian summer monsoon rainfall and Arabian Sea upwelling to orbital forcing. *Communications Earth & Environment*, 5(1), 409 (2024).

Zhuang, G., Pagani, M. and Zhang, Y. G. Monsoonal upwelling in the western Arabian Sea since the middle Miocene. *Geology* **45**, 655–658 (2017).

#4. Missing experiments: The authors conducted experiments where global orography is set to pre-industrial levels, except over Africa, where it is adjusted to 25 Ma, 15 Ma, and 5 Ma. However, for a well-constructed study, I would first expect results from experiments where global orography is set to match these time periods fully. Following that, the authors should demonstrate that modifying only African orography is sufficient to capture the majority of wind and rainfall responses. This approach would strengthen their case for the impact of African orography—although, I still believe it won't support the claim that monsoon winds and rainfall are decoupled.

Our response: Thank you for raising this important point. Although this is a valuable idea, conducting fully global topography experiments poses a challenge due to the lack of high spatiotemporal resolution paleo-geographic and topography reconstructions. Nonetheless, we have carefully reassessed our approach and findings, and we demonstrate below why our methodology remains robust in addressing the role of African topography in modulating the SASM system.

In this study, the topography reconstruction is based on Moucha and Forte (2011), which provide dynamic topography estimates only for Africa for the Early (25Ma), Middle (15Ma) and Late Miocene (5Ma). In contrast, global orography reconstructions for the Miocene are limited and mainly available for the Middle Miocene (~15Ma) (Frigola et al., 2018; Poblete et al., 2021). Therefore, the lack of the reconstructed global orography in all three time slices that we focused is not feasible to implement consistent global topography experiments for our study.

Sarr et al. (2022) assessed the impact of multiple topography changes, including the Himalayan-Tibetan Plateau and East African Highlands, and found that Himalayan uplift primarily influenced early Miocene rainfall, while East African uplift played a dominant role from the Middle to Late Miocene. Given the significant uplift of the Himalaya-Tibetan Plateau mainly occurred before ~20 Ma (Ding et al., 2022), its impact on monsoon dynamics in our study period (15–5 Ma) is likely secondary compared to African topography changes. These findings suggest that focusing on African topography as the primary boundary forcing is a reasonable and

well-supported approach. Several recent studies examining Miocene climate change have focused on specific topographic regions rather than global orography changes (Burls et al., 2011; Chakraborty et al., 2009; Wei and Bordoni, 2016; Sarr et al., 2022; Zuo et al., 2024). These studies highlight that African topography plays a key role in shaping modern atmospheric circulation, supporting our focus on isolating its effects in our model simulations.

Our results successfully reproduce the well-documented transition from a weakened to a stronger Somali Jet between the Middle and Late Miocene, in agreement with wind-based proxies (Gupta et al., 2015; Betzler et al., 2016). We also capture the increased SASM rainfall during the Middle Miocene, as indicated by rainfall-based proxies (Quade et al., 1989; Clift and Webb, 2019). These findings demonstrate that modifying only African orography is sufficient to explain key features of monsoon evolution and reconcile the proxy records.

We expanded our analysis to examine Arabian Sea upwelling and Ekman pumping, confirming that the changes in the Somali Jet directly modulate ocean dynamics, aligning with wind-proxy reconstructions (Figure R3, Supplementary Figure 4). Our moisture budget analysis (Figure 1) further supports that African topography changes alone induce the observed SASM rainfall changes via dynamic circulation adjustments. We emphasize that decoupling does not mean a complete lack of relationship between monsoon winds and rainfall but rather a shift in their response due to changes in the Somali Jet structure and ocean-atmosphere interactions. Even when we account for potential uncertainties in CO₂ levels, our simulations with elevated CO₂ (MC experiments) do not show the same decoupling, reinforcing the dominant role of African topography.

Although additional fully global topography experiments are valuable, such experiments would require global reconstructions that do not currently exist, they are beyond the scope of our paper. Nonetheless, both previous studies and our further analysis confirm that African topography is the primary control on the SASM evolution during the studied period (15–5 Ma). Our model results are consistent with proxy records, supporting the validity of our approach.

In the revised manuscript, we expanded discussion on the rationale for focusing on African topography (Lines 72-76), and emphasized that our results align with prior studies that similarly focused on African topography without requiring global orography changes (Sarr et al., 2022) (Lines 76-79, 119-120). And we clarified that our further analysis conforms that the decoupling of Somali Jet and rainfall induced African topography changes are robust (Lines 121-132, and 142-151).

Reference:

Bialik, O. M. et al. Monsoons, upwelling, and the deoxygenation of the Northwestern Indian ocean in response to middle to late Miocene global climatic shifts. Paleoceanogr. Paleoclimatol. 35, e2019PA003762 (2020).

Betzler, C. et al. The abrupt onset of the modern South Asian Monsoon winds. Sci. Rep. 6, 29838 (2016).

Burls, N.J., Bradshaw, C.D., De Boer, A.M., Herold, N., Huber, M., Pound, M., Donnadieu, Y., Farnsworth, A., Frigola, A., Gasson, E. and von der Heydt, A.S.. Simulating miocene warmth: insights from an opportunistic multi - model ensemble (MioMIP1). Paleoceanography and Paleoclimatology, 36(5), p.e2020PA004054 (2021).

Clif, P. D. and Webb, A. A. G. History of the Asian monsoon and its interactions with solid earth tectonics in Cenozoic South Asia. Geol. Soc. Lond. Spec. Publ. 483, 875–880 (2019).

Chakraborty, A., Nanjundiah, R. S. and Srinivasan, J. Impact of African orography and the Indian summer monsoon on the low-level Somali jet. Int. J. Clim. 29, 983–992 (2009).

Ding, L., Kapp, P., Cai, F., Garzzone, C.N., Xiong, Z., Wang, H. and Wang, C.. Timing and mechanisms of Tibetan Plateau uplift. Nature Reviews Earth & Environment, 3(10), 652-667 (2022).

Ding, L. et al.. Quantifying the rise of the Himalaya orogen and implications for the South Asian Monsoon. Geology 45, 215–218 (2017).

Frigola, A., Prange, M. and Schulz, M.. *Boundary conditions for the middle Miocene climate transition (MMCT v1. 0). Geoscientific Model Development, 11(4), 1607-1626 (2018).*

Gupta, A. K., Yuvaraja, A., Prakasam, M., Clemens, S. C. & Velu, A. *Evolution of the South Asian monsoon wind system since the late Middle Miocene. Palaeogeogr. Palaeoclimatol. Palaeoecol. 438, 160–167 (2015).*

Huang, Y., Clemens, S. C., Liu, W., Wang, Y. & Prell, W. L. *Large-scale hydrological change drove the late Miocene C4 plant expansion in the Himalayan foreland and Arabian peninsula. Geology 35, 531–534 (2007).*

Moucha, R. and Forte, A. M. *Changes in African topography driven by mantle convection. Nat. Geosci. 4, 707–712 (2011).*

Nigrini, C. *Composition and biostratigraphy of radiolarian assemblages from an area of upwelling (northwestern Arabian Sea, lag 117). Proc. ODP Sci. Results 117, 89–126 (1991).*

Poblete, F., Dupont-Nivet, G., Licht, A., Van Hinsbergen, D.J., Roperch, P., Mihalynuk, M.G., Johnston, S.T., Guillocheau, F., Baby, G., Fluteau, F. and Robin, C.. *Towards interactive global paleogeographic maps, new reconstructions at 60, 40 and 20 Ma. Earth-Science Reviews, 214, 103508 (2021).*

Sarr, A.C., Donnadieu, Y., Bolton, C.T., Ladant, J.B., Licht, A., Fluteau, F., Laugié, M., Tardif, D. and Dupont-Nivet, G.. *Neogene South Asian monsoon rainfall and wind histories diverged due to topographic effects. Nature Geoscience, 15(4), 314-319 (2022).*

Slingo, J., Spencer, H., Hoskins, B., Berrisford, P. & Black, E. *The meteorology of the Western Indian Ocean, and the influence of the East African Highlands. Phil. Trans. R. Soc. A. 363, 25–42 (2005).*

Tomson, J. R. et al. *Tectonic and climatic drivers of Asian monsoon evolution. Nat. Commun. 12, 4022 (2021).*

Wei, H.-H. & Bordoni, S. *On the role of the African topography in the South Asian monsoon. J. Atmos. Sci. 73, 3197–3212 (2016).*

Zuo, M., Sun, Y., Zhao, Y., Ramstein, G., Ding, L. and Zhou, T.. *South Asian summer monsoon enhanced by the uplift of the Iranian Plateau in Middle Miocene. Climate of the Past*, 20(8), 1817-1836 (2024).

Zhuang, G., Pagani, M. and Zhang, Y. G. *Monsoonal upwelling in the western Arabian Sea since the middle Miocene. Geology* 45, 655–658 (2017).

Minor comments:

#1. Evolution of SST in the Indian Ocean: The development of positive SST anomalies in the western equatorial Indian Ocean is a critical component of the authors' proposed mechanism. However, the manuscript does not provide sufficient explanation for why or how these anomalies form. A detailed examination of the seasonal cycle of SST and winds in this region could offer valuable insights into this process. Additionally, understanding the impact of East African orography on the seasonal cycle (pre-monsoon in particular) of the East African monsoon may be key to explaining the emergence of these positive SST anomalies during the monsoon season. Addressing these points would significantly strengthen the authors' argument.

Our response: Following your valuable comments, we examined the seasonal variation of Somali Jet and SST changes over the western Indian Ocean (Figure R9). Our results indicate that the Somali Jet weakens from the pre-monsoon months (April-May), reaching its lowest intensity during boreal summer (June-July-August, JJA) in the Early and Middle Miocene simulations. This weakened Somali Jet reduces wind-driven surface cooling, allowing for rapid warming in the western equatorial Indian Ocean starting from the pre-monsoon season. The warm SST anomalies intensify during JJA, coinciding with the period of strongest monsoon winds. This pattern suggests that the lower East African Highlands contribute to a weaker Somali Jet, which in turn promotes surface ocean warming through reduced heat loss from the ocean to the atmosphere. In the Late Miocene and pre-industrial periods, the higher East African Highlands enhance the Somali Jet, leading to stronger wind-induced ocean cooling and less pronounced SST anomalies. The results confirm that

topography-driven weakening of the Somali Jet is the primary driver of SST changes in the western Indian Ocean.

Moreover, the seasonal variations depicted in Figure R10 further highlight the crucial role of positive SST anomalies in the western Arabian Sea in establishing the positive air-ocean feedbacks. In Figure R10, warm SST anomalies first appear in the western equatorial Indian Ocean by June, concurrent with a significant reduction in Somali Jet intensity. As the monsoon season progresses, the warm SST anomaly intensifies, and easterly wind anomalies propagate eastward across the equatorial Indian Ocean, this results in further cooling of eastern Indian Ocean, enhancing the west-east SST gradient – a pattern characteristic of a positive IOD-like state. The largest SST anomalies occur during the Early and Middle Miocene, when the East African Highlands are much lower, confirming the critical role of African topography in shaping the seasonal evolution of SST in the Indian Ocean.

All above results, indeed, indicate that the development of positive SST anomalies in the western equatorial Indian Ocean is a critical component of establishing positive ocean-atmosphere feedbacks. This is clarified in Lines 261-269 in the revised manuscript, providing a more detailed discussion of the seasonal evolution of SST anomalies and their connection to East African topography. And we have also included the Figure R9 and Figure R10 as Supplementary Figure 9 and 10 in the revised Supplementary materials. We clarified the role of pre-monsoon conditions in setting up SST anomalies, highlighting how changes in East African topography alter the monsoon onset and subsequent air-sea interactions.

Figure R9. Strong coupling of the intensity of Somali Jet and sea surface temperature (SST) over the western Arabian Sea. Seasonal variation of changes in intensity of Somali Jet (m s^{-1} ; see Methods for detail) and SST ($^{\circ}\text{C}$) over tropical western Indian Ocean (10°S - 10°N , 40° - 55°E). The vertical shaded areas represent the boreal summer (June to August).

Figure R10. Seasonal variation. Changes in SST (shading, °C), sea level pressure [contours; hPa; solid (dashed) lines represent the positive (negative) values], and surface wind (vectors, m s^{-1}) from May to September in the MT25 (left column), MT15 (middle column) and MT05 (right column) simulations compared to pre-industrial simulation. Gray stippling and vectors in each panel denote regions in which the changes are significant at the 95% confidence level according to Student's *t*-test.

#2. Line 147: “Fig. 11” it took me sometime to understand that it is not Fig. eleven.
Maybe authors should refer to it in capital letter to avoid confusion β

Our response: Revised.

Reply to Reviewer #1's comments

Remarks to the Author:

Thanks for addressing my comments, I am satisfied with the responses.

Our response: We would like to thank you for professional review work, constructive comments, and valuable suggestions on our manuscript. Your time and efforts are greatly appreciated.

#1. Fig. R1 – I am a bit confused looking at the response at 850 and 700 hPa. The changes to circulation should be somewhat coherent at these levels. At 850 hPa, (Fig. R1 d-f), the changes to circulation seem to be emanating from equatorial Indian Ocean to Western Ghats. This circulation feature changes abruptly at 700 hPa. The changes in rainfall are concentrated over Western Ghats and Northwest India (which receives less rainfall anyway). No changes are seen along the monsoon trough.

Our response: Thanks for the comment. The apparent inconsistency in circulation patterns between 700 hPa and 850 hPa indeed reflects the influence of multiple dynamical processes operating at different vertical levels of the troposphere. At 850 hPa, the circulation anomalies (Figure R1d-f) are more directly affected by surface forcing, particularly the modifications to African topography. These lower-level anomalies exhibit a coherent southwesterly anomaly from the equatorial Indian Ocean toward the Western Ghats, aligning well with near-surface anomalies at 925 hPa (Figure R1 a-c). This reflects the immediate influence of reduced pyrographic blocking by the East African Highlands on the Somali Jet and associated moisture transport pathways.

In contrast, the anomalous cyclonic circulation is more distinct at 700 hPa than at 850 hPa, influenced not only by topographic changes but also by mid-tropospheric heating gradients arising from convection anomalies. Specially, suppressed convection over tropical central Africa reduces latent heat releases (Supplementary Figure 5 and 7), inducing subsidence and initiating equatorial wave responses are shown in the LBM results with imposed cooling over Africa (Supplementary Figure

8), generates a Rossby wave response that modifies the mid-level circulation structure across the Indian Ocean sector.

The discrepancy in circulation coherence between these two levels therefore reflects the vertical structure of the atmospheric response: while the 850 hPa winds reflect terrain-induced mechanical effects, the 700 hPa anomalies are dynamically modulated by remote diabatic heating and associated wave responses.

Regarding the spatial distribution of rainfall, we agree that increased SASM rainfall is concentrated over the Western Ghats and Northwest India – regions where topographic enhancement and convergence of moisture-laden winds are most pronounced. Our results demonstrate that anomalous cyclonic and anticyclonic circulation over the Arabian Sea and Bay of Bengal are key in strengthening moisture convergence into these regions (Figure 1d-f in the revised manuscript and Supplementary Figure 5 in the revised Supplementary materials). Figure R2 further illustrates this convergence structure.

Moreover, our moisture budget analysis confirms that the dominant contribution to increased SASM rainfall during the Miocene arises from dynamically induced circulation changes rather than local thermodynamic (moisture content) effects (Supplementary Figure 6 in the revised Supplementary materials). These results underscore the role of both direct atmospheric processes and indirect ocean-atmosphere feedbacks in driving the observed rainfall response (as synthesized in Figure 5 of the revised manuscript).

Figure R1. Changes in atmospheric circulation due to African topography changes during summer. Changes in wind speed (shading, m s^{-1}) and horizontal wind (vectors, m s^{-1}) at 925 hPa in the **a** MT25, **b** MT15 and **c** MT05 simulations relative to the pre-industrial simulations. **d-f** Same as **a-c**, but for the 850-hPa winds. The black solid boxes mark the SASM region (10° - 30° N, 60° - 95° E). The blue solid boxes (10° S- 10° N, 40° - 50° E) mark the Somali Jet.

Figure R2. Changes in moisture transport due to African topography changes during summer. Changes in vertical integral water vapor flux from 1000 hPa to 10 hPa (vectors, $\text{kg m}^{-1} \text{s}^{-1}$) and its divergence (shading, mm day^{-1}) in the **a** MT25, **b** MT15 and **c** MT05 simulations relative to the pre-industrial simulations. Red bold vectors in **a** and **b** indicate the moisture channel anomalies. In **a-c**, gray stippling and vectors denote regions in which the changes are significant at the 95% confidence level according to Student's t -test, and solid boxes mark the SASM region.

Reply to Reviewer #2's comments

Remarks to Author:

The authors responded very carefully and very extensively to the reviewers' comments, implemented most of the comments and otherwise convincingly explained why they did not take the comments into account. As a result, the manuscript has improved substantially. The structure and general statements are much clearer. Additional analysis and verification of the significance of the results reinforce the statements. I have rarely seen such a carefully prepared reply. My compliments to the authors. I recommend the publication after a few minors, mostly technical revisions.

Our response: Your positive assessment is a tremendous source of motivation for our team. We are delighted that the revised manuscript now meets with your approval. We have addressed your concerns in a point-by-point manner below, and hope that you will find the added information suitable and sufficient for publication. The original comments are quoted in blue and the changes in the revised manuscript have been highlighted in red.

General Comments:

#1. The section starting L152 (Weakened Somali Jet... is now somewhat redundant, as it doesn't bring much new compared to the section before. In the previous section, the weakening of the jet and the effects on the upwelling are already presented. So you could simply merge this section with the previous one.

Our response: Thanks for your comment. We have merged this section with previous one.

#2. I like that some sections have the main message formulated as a heading. Perhaps this style could be applied to all headings.

Our response: Thanks for raising this important point. We have further applied the style to all headings as follows:

- 1) Changed heading of “Moisture budget analysis of changes in SASM rainfall” to “Dynamic effects induced by atmospheric circulation anomalies control the changes in SASM rainfall” (Lines 168-169)
- 2) Changed heading of “The effects of the $p\text{CO}_2$ forcing versus topographic forcing” to “The effects of the $p\text{CO}_2$ forcing show no decoupling effect” (Line 303)

#3. Other comments.

#(1) L229 "consistent with a previous study" -> "a" is missing

Our response: Revised.

#(2) L235 "such as a positive precipitation..." -> "a" is missing

Our response: Revised.

#(3) L254 " a lower East African Highlands" -> delete the "a"

Our response: Revised.

#(4) L290 it is "enhanced southerly winds"

Our response: Revised.

#(5) L345 "the ... Somali Jet ..." -> add "the"

Our response: Revised.

#(6) L349 it is "increasing local ..."

Our response: Revised.

#(7) L361 it is "Himalaya-Tibetan"

Our response: Revised.

#(8) Fig1: why is there no bar for M05?

Our response: Thanks for your comment. In fact, there is a bar for M05. However, because the changes in the ratios of SASM rainfall are much smaller for the MT05 simulation (0.2%) compared to the MT25 (15.3%) and MT15 (11.4%) simulations, the bar for M05 is not clearly visible in Figure 1h.

#(9) Suppl. Fig 13 seems to be a bit blurry.

Our response: Thanks for your reminder. We have reviewed all the figures in the Supplementary materials and have replaced them with clearer versions.

Reviewer #2 (Remarks on code availability):

The code is now available, but I have not checked it, because I'm not familiar with ncl. A README file is not included

Our response: Thanks for your reminder. We have provided the README file in the new submission on Zenodo at <https://doi.org/10.5281/zenodo.15605414>.

Reply to Reviewer #3's comments

Remarks to the Author:

I have reviewed the authors' responses and the revised manuscript. Overall, the authors have addressed most of my concerns satisfactorily. However, one important issue remains unresolved—the matter of the decoupling between monsoon winds and South Asian Summer Monsoon (SASM) rainfall.

My original concern was that the authors seem to suggest a decoupling between monsoon winds and SASM rainfall during the Miocene, while their moisture budget analysis clearly indicates a strong relationship between atmospheric dynamics and SASM precipitation. Based on their response, it appears that the authors are actually referring to a decoupling between the Somali jet and SASM rainfall, rather than between large-scale monsoon winds and SASM rainfall.

In fact, their results show that large-scale monsoon winds—modulated by Miocene topography—play a key role in driving changes in SASM rainfall, independent of changes in the Somali jet. This implies that the modern-day coupling between the Somali jet and SASM rainfall does not necessarily hold on geological timescales.

The current phrasing—"decoupling between monsoon winds and SASM rainfall"—used in both the title and the main text is therefore misleading and technically inaccurate. I strongly recommend that the authors revise this wording to "decoupling between the Somali jet and SASM rainfall".

With this correction, I am happy to recommend the manuscript for publication.

Our response: We sincerely appreciate your professional and valuable comments on our manuscript. Following your suggestions, we have thoroughly reviewed the entire manuscript and phrased “decoupling between monsoon winds and SASM rainfall” to “decoupling between the Somali jet and SASM rainfall”. This is clarified in Lines 1-2, 36, 80, 93, 97, 124, 148, 312, 333, 358, and 449 of the revised manuscript.

Reviewer's comments on
“Impacts of African topography on the decoupling of South Asian summer monsoon winds and rainfall during the Miocene ”

by Z. Han et al.

A) General comments

Recent studies have shown that monsoon winds in the Arabian Sea and the South Asian monsoon are positively correlated on interannual timescales but not necessarily on longer orbital timescales. In this study, Z. Han et al. extend this concept, suggesting a similar decoupling over much longer tectonic timescales, supported by proxy evidence. They propose a mechanism based on climate model experiments, linking the uplift of African orography during the middle to late Miocene to the South Asian Summer Monsoon (SASM). According to their model, the less elevated East African orography induces anomalous latent heating in the atmosphere, triggering circulation changes via Gill's response. These circulation shifts alter low-level monsoon winds, with the resulting SST anomalies (resembling a positive IOD) further amplifying both monsoon winds and rainfall.

Initially, I was very interested in the study and looked forward to reading the manuscript. However, the results seem to suggest quite the opposite of the authors' primary claim—that monsoon winds and rainfall are, in fact, decoupled on these tectonic timescales. There are three key issues:

1. The regions chosen to represent monsoon winds do not overlap with the proxy regions, and in the proxy regions, the changes in winds shown by the climate model experiments are not robust.
2. Their moisture budget analysis indicates that SASM rainfall changes are strongly linked to the dynamic component (i.e., monsoon winds), directly contradicting their central hypothesis of decoupling.
3. The authors interpret upwelling proxies as being linked to monsoon wind speed. However, it is well known that changes in wind speed, the latitude, or width of the low-level jet (LLJ) can all influence upwelling. The correct interpretation should be that upwelling is not coupled with rainfall.

Due to these issues, I do not find the author's primary claim convincing. However, I find their detailed mechanism linking African orography to SASM rainfall quite interesting and could offer an alternative focus for the study. Unfortunately, this connection between African orography and SASM rainfall is already well-established in the literature. Moreover, the connection between changes in African rainfall and its impact on upwelling in the northern Arabian Sea and SASM rainfall has been demonstrated in previous studies. While the authors' mechanism is well-constructed, it lacks the novelty required for publication in a high-impact journal like Nature Communications.

For these reasons, I recommend rejecting the manuscript in its current form. I encourage the authors to refine their focus on the proposed mechanism, enhance the rigor of their experiments. The current experiments need refinement to better align with their claims, and addressing these issues will significantly improve the paper's overall impact. My major and minor concerns, outlined below, should provide helpful guidance for their revisions.

B) Major concerns:

1. **Choice of regions for analysis:** The authors use proxy data to demonstrate that upwelling in the northern Arabian Sea and SASM rainfall were not coupled during the middle to late Miocene monsoon evolution. However, in their model analysis, they selected a study region significantly south of the proxy region. Analyzing this area in isolation leads them to conclude that the low-level jet (LLJ) wind speed and monsoon rainfall are not coupled. However, they overlook an important detail: the LLJ, and consequently the monsoon winds, show a significant increase in speed in the region immediately west of their chosen area. This stronger LLJ impacts the western coast of India, producing substantial rainfall over the Western Ghats and a slight increase in rainfall across mainland India.

Therefore, the authors' claim that the LLJ/monsoon winds and SASM rainfall are decoupled is not fully supported. In fact, a more careful examination of the wind changes in the proxy region reveals a much more complex response, which the authors have not addressed in their analysis.

Furthermore, I find the authors' choice of region for defining the SASM problematic. The precipitation response plots indicate a pronounced change in rainfall along the western coast, extending into the adjacent Arabian Sea, while most of inland India—the core SASM region—shows insignificant changes in rainfall. The rainfall signal used by the authors to support their claim is predominantly a result of offshore precipitation along the western coast of India, which is a direct consequence of changes in the LLJ/monsoon winds.

Thus, I find the authors' conclusion in the model analysis unconvincing and sensitive to their choice of region. Their claim appears somewhat overstated and does not fully align with the complexity of the wind-rainfall relationship seen in the data.

2. **Moisture budget analysis:** My concerns are further heightened by the authors' moisture budget analysis. Their results clearly indicate that changes in precipitation over the SASM region are driven by the dynamic component (i.e., changes in monsoon winds). This presents a significant contradiction: on one hand, the authors claim that monsoon rainfall and winds are decoupled, yet their own analysis shows that the changes in rainfall are directly linked to wind dynamics. This inconsistency undermines their central argument and raises questions about the validity of their conclusions.
3. **Interpretation of upwelling proxies:** In this study, the authors claim that monsoon winds and rainfall are decoupled, based on proxy records. However, the correct interpretation should be that upwelling in the northern Arabian Sea and rainfall are decoupled. Upwelling in this region is influenced by various factors, including wind speed, the latitude, and width of the low-level jet (LLJ). Even slight changes to one or more of these LLJ characteristics can modulate upwelling and, by extension, rainfall (see recent studies on impact of orbital forcing on upwelling and rainfall for example). Therefore, the primary claim of this study—that monsoon winds and rainfall are decoupled—seems questionable. The relationship between winds, upwelling, and rainfall is more complex than the authors suggest, and their conclusions may oversimplify these dynamics. This is further supported by their figures of SST anomalies. The SST anomalies during the middle to late Miocene are negative in the northern Arabian Sea, indicating enhanced upwelling. This region, which spans the coast of Yemen and Oman and extends into the open sea, encompasses most of the proxy locations. This again indicates that upwelling-rainfall are decoupled not necessarily monsoon winds and rainfall.
4. **Missing experiments:** The authors conducted experiments where global orography is set to pre-industrial levels, except over Africa, where it is adjusted to 25 Ma, 15 Ma, and 5 Ma. However, for a well-constructed study, I would first expect results from experiments where global orography is set to match these time periods fully. Following that, the authors should demonstrate that modifying only African orography is sufficient to capture the majority of wind and rainfall responses. This approach would strengthen their case for the impact of African orography—although, I still believe it won't support the claim that monsoon winds and rainfall are decoupled.

C) Minor points:

1. **Evolution of SST in the Indian Ocean:** The development of positive SST anomalies in the western equatorial Indian Ocean is a critical component of the authors' proposed mechanism. However, the manuscript does not provide sufficient explanation for why or how these anomalies form. A detailed examination of the seasonal cycle of SST and winds in this region could offer valuable insights into this process. Additionally, understanding the impact of East African orography on the seasonal cycle (premonsoon in particular) of the East African monsoon may be key to explaining the emergence of these positive SST anomalies during the monsoon season. Addressing these points would significantly strengthen the authors' argument.
2. Line 147: "Fig. 11" it took me sometime to understand that it is not Fig. eleven. Maybe authors should refer to it in capital letter to avoid confusion.